# PROTEIN STRUCTURE TOKENIZATION VIA GEOMETRIC BYTE PAIR ENCODING

**Michael Sun**[1,2]**, Weize Yuan**[3]**, Gang Liu**[4]**, Wojciech Matusik**[1]**, Marinka Zitnik**[2]
[1]MIT CSAIL    [2]Harvard Medical School    [3]Apple    [4]Notre Dame
{msun415,wojciech}@csail.mit.edu,   marinka@hms.harvard.edu

## ABSTRACT

Protein structure is central to biological function, and enabling multimodal protein models requires joint reasoning over sequence, structure, and function. A key barrier is the lack of principled protein structure tokenizers (PSTs): existing approaches fix token size or rely on continuous vector codebooks, limiting interpretability, multi-scale control, and transfer across architectures. We introduce GEOBPE, a geometry-grounded PST that transforms continuous, noisy, multi-scale backbone conformations into discrete "sentences" of geometry while enforcing global constraints. Analogous to byte-pair encoding, GEOBPE generates a hierarchical vocabulary of geometric primitives by iteratively (i) clustering Geo-Pair occurrences with k-medoids to yield a resolution-controllable vocabulary; (ii) quantizing each Geo-Pair to its closest medoid prototype; and (iii) reducing drift through differentiable inverse kinematics that optimizes boundary glue angles under an $SE(3)$ end-frame loss. GEOBPE offers compression ($>10\times$ reduction in bits-per-residue at similar distortion rate), data efficiency ($>10\times$ less training data), and generalization (maintains test/train distortion ratio of $1.0 - 1.1$). It is architecture-agnostic: (a) its hierarchical vocabulary provides a strong inductive bias for coarsening residue-level embeddings from large PLMs into motif- and protein-level representations, consistently outperforming leading PSTs across 12 tasks and 24 test splits; (b) paired with a transformer, GEOBPE supports unconditional backbone generation via language modeling; and (c) tokens align with CATH functional families and support expert-interpretable case studies, offering functional meaning absent in prior PSTs. Code is available at https://github.com/shiningsunnyday/PT-BPE.

## 1 INTRODUCTION

Protein language models (PLMs) trained on large sequence databases capture evolutionary constraints (Rives et al., 2021) and support de novo sequence design (Lin et al., 2023b), but they do not explicitly model fold geometry and may underperform on tasks where function depends on structural interactions (Abramson et al., 2024; Gelman et al., 2025). In natural language processing, byte-pair encoding (BPE) constructs a vocabulary by iteratively merging the most frequent symbol pairs, producing a hierarchical representation of text (Larsson & Moffat, 2002). Despite BPE's success on sequential data, there is no geometric analog that can encode and decode protein backbone conformations. The central difficulty is discretizing continuous, noisy structural variability while preserving global consistency. Because protein folds are organized into modular substructures (Petsko & Ringe, 2004), a protein structure tokenizer should (a) build a hierarchical vocabulary of structural motifs and (b) segment folds into hierarchical decompositions, producing symbolic and interpretable representations of backbone geometry.

Recently, vector-quantized variational autoencoders (VQ-VAEs) have become the most popular class of protein structure tokenizers (PSTs), as adopted by ESM3 (Hayes et al., 2025) and others. VQ-VAEs learn an autoencoder that compresses and reconstructs a protein structure with $N$ residues to and from $N$ quantized latent codes, which are discrete "words" drawn from a vocabulary of learnable embeddings (Van Den Oord et al., 2017). While powerful, VQ-VAEs lack the efficiency, interpretability and modularity of BPE tokenizers: (1) using a fixed codebook can create performance bottlenecks and imbalance token usage frequency, handicapping downstream performance (Yuan et al., 2025); (2) using vectors as tokens over real data hinders interpretability, as rows of a

2D matrix do not capture the hierarchical relationships between sub-words like in BPE; (3) lastly, fixing all tokens to have the same size prevents multi-scale resolution, which is key to tasks that identify naturally occurring higher-level functional activity which span variable residue lengths.

**Present work.** We develop Geometric Byte-Pair Encoding (GEOBPE), a tokenizer that discretizes continuous protein backbones into symbolic "sentences" of structural motifs while learning a hierarchical vocabulary. The design is motivated by two requirements: (i) protein folds contain modular substructures that should be captured as reusable tokens, and (ii) discrete approximations must preserve global geometric consistency. To meet these requirements, GEOBPE alternates between local updates and global corrections. At each step, frequent motif pairs are clustered with k-medoids and replaced by representative prototypes, recursively building higher-order motifs. This local quantization inevitably introduces geometric drift, which GEOBPE corrects by optimizing boundary glue angles through differentiable inverse kinematics under an $SE(3)$ end-frame loss. The output after each iteration is a segmentation of the backbone into quantized motifs and glue parameters; the sequence of iterations yields a hierarchical decomposition of the fold, represented as a merge tree of structural motifs (Fig. 1). Our contributions are as follows: ① GEOBPE is the first geometry-grounded BPE analog for protein backbones, which builds a hierarchical vocabulary of motifs and tokenizes structures through an alternating global-local decomposition with glue-aware reconstruction. ② On benchmark datasets, GEOBPE traces a smooth Pareto front of compression-distortion tradeoffs, achieving up to 0.27-0.36× the bits-per-residue of ProToken and strong out-of-distribution generalization (test/train RMSD ratio 1.16-1.28 vs. 6.4× for VQ-VAE). It also matches downstream accuracy when trained on as little as 1% of the pretraining data. ③ Hierarchical vocabularies from GEOBPE improve representation quality on tasks such as binding site prediction and fold classification, and a transformer trained on its tokens enables unconditional backbone generation. ④ Tokens align with CATH domain annotations and are supported by expert case studies, providing functional protein insights and multi-resolution interpretability.

**Figure 1:** GEOBPE tokenizes a protein into discrete motifs linked by boundary glue angles and learns a hierarchical vocabulary of frequent structural primitives via k-medoids and recursively merging Geo-Pairs; at each step glue angles are optimized with differentiable inverse kinematics to preserve the global fold. Tokenization yields a merge tree that provides multi-resolution and interpretable representations of protein structure.

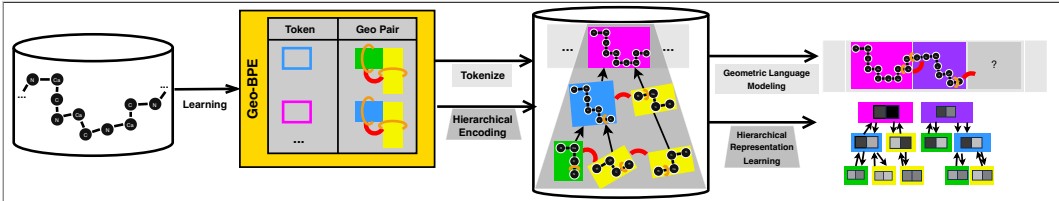

## 2 RELATED WORK

**Protein Structural Alphabets.** Structural alphabets approximate protein folds as successions of geometric motifs (Branden & Tooze, 2012). de Brevern et al. (2000) introduced 16 five–residue protein blocks from Protein Data Bank (PDB) structures, assigning fragments by RMSD. Later work showed that over 90% of residues can be covered by such alphabets (de Brevern et al., 2002) and analyzed their quality and specificity (de Brevern, 2005). Alphabet strings provide 1D encodings of 3D geometry, enabling the use of sequence alignment for fold analysis and prediction (Mahajan et al., 2015; Vetrivel et al., 2017). Camproux et al. (1999) proposed 12 building blocks via Hidden Markov Models (HMMs) and extended it to capture whole-protein conformational variability (Camproux et al., 2004). HMMs use inter-alpha-carbon distances within four residues as observed variables. Broader tertiary descriptors, such as inter-residue distances or moment invariants (Durairaj et al., 2020), capture non-contiguous context; Mackenzie (2016) found ∼600 motifs describe more than half of structural space ($39 \cdot 10^6$ conformations), indicating variability collapses into limited modes. Such descriptors extend to protein-level retrieval and classification (Durairaj et al., 2020; Van Kempen et al., 2024; Barrio-Hernandez et al., 2023). GEOBPE builds on these insights by treating structural motifs as extensible primitives and dynamically adjusting alphabet size and token resolution, unlike fixed structural alphabets.

**Protein Structure Tokenizers.** Modern PSTs, most notably VQ-VAEs, construct structural alphabets by training deep autoencoders with vectorized codebooks that map continuous structure to dis-

crete codes (Van Den Oord et al., 2017). Building on this idea, FoldSeek (Van Kempen et al., 2024) introduced 3Di alphabets (20 discrete codes learned with VQ-VAE) that compress local structural features for efficient search and homology detection. Subsequent works integrate 3Di alphabets with PLMs: Heinzinger et al. (2024) translate between 3Di and amino acid sequences; Su et al. (2023) define "3Di-residue" tokens and show pretraining with this vocabulary improves prediction; and Li et al. (2024) use disentangled attention to jointly model 3Di and residue tokens with a structure quantization module. End-to-end VQ-VAEs avoid predefined descriptors by training equivariant encoders and decoders to tokenize structure directly, achieving near-perfect reconstruction but at high computational cost. Large-scale efforts such as ESM3 (Hayes et al., 2025), trained on 236 million structures, highlight the central role of tokenizers in scaling multimodal PLMs. Recent work benchmarks tokenizer performance itself: AIDO.St and ProTokens show that stronger compression improves retrieval (Van Kempen et al., 2024; Zhang et al., 2024c) but reduces reconstruction quality, and both Zhang et al. (2024c) and Lin et al. (2023a) integrate tokenizers tightly with transformers. GEOBPE differs by using its hierarchical vocabulary as an inductive bias for representation learning and by supporting geometry-grounded language modeling without latent space vector quantization.

**Byte-Pair Encoding for Biological Data.** BPE underlies modern language models and has been applied to biological sequences with mixed outcomes. On genomes, BPE achieves superior compression and improves over k-mers in language models (Dotan et al., 2024; Zhou et al., 2023), though Nguyen et al. (2023) find the opposite using Hyena. For functional tasks, BPE often performs best (Dotan et al., 2024), while on nucleotide-resolution tasks it can underperform (Lindsey et al., 2025). These results indicate tokenizer utility depends on task scale and architecture, motivating GEOBPE's architecture-agnostic design and multi-scale resolution. Linguistic differences between text and biological sequences further complicate direct transfer: BPE tokens do not align with domain boundaries (Suyunu et al., 2024) or regulatory motifs (Lindsey et al., 2025). Other studies emphasize the importance of vocabulary design, reduced amino acid alphabets impair structure prediction (Ieremie et al., 2024), while BPE vocabularies of 50–200 tokens are often optimal for sequence tasks (Tan et al., 2024). Overall, existing tokenizers, including BPE, lack *versatility for protein structures*. GEOBPE extends BPE by grounding tokenization in geometry, exposing parameters for quantization, vocabulary, and efficiency, while uniquely providing fine-grained resolution control and a hierarchical motif vocabulary.

## 3 METHODS

We first establish backbone geometry notations in Sec. 3.1. Sec. 3.2 presents the GEOBPE algorithm, detailing its components for motif clustering, adaptive quantization, and glue-aware refinement. Finally, Sec. 3.3 formalizes the principles that an ideal protein structure tokenizer should satisfy and evaluates how GEOBPE meets them.

### 3.1 NOTATION & PRELIMINARIES

**Global Backbone Formulation.** Let a protein backbone $t^{(\tau)}$ with $N^{(\tau)}$ residues be represented by the Cartesian coordinates $\{(N_i, \mathrm{CA}_i, C_i) \in \mathbb{R}^{3\times3}\}_{i=1}^{N^{(\tau)}}$ of backbone atoms (oxygen and $C_\beta$ omitted). Define bond lengths, bond angles, and dihedrals:

$$\ell_i^{N-CA} = \|N_i - \mathrm{CA}_i\|, \ \ell_i^{CA-C} = \|\mathrm{CA}_i - C_i\| \ ; \ \ell_i^{C-N} = \|C_i - N_{i+1}\|.$$
$$\theta_i^{NCAC} = \angle(N_i, \mathrm{CA}_i, C_i) \ , \ \theta_i^{CACN} = \angle(\mathrm{CA}_i, C_i, N_{i+1}) \ \ \theta_i^{CNCA} = \angle(C_i, N_{i+1}, \mathrm{CA}_{i+1}).$$
$$\psi_i = \angle(N_i, \mathrm{CA}_i, C_i, N_{i+1}), \ \omega_i = \angle(\mathrm{CA}_i, C_i, N_{i+1}, \mathrm{CA}_{i+1}), \ \phi_i = \angle(C_i, N_{i+1}, \mathrm{CA}_{i+1}, C_{i+1}).$$

We annotate these definitions in a toy ($N^{(\tau)} = 2$) example in Fig. 2 (top). The full internal representation thus contains $3N^{(\tau)}-1$ bond lengths, $3N^{(\tau)}-2$ bond angles, and $3N^{(\tau)}-3$ dihedrals and is invariant to any $(R, t) \in \mathrm{SE}(3)$.

**Local Formulation (Bond–Residue).** For residue $i$ we define the *bond–residue* as the ordered triple $(N_i - \mathrm{CA}_i), (\mathrm{CA}_i - C_i), (C_i - N_{i+1})$ together with its internal angles. For $i < N^{(\tau)}$ this includes the lengths $\ell_i^{N-CA}$, $\ell_i^{CA-C}$, $\ell_i^{C-N}$, the bond angles

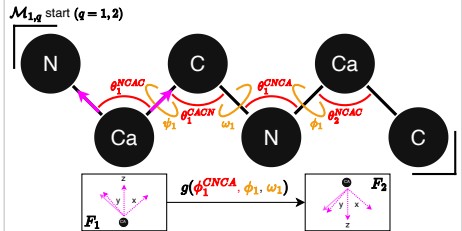

**Figure 2:** Toy backbone, with internal angles, glues $T_1$, motif $\mathcal{M}_{1:2}$ and per-link transform $G_1$.

$\theta_i^{NCAC}$, $\theta_i^{CACN}$, and the peptide dihedral $\psi_i$ about $CA_i - C_i$. For $i = N^{(\tau)}$, it includes only bond lengths $\ell_{N^{(\tau)}}^{N-CA}$, $\ell_{N^{(\tau)}}^{CA-C}$, and angle $\theta_{N^{(\tau)}}^{NCAC}$ (the $(C-N)$ bond, $\theta_{N^{(\tau)}}^{CACN}$, $\theta_{N^{(\tau)}}^{CNCA}$, and $\{\psi, \omega, \phi\}_{N^{(\tau)}}$ dihedrals are absent).

**Glue Parameters Between Neighboring Bond–Residue.** Neighboring bond–residues $i$ and $i+1$ are connected by a set of *glue* angles that place the bonds of residue $i+1$ relative to residue $i$. These are $\Gamma_i = \left\{ \theta_i^{CNCA}, \ \phi_i, \ \omega_i \right\}$, i.e., one bond angle $\theta_i^{CNCA}$ (to place $N_{i+1}$–$CA_{i+1}$) and two dihedrals $\phi_i$ and $\omega_i$ (to orient $CA_{i+1} - C_{i+1}$ and the peptide plane). We adopt $(\omega, \phi)$ here to emphasize the two independent dihedral DOFs spanning the peptide and CA torsions.

**Motif Formulation.** A *bond–residue motif* $\mathcal{M}_{p:q}$ is a contiguous block of bond–residues $i = p, \ldots, q$ $(1 \le p \le q \le N^{(\tau)})$. Its internal parameter set is the union over the internal bond lengths and angles of bond-residues $p, \ldots, q$ together with the internal glue angles $\{\Gamma_i\}_{i=p}^{q-1}$ that connect consecutive bond–residues inside the motif. Given $q \le r \le N^{(\tau)}$, we obtain a *Geo-Pair occurrence* $(\mathcal{M}_{p:q}, \Gamma_q, \mathcal{M}_{q:r})$ from the internal parameters of $\mathcal{M}_{p:q}$ and $\mathcal{M}_{q+1:r}$, plus the *external* glue angles $\Gamma_q$ connecting the last and first bond-residues of $\mathcal{M}_{p:q}$ and $\mathcal{M}_{q+1:r}$.

**Entry/Exit Frames.** For residue $i$, define $F_i = (R_i, t_i) \in SE(3)$ with origin $t_i = CA_i$ and axes chosen so that the x-axis points from $CA_i$ toward the $C_i$, the $y$-axis is the normalized component of the $CA_i - N_i$ direction orthogonal to $x$, and the $z$-axis completes a right-handed triad.

**Per-Link Transform.** Define the transform between consecutive residue frames $G_i := F_{i+1} F_i^{-1} \in SE(3)$. By construction, $G_i$ is a deterministic function of the internal coordinates local to the link $i \to i+1$, namely $G_i = g(\ell_i^{CA-C}, \ell_i^{C-N}, \ell_{i+1}^{N-CA}, \theta_i^{NCAC}, \theta_i^{CACN}, \theta_i^{CNCA}, \psi_i, \omega_i, \phi_i)$, and, in particular, depends on the *glue set* $\Gamma_i = \{\theta_i^{CNCA}, \phi_i, \omega_i\}$, illustrated in Fig. 2 (bottom).

**Entry/Exit Transforms.** For a motif $\mathcal{M}_{p:q}$, define $F_{p:q}^{\text{entry}} := F_p, F_{p:q}^{\text{exit}} := F_q$. The *internal* entry$\to$exit transform is $T_{p:q}^{\text{int}} = F_{p:q}^{\text{exit}} (F_{p:q}^{\text{entry}})^{-1} = (G_{q-1}) \cdots (G_p)$, which depends only on the internal coordinates of $\mathcal{M}_{p:q}$. The *external glue* transform between consecutive motifs $\mathcal{M}_{p:q}$ and $\mathcal{M}_{q+1:r}$ is precisely the boundary link $T_{q \to q+1}^{\text{glue}} = F_{q+1} F_q^{-1} = G_q$, and is parameterized by the glue set $\Gamma_q$ (and the adjacent three bond lengths).

**Figure 3:** (Top) GeoBPE tracks a Geo-Pair Encoding, a dictionary mapping Geo-Pair keys to occurrences at all times. Each step pops the most frequent Geo-Pair key, gathers the occurrences (◖,◗,◖,◗,...) and fixes $K$ prototypes (▮,▮,▮) to add to $\mathcal{V}$. All occurrences are quantized to the closest prototype (e.g. ◖ $\to$ ▮). Glue angles (⌒,⌣) are optimized to correct for the drift introduced. (Bottom) Toy example with two backbones; we initialize residue-orientation modes using two prototypes (▯,▯), pop the frequent Geo-Pair (▭), quantize occurrences (▭$\to$ ▬), and optimize glue angles.

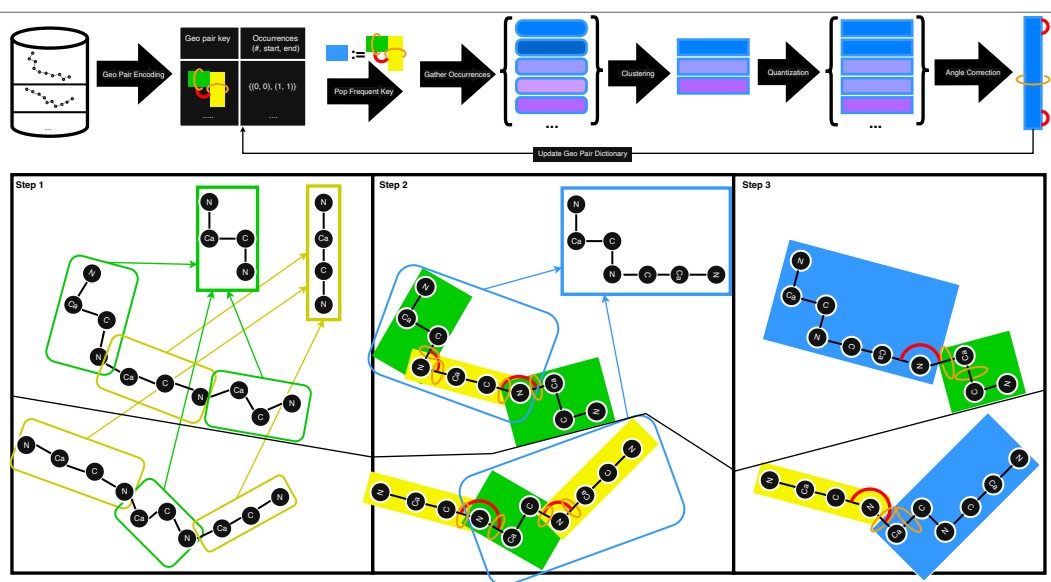

**Core GeoBPE notation.** We define the main objects used throughout the algorithmic description. We index training backbones by $\tau = 1, \ldots, T$, writing the $\tau$-th backbone as $t^{(\tau)}$ with

$N^{(\tau)}$ residues. A *segmentation* of $t^{(\tau)}$ into bond–residue motifs is the ordered tuple $\mathcal{P}^{(\tau)} = \left( \mathcal{M}_{p_1:q_1}^{(t_\tau)}, \ldots, \mathcal{M}_{p_{M_\tau}:q_{M_\tau}}^{(t_\tau)} \right)$, with $1 = p_1 \leq q_1 < p_2 \leq \cdots \leq q_{M_\tau} = N^{(\tau)}$. The corresponding *merge hierarchy* $\mathcal{F}^{(\tau)}$ is a binary forest whose frontier leaves, in order, equal $\mathcal{P}^{(\tau)}$; each internal node represents a merged motif and stores its span $[p{:}q]$.

**Geo-pair keys and occurrences.** Given two adjacent motifs $(\mathcal{M}_{p:q}, \mathcal{M}_{q+1:r})$ and their boundary glue $\Gamma_q$, we define a canonical, hashable *geo-pair key* $\kappa = \text{COMPUTEGEOKEY}(\mathcal{M}_{p:q}, \mathcal{M}_{q+1:r})$ (Alg. 22). For each key $\kappa$ we collect its *occurrence set* $\mathcal{O}(\kappa)$ consisting of all such adjacent motif pairs across the dataset.

**Prototypes and vocabulary.** For a geo-pair key $\kappa$, GEOBPE clusters its occurrences and stores a small set of representative prototypes $\mathcal{A}_\kappa = \{\Pi_j^{(\kappa)}\}_{j=1}^{K_{|\kappa|}}$, where each $\Pi_j^{(\kappa)}$ is the internal-parameter tuple of a medoid occurrence and $K : \mathbb{Z}^+ \setminus \{1\} \mapsto \mathbb{Z}^+$ controls how many prototypes by motif (bond) length. The *vocabulary* is the map $\mathcal{V} : \kappa \mapsto \mathcal{A}_\kappa$, initially containing residue-level codebooks $\mathcal{A}_3, \mathcal{A}_2$ and growing as new geo-pair keys are introduced.

**Geo-pair dictionary and priorities.** At any time GEOBPE maintains a *priority-ordered* dictionary $\mathcal{D}$ that maps each key $\kappa$ to its occurrence set $\mathcal{O}(\kappa)$. Keys are ordered by tuples

$$\pi(\kappa) = \big( \rho(\kappa), -|\mathcal{O}(\kappa)|, \kappa \big), \quad \rho(\kappa) = \mathbf{1}[\kappa \notin \text{dom}(\mathcal{V})],$$

so that compressible keys with existing prototypes ($\rho = 0$) are popped first, followed by a new key with the largest count $|\mathcal{O}(\kappa)|$ per iteration.

## 3.2 GEOBPE ALGORITHM

GEOBPE (Algo. 1, Fig. 3) is organized around four components: (1) clustering motif (individual bond-residues once at the start, Geo-Pairs every step thereafter) occurrences into representative structural prototypes, (2) maintaining an ordered map to track frequent Geo-Pairs, (3) adaptively hard-quantizing noisy Geo-Pairs to their assigned prototypes, and (4) applying rigid-body refinement to enforce global geometric consistency.

Components (1)-(4) are designed to answer three new key questions when re-interpreting BPE to work with continuous backbone geometry rather than discrete bytes: (a) how do we *ground* continuous backbone states to discrete keys for Geo-Pair *counting*, (b) how do we *update* the backbone states once we have popped the most frequent Geo-Pair, (c) how do we synchronize the Geo-Pair dictionary with how we updated the backbone states. Component (1) answers to (a), (3) & (4) to (b), and (2) to (c). The guiding question encompassing (a)-(c) is: **what is the exact relationship between the Geo-Pair dictionary** (needed for discrete BPE operations) **and the continuous backbone states** (which should both reflect new keys and preserve original fidelity)? GEOBPE implements a two-way connection through four stages: grounding, in which continuous motif states define discrete prototype keys; quantization, in which internal parameters are overwritten with those of the assigned prototypes; rigid-body refinement, in which backbone internal states self-correct to minimize global distortion; and synchronization, in which the Geo-Pair dictionary is re-synchronized to reflect the high-fidelity backbone states after quantization and refinement.

**(1) Extracting Dominant Modes from a Set of Motif Occurrences.** The core subroutine invoked by GEOBPE is Algo. 6, which clusters a set of length $L$ raw backbone fragments into $K$ representative prototypes. This induces a hard quantization of the fragment space, since every possible occurrence is assigned to exactly one prototype. Because RMSD defines a metric over fragments, the clustering yields a Voronoi partition of this space. Importantly, the medoids are themselves observed fragments, so each quantized symbol retains a concrete structural interpretation: it represents the closest empirically observed conformation, providing a denoised approximation of local variability. Each time we quantize, we substitute every non-medoid occurrence by its assigned medoid, replacing all internal parameters with the medoid's internal parameters, making it an exact copy of that medoid (same length and per-position angles).

**(2) Constructing a Structural Motif Alphabet.** GEOBPE begins by quantizing all bond-residues and glue angles (Algo. 18) and building an *ordered* map $\mathcal{D}$ of discrete geo-pair grounding keys $\kappa$ to occurrences $O(\kappa)$ (Algo. 21, see Core GeoBPE notation). In each iteration (Algo. 9), GEOBPE pops

---

**Algorithm 1** GEOBPE: Protein structure tokenizer with geometric byte-pair encoding

---

**Require:** Backbones $\{t^{(1)}, \ldots, t^{(T)}\}$ with lengths $N^{(\tau)}$; optional backbones $\{t^{(\xi)}\}$ to tokenize; residue codebook sizes $(K_3, K_2)$; glue-IK weights $(w_R, w_t)$; maximum merge iterations $S_{\max}$.

**Ensure:** Final vocabulary $\mathcal{V}$ (motif prototypes), final segmentations $\{\mathcal{P}^{(\tau)}\}$, final merge hierarchies $\{\mathcal{F}^{(\tau)}\}$, and the priority-ordered geo-pair map $\mathcal{D}$.

1: **Empirical quantizer estimation (once).** Collect samples over all backbones for the 9 types $\{\ell^{N-CA}, \ell^{CA-C}, \ell^{C-N}\}, \{\theta^{NCAC}, \theta^{CACN}, \theta^{CNCA}\}, \{\phi, \psi, \omega\}$. Wrap angles to $[0, 2\pi)$ and build circular histograms with edges $0 = \beta_0 < \cdots < \beta_B = 2\pi$ that tile the circle; define $Q$ by snapping to bin centers. For lengths, build linear histograms and snap to centers.

2: **Per-residue initialization** (Algo. 18). Cluster interior and terminal bond–residues via RMSD_PARTITION to obtain codebooks $\mathcal{A}_3, \mathcal{A}_2$; overwrite each residue's internals by its assigned prototype. Set the initial segmentation for each backbone:

$$\mathcal{P}^{(\tau)} = (\mathcal{M}_{1:1}^{(t_\tau)}, \ldots, \mathcal{M}_{N^{(\tau)}:N^{(\tau)}}^{(t_\tau)}).$$

**Initialize hierarchies:** for each $\tau$, create a binary forest $\mathcal{F}^{(\tau)}$ whose leaves are the bond–residue motifs $\mathcal{M}_{i:i}^{(t_\tau)}$, in order; its frontier equals $\mathcal{P}^{(\tau)}$. Initialize the vocabulary with base prototypes:

$$\mathcal{V} \leftarrow \{\text{residue-level keys} \mapsto \mathcal{A}_3, \mathcal{A}_2\}.$$

3: **Global glue refinement** (Algo. 12). Optimize all boundary glues $\Gamma_i = \{\theta_i^{CNCA}, \omega_i, \phi_{i+1}\}$ via differentiable FK with $(w_R, w_t)$; snap each to the nearest bin center using $Q_{\theta^{CNCA}}, Q_\omega, Q_\phi$.

4: **Build the priority-ordered geo-pair map** (Algo. 21). Using the frontier leaves of each $\mathcal{F}^{(\tau)}$ (equivalently, $\mathcal{P}^{(\tau)}$), construct the occurrence sets $\mathcal{O}(\kappa)$ and insert:

$$\mathcal{D}\big[(\rho(\kappa), -|\mathcal{O}(\kappa)|, \kappa)\big] \leftarrow \mathcal{O}(\kappa), \quad \rho(\kappa) = \mathbf{1}[\kappa \notin \text{dom}(\mathcal{V})].$$

5: **BPE loop – calls (Algo. 9) each step.**

6: **for** $s = 1$ **to** $S_{\max}$ **do**

7: $\quad (\{\mathcal{P}^{(\tau)}\}, \{\mathcal{F}^{(\tau)}\}, \mathcal{D}, \mathcal{V}) \leftarrow \text{STEP}\big(\{\mathcal{P}^{(\tau)}\}, \{\mathcal{F}^{(\tau)}\}, \mathcal{D}, \mathcal{V}, \{Q_{\theta^{CNCA}}, Q_\omega, Q_\phi\}, (w_R, w_t)\big)$

8: **end for**

9: **Tokenize new/unseen backbones (Algo. 10)** for each $\xi$,

$$(\mathcal{P}^{(\xi)}, \mathcal{F}^{(\xi)}) \leftarrow \text{TOKENIZE}\big(t^{(\xi)}, \mathcal{A}_3, \mathcal{A}_2, \mathcal{V}, \{Q_{\theta^{CNCA}}, Q_\omega, Q_\phi\}\big)$$

10: **return** $\mathcal{V}, \{\mathcal{P}^{(\tau)}\}, \{\mathcal{F}^{(\tau)}\}, \mathcal{D}$ and (if given) $\{\mathcal{P}^{(\xi)}\}, \{\mathcal{F}^{(\xi)}\}$

---

the most frequent Geo-Pair key, runs Algo. 6 on mapped occurrences, quantizes the occurrences, runs rigid-body refinement, and updates $\mathcal{D}$ to account for the new quantized backbone states.

**(3) Multi-Resolution & Adaptive (Re-)Quantization.** One-time quantization is a lossy procedure and is only needed to index Geo-Pairs occurrences in the current step. Thus, each GEOBPE iteration can re-quantize occurrences by referencing the original, even if prior iterations have quantized the same regions already. This allows resolution to adapt based on the size of the motif (e.g., coarse-grained for smaller motifs, fine-grained for larger ones), providing precise control over compression-reconstruction tradeoffs (see App. A).

**(4) Minimizing Distortion via Rigid-Body Refinement.** Let $T_{i:j}^{\text{int}}$ denote the entry→exit SE(3) map of a motif $\mathcal{M}_{i:j}$ determined by its internal coordinates. For an occurrence $u$ with original motif $\mathcal{M}_{i_u:k_u}^{(t_u)}$, the rounding step replaces it by its assigned medoid segment:

$$\mathcal{M}_{i_u:k_u}^{(t_u)} \longrightarrow \mathcal{M}_{i_{\widehat{m}_{c(u)}}:k_{\widehat{m}_{c(u)}}}^{(t_{\widehat{m}_{c(u)}})},$$

where $\widehat{m}_{c(u)}$ is the medoid index returned by RMSD_PARTITION (an index into $\mathcal{S}$). Let $T_u^{\text{occ}} := T_{i_u:k_u}^{\text{int}}$ and $T_u^{\text{med}} := T_{i_{\widehat{m}_{c(u)}}:k_{\widehat{m}_{c(u)}}}^{\text{int}}$. Rounding thus replaces $T_u^{\text{occ}}$ by $T_u^{\text{med}}$, and the induced discrepancy $\Delta T_u := T_u^{\text{occ}}(T_u^{\text{med}})^{-1}$ is the *drift* introduced by quantization. If left uncompensated, products of such $\Delta T_u$ across a chain accumulate and move exit frames off their original targets. Each boundary provides 3 *gluing* degrees of freedom ($\Gamma_i$) that can absorb this drift. To exactly recover the

original exit (in the idealized case), the boundary transform at the link $i_u - 1 \to i_u$ should satisfy:

$$\overbrace{G_{i_u-1}^{\text{new}}}^{\text{opt vars}} T_u^{\text{med}} \approx G_{i_u-1}^{\text{orig}} T_u^{\text{occ}} \qquad \implies \qquad G_{i_u-1}^{\text{new}} \approx G_{i_u-1}^{\text{orig}} \Delta T_u,$$

where the quantization drift is $\Delta T_u := T_u^{\text{occ}} \left(T_u^{\text{med}}\right)^{-1}$. Since $G_{i_u-1}$ is controlled by only three gluing DOFs, we solve for $G_{i_u-1}^{\text{new}}$ in least squares via the end-frame fitting objective:

$$\mathcal{L}_u(\Gamma_{i_u-1}) = w_R \left\| \log\left((\widehat{R}_{k_u})^\top R_{k_u}^\star\right) \right\|_2^2 + w_t \left\| \widehat{t}_{k_u} - t_{k_u}^\star \right\|_2^2,$$

with forward kinematics $\widehat{F}_{k_u} = F_{i_u-1}^\star G_{i_u-1}^{\text{new}} T_u^{\text{med}}$, $F_{k_u}^\star = F_{i_u-1}^\star G_{i_u-1}^{\text{orig}} T_u^{\text{occ}}$. When quantizing many motifs on the same backbone, performing this optimization each time can become computationally prohibitive. Instead, we adopt a global (batch) alternative which treats all gluing DOFs as parameters, with a global end-frame fitting loss. This provides maximum flexibility in drift compensation. The algorithmic details are in Algo. 19 and 12.

**Transferring Hierarchical Inductive Biases.** GEOBPE adapts the receptive field of a base residue-level feature extractor $\Theta$ to that of the whole structure, connecting multiple scales through recursive aggregation. Algo. 1 emits merge hierarchies $\mathcal{F}$ as a forest: leaf nodes represent residues and parent nodes represent motifs. The key insight is to use $F$ as a recursive computation tree. Leaf nodes are initialized with pretrained features, then embeddings propagate up along the parent-child relations of $\mathcal{F}$ until the forest roots (aligned with $\mathcal{P}$); a final step aggregates the forest roots into a protein-level contextual embedding; then they are propagated down until the leaf nodes. The final leaf nodes output residue-level embeddings induced by the hierarchical $\mathcal{V}$ and informed by multi-scale GEOBPE tokenization. These features support supervised learning on fine-grained residue-level tasks (e.g., active site prediction) and coarse-grained global predictions (e.g., fold classification). See Algo. 15 for details.

## 3.3 PRINCIPLES OF PROTEIN STRUCTURE TOKENIZATION

Let $\mathcal{X} = (\mathbb{R}^{3\times3})^*$ be the space of backbone coordinate tensors and let $\mathcal{V}$ be a finite codebook. A tokenizer is a tuple $\mathsf{T} = (\mathcal{V}, \text{Enc}, \text{Dec})$: $\text{Enc} : \mathcal{X} \to \mathcal{V}^*$ mapping a structure $\mathbf{x}$ to a finite token sequence $\mathbf{q} = \text{Enc}(\mathbf{x})$, $\text{Dec} : \text{Im}(\text{Enc}) \to \mathcal{X}$ mapping $\tilde{\mathbf{x}} = \text{Dec}(\text{Enc}(\mathbf{x}))$. For dataset $\mathcal{D} \subset \mathcal{X}$ and distortion $d : \mathcal{X} \times \mathcal{X} \to [0, \infty)$ (e.g., Kabsch-aligned RMSD per residue), define:

$$\Delta(\mathsf{T}; \mathcal{D}) = \frac{1}{|\mathcal{D}|} \sum_{\mathbf{x} \in \mathcal{D}} d(\mathbf{x}, \text{Dec}(\text{Enc}(\mathbf{x}))), \qquad \text{BPR}(\mathsf{T}; \mathcal{D}) = \frac{\text{L}(\mathsf{T}) + \sum_{\mathbf{x} \in \mathcal{D}} \text{L}(\text{Enc}(\mathbf{x}))}{\sum_{\mathbf{x} \in \mathcal{D}} N(\mathbf{x})} \text{ bits/res}$$

where $\text{L}(\mathsf{T}) \geq 0$ is the description length of $(\mathcal{V}, \text{Enc}, \text{Dec})$ and $N(\mathbf{x})$ is the residue count; under a uniform per-token code, $\text{L}(\text{Enc}(\mathbf{x})) = |\text{Enc}(\mathbf{x})| \log_2 |\mathcal{V}|$. We setup the following principles for an ideal tokenizer $\widehat{\mathsf{T}}$ and empirically explore the degree GEOBPE satisfies them.

**Principle 1: Pareto-optimal on $\mathcal{D}$.** $\widehat{\mathsf{T}}$ is Pareto-optimal on $\mathcal{D}$ iff no $\mathsf{T}'$ satisfies $\text{BPR}(\mathsf{T}'; \mathcal{D}) \leq \text{BPR}(\widehat{\mathsf{T}}; \mathcal{D})$ and $\Delta(\mathsf{T}'; \mathcal{D}) \leq \Delta(\widehat{\mathsf{T}}; \mathcal{D})$, with at least one strict. We empirically explore this principle by evaluating Pareto-efficiency among leading PSTs and codebook configurations in Fig. 4.

**Principle 2: Out-of-distribution (OOD) generalization.** $\widehat{\mathsf{T}}$ generalizes OOD if, on unseen test set $\mathcal{D}_{\text{test}} \subset \mathcal{X}, \Delta(\mathsf{T}; \mathcal{D}_{\text{test}}) \approx \Delta(\mathsf{T}; \mathcal{D})$. We depict generalization gaps of leading PSTs in Fig. 4.

**Principle 3: Downstream transfer via codebook/vocabulary.** Let $\mathcal{V}$ be the vocabulary of $\mathsf{T}$ and let $N(\mathbf{x})$ be the residue count. Let $\Theta$ parameterize a pretrained feature extractor. The codebook/vocabulary $\mathcal{V}$ *induces* per-residue features $r_{\mathcal{V}}(\mathbf{x}) = \Psi_{\mathcal{V}}(F_\Theta(\mathbf{x})) \in (\mathbb{R}^d)^{N(\mathbf{x})}$. An ideal tokenizer of protein *structures* should go beyond pure compression; it should learn useful signals related to function. We loosely define the ability of a PST to transfer useful signals by test performance on a battery of downstream tasks when parameterizing samples $\mathbf{x}$ by the vocabulary $\mathcal{V}$ together with a feature extractor $\Theta$. We benchmark downstream transfer of GEOBPE against others in Table 1 (GEOBPE-TRANSFER).

## 4 EXPERIMENTS

We answer ten research questions (Q1-Q10) to benchmark the performance, efficiency, and application integration potential of GEOBPE against other popular tokenizers.

- **Tokenization Performance**: (Q1) How many bits are needed to store the tokenizer and tokenized inputs? (Q2) How faithful is the reconstruction? (Q3) How does performance generalize to unseen data? (Q4) How many samples are needed to train the tokenizer?
- **Token Efficiency**: (Q5) How frequent and balanced is vocabulary utilization? (Q6) Does small-scale language modeling generate better structures with GEOBPE or VQ-VAE tokens?
- **Downstream Transfer**: (Q7) How much transferrable signal does the tokenizer capture about the data? (Q8) How much does the vocabulary help on representation learning tasks?
- **Interpretability**: (Q9) How well do GEOBPE tokens agree with "ground-truth" domain annotations? (Q10) Can experts *understand* GEOBPE through real-world case studies?

**Datasets.** We follow the same dataset splits as in Yuan et al. (2025). Pretraining uses structures from the Protein Data Bank following OpenFold2's protocol and retained a non-redundant subset of $\approx$48K protein chains, which were split into training/validation sets, with CAMEO and CASP14 reserved as held-out test sets for evaluating OOD generalization and token efficiency. For downstream evaluation, we use 8 datasets, spanning residue-level classification (ligand binding, catalytic, conserved, repeat, and epitope), residue-level regression (structural flexibility prediction), and protein-level classification. Together these datasets probe functional relevance, structural variability, token distinctiveness, and efficiency across a wide range of proteins. For citations and details, see App. B.

**Baselines.** We compare with VQ-VAEs, the leading family of discrete PSTs (Hayes et al., 2025; Van Kempen et al., 2024; Lin et al., 2023a; Yuan et al., 2025). They consist of (1) a structure encoder maps structure $x$ into a continuous representation $z \in \mathbb{R}^{N \times D}$; (2) a vector quantization layer discretizes each $z_i$ by selecting $k_i = \arg\min_j d(z_i, q_j)$ from a learnable codebook $Q \in \mathbb{R}^{K \times D}$; and (3) a structure decoder reconstructs $\tilde{x} \approx \mathbf{x}$ from the discrete codes $\boldsymbol{q}_k = \{\boldsymbol{q}_{k_j}\}_{j=1}^L$. We also compare with Inverse Folding (IF) *continuous* PSTs, which skips the quantization step $\mathbf{z} \to \boldsymbol{q}_k$ and trained to recover the amino acid sequence from $\boldsymbol{z}$ (Dauparas et al., 2022; Yang et al., 2023).

**Downstream Transfer.** For VQ-VAEs, $\Theta$ and $\mathcal{V}$ are *jointly* learned, so we set $r_{\mathcal{V}}^{\text{VQ-VAE}}(\mathbf{x}) := r^{\text{VQ-VAE}}(\mathbf{x}) \leftarrow \text{Enc}(\mathbf{x})$. For GEOBPE-TRANSFER, we use $\Theta \leftarrow \text{ESM3}$ to demonstrate how $\mathcal{V}^{\text{GEOBPE}}$ can transfer useful signals from $\mathcal{F}_{\Theta}(\mathbf{x})$ to $r_{\mathcal{V}}(\mathcal{F}_{\Theta}(\mathbf{x}))$.

**Performance Metrics.** *Compression* measures Bits-Per-Residue (BPR), as defined in Sec. 3.3. *Distortion* ($\Delta$) use standard RMSD and LDDT. *Token Efficiency* uses Codebook Utility Rate (UR), Perplexity (details in App. D) and *Small Structure Language Model Evaluation (SSLM-Eval)* (details in App. E). SSLM-Eval compares tokenizers (GEOBPE vs VQ-VAEs) via integration with a small $\sim$7.3M Transformer architecture after respectively tokenizing the pretraining data splits (Algo. 2). Under the same data, model, training and compute resources, the respective models generate new sentences, detokenizes them into structures, and we compare *relative* generation metrics (Algo. 3, 4, 5). *Downstream Transfer* covers 12 tasks (24 test splits) using AUROC (%) for functional site prediction, Spearman's $\rho$ (%) for flexibility prediction, and Macro F1 (%) for fold classification. *Expert Agreement* measures Domain & Segment Recall/Precision/F1/IOU (details in App. G).

**Computational Complexity / Implementation Details.** We analyze the theoretical complexity of GEOBPE in App. J and justify the steps we took towards efficient implementation and use.

## 5 RESULTS

**Tokenizer Performance.** We find GEOBPE and ProToken form the Pareto front under both $\Delta \in \{\text{RMSD}, \text{LDDT}\}$. GEOBPE achieves $0.271 - 0.358\text{x}$ and $0.016 - 0.021\text{x}$ the BPR of ProToken and ESM3, dropping LDDT by only $18 - 22\%$ and $22 - 25\%$, which are impressive feats considering GEOBPE's training data was only $\approx 7\%$ and $0.02\%$ the size. We also observe GEOBPE's strong OOD generalization, with test/train RMSD peaking at $1.16$ (CAMEO) and $1.28$ (CASP), showing negligible degradation reconstructing unseen data; VQ-VAE/AminoASeed, using identical data splits, show degradation as high as $6.4\text{x}$ test RMSD. Crucially, as the GEOBPE codebook grows, the variants trace a near-linear path along the Pareto front toward ProToken, elastically trading off BPR for lower distortion, a feature other tokenizers do not have (as codebook dimensions are fixed).

**Token Efficiency.** We report UR & Perplexity averaged over held-out test sets to gauge codebook/vocabulary usage on unseen data, the setting where the tokenizer is deployed. In Table 10, we see all methods except VQ-VAE and ESM3 achieve an average UR of $> 40\%$; all except VQ-VAE achieve $0.2$ average Perplexity. An ideal tokenizer avoids codebook collapse, but exactly uniform token usage may not be desirable. We introduce SSLM-Eval to stress test whether codebook efficiency actually leads to generative efficiency. SSLM-Eval is a holistic way to compare

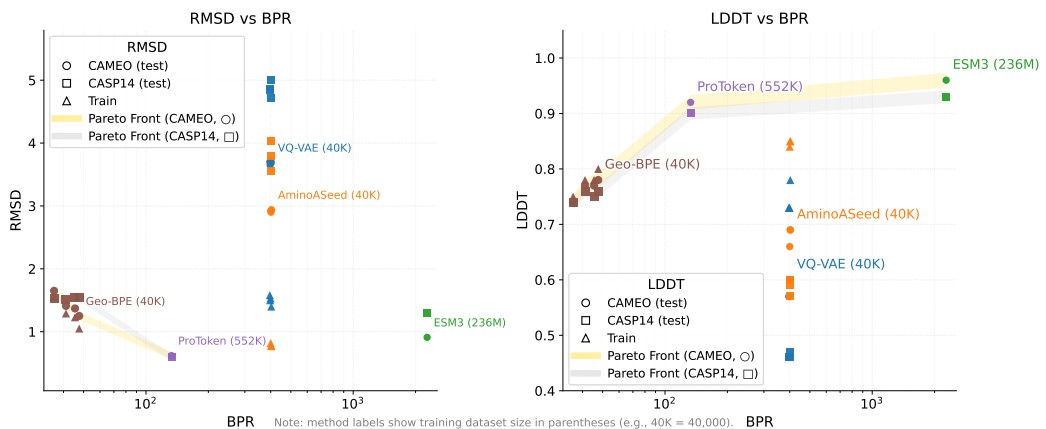

**Figure 4:** Plots of $(\mathrm{BPR}(\mathsf{T}; \mathcal{D}), \Delta(\mathsf{T}; \mathcal{D}_{\text{test}}))$ across tokenizers for $\Delta \in \{\mathrm{RMSD, LDDT}\}$. We vary $|\mathcal{V}| \in \{128, 256, 512, 1024\}$ for VQ-VAE/AminoASeed and $|\mathcal{V}| \in \{600, 2500, 6000, 21000\}$ for GEOBPE to sample multiple points; we observe GEOBPE sweeps a smooth tradeoff curve. Hyperparameters in App. K.

tokenizers using both encoder token efficiency *and* decoder's generative efficiency. In Table 11, we find GEOBPE-TRANSFER is capable of generating 99% unique and designable backbones, achieving up to 49% higher scTM and maintaining higher diversity than both VQ-VAE methods using the same data splits. We visualize some realistic, novel backbones GEOBPE-TRANSFER generated in App. E.4. Interestingly, the "less-efficient" VQ-VAE generated 58% more diverse backbones, demonstrating uniform token usage can be counterproductive to language modeling.

**Downstream Task Transfer.** In Table 1, we see GEOBPE-induced features rank first, on average, across both function and structure property prediction tasks. The relative performance gaps 15.44% and 43.28% quantify the add-on benefits of GEOBPE-induced features. GEOBPE-induced features reverse the trend that discrete PSTs produce less informative representations for downstream tasks (due to quantization-related issues (Yuan et al., 2025)), highlighting that *hierarchical structure* from discrete vocabularies raises the ceiling on downstream transfer.

**Further Ablations.** We include a comprehensive series of ablation studies in App. A demonstrating GEOBPE's *data-efficiency*, GEOBPE-TRANSFER's *task-agnosticism*, GEOBPE tokenizer's *scalability*, GEOBPE tokens' *adaptive resolution* over iterations, and *performance vs runtime tradeoffs* in components (1), (3) & (4). Key findings include: (i) GEOBPE shows *better* OOD generalization when fitted on 1% training data; (ii) GEOBPE-TRANSFER predictions are no worse when GEOBPE was fitted with (a) 1% of the pretraining PDBs, (b) the downstream task-specific PDBs; (iii) GEOBPE performance gains diminish beyond $M_{\max} = 5000$ randomly sampled motif occurrences used to extract prototypes, taming a complexity term that depends on $M_{\max}$.

## 6 DISCUSSION

**Case Study: Agreement with PFAM Annotations.** We ran CATH Functional Families (FunFams) (Das et al., 2015b) to obtain domain boundaries and compared them against GEOBPE-derived motifs. Because sequence conservation is linked to structural preservation, we expect overlap between predicted motifs and functional domains. In Table 2, GEOBPE achieves 99.97% domain recall with mean F1 = 0.996 and IOU = 0.992, showing near-perfect agreement across 10 datasets. *The agreement is not only geometric but also functional:* GEOBPE tokens frequently coincide with boundaries of ligand-binding grooves, transmembrane cavities, and scaffolding helices, capturing motifs that underlie molecular recognition and catalysis. This suggests GEOBPE does more than segment folds consistently: it surfaces interpretable structural primitives that map onto biochemical roles, offering a functional vocabulary absent in prior PSTs. Details are in App. G

**Case Study: Human Expert Analysis of Interpretability.** We conducted three expert evaluations of GEOBPE-derived hierarchies (App. H). Across proteins, the discovered motifs align with functionally meaningful substructures, including regions mediating ligand binding, molecular recognition, and structural gating. In the SLC25A20 transporter (Fig. 9), GEOBPE isolates a transmembrane binding cavity formed by helices and polar residues. In the 14-3-3:Tau complex (Fig. 10), it identifies a canonical phospho-binding groove stabilized by charged side chains. Recurrent local

**Table 1:** Downstream transfer performance benchmark. We underline and **bold** the best continuous and discrete PSTs, respectively; ▬ indicates the best method across both. The relative performance v.s. ESM3 for GEOBPE-TRANSFER is included. Omitted rows in Table 6; GEOBPE hyperparameters are in App. K.

| Task | Split | Continuous PST | | Discrete PST | | | | | |
|------|-------|----------------|----|-----------|-----------|------|----------|-----------|-----------------------------|
| | | ProteinMPNN | MIF | FoldSeek | ProTokens | ESM3 | VanillaVQ | AminoAseed | GEOBPE-TRANSFER (v.s. ESM3) |
| **Functional Site Prediction (AUROC%)** | | | | | | | | | |
| BindInt | Fold | 51.83 | 50.38 | 53.18 | 44.66 | 44.30 | 47.25 | 47.11 | **59.19 (+33.61%)** |
| | SupFam | 94.00 | 94.56 | 46.26 | 86.05 | 90.77 | 86.71 | 90.53 | **91.31 (+0.59%)** |
| BindBio | Fold | 78.42 | 85.79 | 32.37 | 58.47 | 62.84 | 62.02 | 65.73 | **94.94 (+51.08%)** |
| | SupFam | 81.00 | 87.27 | 52.44 | 60.47 | 65.22 | 62.92 | 68.30 | **95.94 (+47.10%)** |
| BindShake | Org | 75.52 | 79.90 | 53.43 | 59.10 | 66.10 | 67.04 | 69.61 | **87.73 (+32.72%)** |
| CatInt | Fold | 61.05 | 59.62 | 53.43 | 58.16 | 61.09 | 58.89 | 62.19 | **66.21 (+8.38%)** |
| | SupFam | 93.40 | 96.49 | 51.41 | 83.85 | 89.82 | 85.00 | **91.91** | 88.65 (-1.30%) |
| CatBio | Fold | 82.49 | 85.85 | 56.33 | 67.68 | 65.33 | 67.58 | 65.95 | **95.01 (+45.43%)** |
| | SupFam | 93.19 | 96.97 | 53.78 | 64.05 | 74.65 | 70.92 | 87.59 | **95.90 +28.47%** |
| Con | Fold | 57.18 | 58.43 | 49.20 | 57.20 | 55.22 | 56.98 | 57.23 | **71.96 (+30.32%)** |
| | SupFam | 84.68 | 92.66 | 51.31 | 70.64 | 80.53 | 74.60 | **86.60** | 84.84 (+5.35%) |
| | | | | | | ...2 tasks omitted (Rep, Ept)... | | | |
| **Average AUROC%** | | 75.92 | 79.82 | 51.90 | 65.37 | 69.24 | 68.30 | 72.43 | **80.20 (+18.13%)** |
| **Physicochemical Property Prediction (Spearman's $\rho$%)** | | | | | | | | | |
| FlexRMSF | Fold | 62.37 | 59.60 | 15.35 | 13.81 | 44.53 | 44.22 | **44.63** | 40.89 (-8.17%) |
| | SupFam | 59.24 | 56.80 | 11.99 | 7.62 | 39.08 | 38.98 | 40.99 | **47.17 (20.70%)** |
| | | | | | | ...2 tasks omitted (FlexBFactor, FlexNEQ)... | | | |
| **Average $\rho$%** | | 54.41 | 52.73 | 7.80 | 9.84 | 37.35 | 33.49 | 38.08 | **45.26 (+21.18%)** |
| **Structure Property Prediction (Macro F1%)** | | | | | | | | | |
| Homo | Fold | 25.66 | 22.56 | 11.57 | 5.84 | **30.02** | 18.17 | 29.87 | 23.60 (-21.39%) |
| | SupFam | 30.83 | 33.86 | 4.67 | 6.17 | 24.89 | 22.10 | 38.38 | **47.28 (+89.96%)** |
| | Fam | 63.33 | 74.22 | 15.34 | 18.33 | 54.42 | 47.18 | 69.78 | **85.75 (+57.47%)** |
| **Average Macro F1%** | | 39.94 | 43.55 | 10.51 | 10.11 | 36.44 | 29.15 | 46.01 | **52.21 (+43.28%)** |

**Table 2:** We annotate 100 PDBs from each dataset and report % of 1,000 random equal-length segmentations that GEOBPE matches or outscores. Omitted columns are in Table 7. Secondary structure analysis in Table 15.

| | | BindInt | BindBio | BindShake | CatInt | CatBio | Con | | Average |
|--------|----------------|---------------|---------------|---------------|--------------|--------------|-------------|----------------|----------------|
| Domain | Mean Recall | 99.95 (98.35) | 100 (100.0) | 100 (100.0) | 100 (100.0) | 99.99 (93.55) | 99.95 (98.0) | | 99.97 (97.97) |
| | Mean Precision | 98.9 (53.87) | 99.62 (71.92) | 99.76 (68.24) | 99.28 (50.49) | 99.33 (42.78) | 99.19 (63.89) | | 99.25 (54.59) |
| | Mean F1 | 99.42 (68.48) | 99.81 (83.63) | 99.88 (76.49) | 99.64 (62.56) | 99.66 (61.04) | 99.57 (87.03) | *4 columns omitted* | 99.61 (76.82) |
| | Mean IOU | 98.86 (86.32) | 99.62 (83.63) | 99.76 (76.54) | 99.28 (62.44) | 99.32 (60.94) | 99.14 (86.97) | ... 4 columns ... | 99.22 (76.75) |
| Segment | Mean Recall | 100 (100.0) | 100 (100.0) | 100 (100.0) | 100 (100.0) | 100 (100.0) | 100 (100.0) | | 100.00 (100.00) |
| | Mean Precision | 97.16 (72.04) | 81.84 (61.82) | 97.64 (68.33) | 90.4 (63.09) | 98.87 (74.76) | 98.92 (92.0) | | 95.11 (65.62) |
| | Mean F1 | 98.54 (72.04) | 89.05 (61.82) | 98.8 (68.33) | 94.04 (63.09) | 99.43 (74.76) | 99.45 (92.0) | | 97.23 (65.62) |

motifs (aromatic cages, polar bridges, helix-helix clamps) are combined into higher-order scaffolds that mirror established biochemical organization. *These hierarchies capture geometric regularities and also modular design principles conserved across folds and families.* Even in compact domains, such as nucleotide-recognition modules, GEOBPE motifs reveal the coupling between geometric curvature and chemical specificity, meaning that GEOBPE surfaces reusable motifs that are both interpretable and evolutionarily grounded.

**Limitations.** GEOBPE currently does not incorporate sequence or side chains, but *can* via direct extensions, e.g. taking the Cartesian product of the current vocabulary with amino acid types and augmenting the backbone formulation with type-dependent $\chi$-angle spans. The present integration with small-scale Transformers is set up to compare tokenizers' compability with language modeling on structure tokens; only *relative* backbone design metrics are relevant. Generative performance depends on model capacity and data scale, which are *orthogonal* to the tokenizer. In the separate SSLM scalability study, we see *steep gains in generative performance* when both LM parameter count and pretraining data increase ten-fold, conforming with scaling law expectations. The improved numbers are preliminary evidence of GEOBPE's promise as a tokenizer for lage-scale PLMs, but are not competitive with state-of-the-art backbone design models.

## 7 CONCLUSION

We present GEOBPE, a principled geometry-grounded analog of BPE for protein folds. GEOBPE (a) captures natural conformational variability in protein backbones, (b) constructs a hierarchical vocabulary of structural motifs, and (c) produces hierarchical views of folds for downstream representation learning. Its hierarchies reveal conserved modular design principles that connect structure to function. Empirically, GEOBPE advances the state of the art in tokenizer performance, out-of-distribution generalization, token and generative efficiency, downstream transfer, and interpretability. These results establish GEOBPE as a foundation for structure-native protein language models.

REPRODUCIBILITY STATEMENT

Code for GEOBPE and steps to reproduce all experiments in the paper are available at https://github.com/shiningsunnyday/PT-BPE/. We include detailed descriptions for understanding our method in the main text, with mathematical descriptions and pseudocodes in the Appendix. In App. K, we list the key hyperparameters, their effects on algorithm behavior, the default values used in our experiments, and any deviations from the default values used to obtain the results reported in the main text. In App. J, we analyze the computational complexity of our method and describe practical implementation choices used to make the method efficient in practice.

ACKNOWLEDGMENTS

M.S. and M.Z. gratefully acknowledge partial support by NSF CAREER Award 2339524, ARPA-H Biomedical Data Fabric (BDF) Toolbox Program, Amazon Faculty Research, Google Research Scholar Program, AstraZeneca Research, GlaxoSmithKline Award, Roche Alliance with Distinguished Scientists (ROADS) Program, Sanofi iDEA-iTECH Award, Boehringer Ingelheim Award, Merck Award, Optum AI Research Collaboration Award, Pfizer Research, Gates Foundation (INV-079038), Chan Zuckerberg Initiative, Collaborative Center for XDP at Massachusetts General Hospital, John and Virginia Kaneb Fellowship at Harvard Medical School, Biswas Computational Biology Initiative in partnership with the Milken Institute, Harvard Medical School Dean's Innovation Fund for the Use of Artificial Intelligence, and the Kempner Institute for the Study of Natural and Artificial Intelligence at Harvard University. Any findings, conclusions or recommendations expressed in this material are those of the authors and do not necessarily reflect the views of the funders.

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

# A    ABLATION STUDIES

**GEOBPE is task-agnostic, and using task-specific data does not increase downstream performance of GEOBPE-TRANSFER.** For each task $i$, let $\mathsf{T}_i^{\text{task}}$ be a tokenizer fitted using only $\mathcal{D}_i^{\text{train}}$ (with its own vocabulary $\mathcal{V}_i$ but the same feature extractor $F_\Theta$), and define $r_{\mathcal{V}_i}(\mathbf{x}) = \Psi_{\mathcal{V}_i}(F_\Theta(\mathbf{x}))$. We follow the same downstream transfer evaluation. We find an interesting result in Table 3, where directly training on the task-specific dataset does not meaningful change downstream prediction results. A closer look reveals the underlying reason is because the individual tokens do not differ significantly; motifs added to $\mathcal{V}$, in order, are similar across both GEOBPE-TRANSFER and GEOBPE-TRANSFER (task-specific). We can interpret this both positively and negatively. GEOBPE is insensitive to task-specific data and learns the "language" of protein folds consistently. This may be desirable for reusability of a tokenizer, as one does not need to retrain it for different data distributions, as all protein folds obey the same universal principles (Petsko & Ringe, 2004). At the same time, this upper bound tests whether the tokenizer can tailor its vocabulary to individual datasets for potentially higher scores, indicating GEOBPE by itself may lack the parameter capacity to overfit to individual tasks.

**Table 3:** GEOBPE-TRANSFER (1%) runs Algo. 1 with 1% of the pretrain training set, then uses the output vocabulary to induce features; GEOBPE-TRANSFER (task-specific) does not use pretraining data; instead it runs Algo. 1 with downstream data to learn a vocabulary. All use default value parameters in App. K.

| Functional Site Prediction (AUROC%) | | | | | | | | | | | | | | | | |
| Model | BindInt (Fold) | BindInt (SupFam) | BindBio (Fold) | BindBio (SupFam) | BindShake (Org) | CatInt (Fold) | CatInt (SupFam) | CatBio (Fold) | CatBio (SupFam) | Con (Fold) | Con (SupFam) | Rep (Fold) | Rep (SupFam) | Ept (Fold) | Ept (SupFam) | Avg |
|---|---|---|---|---|---|---|---|---|---|---|---|---|---|---|---|---|
| GEOBPE-TRANSFER (1%) | 59.98 | 90.17 | 95.00 | 95.89 | 87.73 | 66.28 | 88.87 | 94.95 | 95.95 | 71.75 | 84.56 | 56.37 | 72.87 | 63.83 | 77.55 | 80.12 |
| GEOBPE-TRANSFER | 59.19 | 91.31 | 94.94 | 95.94 | 87.73 | 66.21 | 88.65 | 95.01 | 95.90 | 71.96 | 84.84 | 56.44 | 72.98 | 64.78 | 77.06 | 80.20 |
| GEOBPE-TRANSFER (task-specific) | 60.16 | 89.93 | 95.05 | 95.92 | 87.73 | 66.28 | 88.82 | 94.98 | 95.90 | 71.85 | 85.92 | 56.33 | 72.72 | 64.78 | 77.04 | 80.23 |

| Physicochemical Property Prediction (Spearman's $\rho$%) | | | | | | | | Structure Property Prediction (Macro F1%) | | | |
| Model | FlexRMSF (Fold) | FlexRMSF (SupFam) | FlexBFactor (Fold) | FlexBFactor (SupFam) | FlexNEQ (Fold) | FlexNEQ (SupFam) | Avg | Homo (Fold) | Homo (SupFam) | Homo (Fam) | Avg |
|---|---|---|---|---|---|---|---|---|---|---|---|
| GEOBPE-TRANSFER (1%) | 40.42 | 47.55 | 34.74 | 32.21 | 56.78 | 55.32 | 44.50 | 21.65 | 50.25 | 84.87 | 52.26 |
| GEOBPE-TRANSFER | 40.89 | 47.17 | 37.28 | 35.61 | 56.65 | 53.98 | 45.26 | 23.60 | 47.28 | 85.75 | 52.21 |
| GEOBPE-TRANSFER (task-specific) | 39.39 | 44.00 | 37.94 | 38.36 | 56.22 | 54.22 | 45.02 | 24.22 | 46.58 | 84.57 | 51.79 |

**GEOBPE-TRANSFER maintains comparable downstream transfer performance even when GEOBPE was fitted on 1% of pretraining data.** In Table 3, we see GEOBPE fitted on just 1% of the pretraining data is enough to transfer, on average, the same amount of performance downstream as GEOBPEfitted on the full dataset. There are no meaningful differences between GEOBPE-TRANSFER and GEOBPE-TRANSFER (1%), with GEOBPE-TRANSFER doing 1.7% better on physicochemical property prediction and GEOBPE-TRANSFER (1%) doing better 0.2% better on functionals ite prediction. These findings can be interpreted both positively and negatively for GEOBPElearned vocabularies: (1) they are *extremely* informative, learning useful signals to transfer downstream with as few as 300 PDB structures; (2) they *underfit* the data, with no noticeable improvements from using more data to learn the tokenizer. Taken together, these findings imply GEOBPE is a lightweight add-on on top of any pretrained features $\Theta$, but feeding more data to GEOBPE yields diminishing downstream returns quickly.

**GEOBPE-TRANSFER does not underfit the data for structure-related downstream tasks.** In Table 4, we see for physicochemical (residue-level regression) and fold-level tasks, the model indicates a clear propensity for more training data, with step-wise gains for every 20% of training data. $20\% \rightarrow 100\%$ training data sees performance lift significantly ($+18.89\%$ average $\rho\%$ and $+45.35\%$, respectively). For residue-level classification tasks, the lift is only marginal ($+1.28\%$). We hypothesize the cause is not limited capacity, but rather that the tasks are localized label predictions; a residue-level receptive field is sufficient when features are informative. This shows GeoBPE's data-efficiency at learning a vocabulary does not limit its capacity on downstream tasks that require multi-scale resolution (e.g. structural flexibility or fold-level classification), which existing fixed-size tokenizers cannot at both residue and structure-level. Thus, the data-efficiency strengths of GeoBPE *training* is orthogonal to downstream modeling. The structure-related tasks in Table 4 see large gains in performance with more training data, implying GEOBPE-TRANSFER scales to complex, hierarchical structural patterns that can only be learned from more data.

**GEOBPE is data-efficient OOD, but more training data can lower training distortion.** In Table 5, we see GEOBPE (1%) consistently achieves lower distortion ($\downarrow$ 7.7% RMSD averaged, $\uparrow$ 0.06 LDDT summed across all four runs and both test splits) than GEOBPE. This suggests a small, well-chosen set of structures is enough for GEOBPE to achieve superior reconstruction on *OOD* structures, and more training data can introduce noise and hinder generalization. However, GEOBPE (1%) obtains $\uparrow$ 5.1% averaged, $\downarrow$ 0.02 LDDT summed across all four runs on the respective training splits. A lower RMSD suggests GEOBPE better captures global structural fidelity;

**Table 4:** We use GEOBPE-TRANSFER from Table 1 (reported as 100%), then vary only the percent of downstream task training data available, fixing the same valid and test sets.

| Task | Split | GEOBPE-TRANSFER | | | | |
|---|---|---|---|---|---|---|
| | | 20% | 40% | 60% | 80% | 100% |
| **Functional Site Prediction (AUROC%)** | | | | | | |
| BindInt | Fold | 58.78 | 61.18 | 59.79 | 59.45 | 59.19 |
| | SupFam | 89.55 | 89.88 | 90.42 | 90.71 | 91.31 |
| BindBio | Fold | 94.86 | 94.82 | 94.96 | 94.97 | 94.94 |
| | SupFam | 95.69 | 95.79 | 95.88 | 95.97 | 95.94 |
| BindShake | Org | 87.40 | 87.59 | 87.65 | 87.64 | 87.73 |
| CatInt | Fold | 64.66 | 65.55 | 66.94 | 66.29 | 66.21 |
| | SupFam | 87.96 | 88.27 | 88.78 | 88.73 | 88.65 |
| CatBio | Fold | 94.95 | 94.92 | 94.94 | 94.96 | 95.01 |
| | SupFam | 95.67 | 95.81 | 95.96 | 95.97 | 95.90 |
| Con | Fold | 71.42 | 71.37 | 71.72 | 71.74 | 71.96 |
| | SupFam | 83.77 | 84.02 | 84.76 | 84.76 | 84.84 |
| Rep | Fold | 55.04 | 54.25 | 56.02 | 56.41 | 56.44 |
| | SupFam | 73.24 | 75.18 | 75.89 | 71.58 | 72.98 |
| Ept | Fold | 63.63 | 53.28 | 61.72 | 61.66 | 64.78 |
| | SupFam | 71.21 | 49.39 | 73.59 | 76.01 | 77.06 |
| **Average AUROC%** | | 79.19 | 77.42 | 79.93 | 79.79 | **80.20** |
| **Physicochemical Property Prediction (Spearman's $\rho$%)** | | | | | | |
| FlexRMSF | Fold | 41.49 | 43.33 | 38.48 | 39.02 | 40.89 |
| | SupFam | 34.74 | 45.41 | 44.61 | 47.06 | 47.17 |
| FlexBFactor | Fold | 23.90 | 27.86 | 33.83 | 34.82 | 37.28 |
| | SupFam | 23.80 | 25.46 | 37.70 | 36.49 | 35.61 |
| FlexNEQ | Fold | 54.40 | 56.07 | 55.98 | 57.53 | 56.65 |
| | SupFam | 51.12 | 53.77 | 52.52 | 54.96 | 53.98 |
| **Average $\rho$%** | | 38.24 | 41.98 | 43.85 | 44.98 | **45.26** |
| **Structure Property Prediction (Macro F1%)** | | | | | | |
| Homo | Fold | 14.35 | 20.55 | 23.74 | 24.25 | 23.60 |
| | SupFam | 27.63 | 35.11 | 43.09 | 43.98 | 47.28 |
| | Fam | 65.79 | 73.66 | 80.77 | 82.04 | 85.75 |
| **Average Macro F1%** | | 35.92 | 43.11 | 49.20 | 50.09 | **52.21** |

**Table 5:** We rerun the GEOBPE experiments used to trace out the Pareto Front in Fig. 4 by using 1% of pretraining data. We include raw numbers of Fig. 4 (bottom rows) for comparison. All other hyperparameter settings are kept the same (App. K).

| | Train | | Valid | | CAMEO | | CASP14 | |
|---|---|---|---|---|---|---|---|---|
| | RMSD | LDDT | RMSD | LDDT | RMSD | LDDT | RMSD | LDDT |
| GeoBPE (1%) ($|V| = 600$) | 1.72 | 0.74 | 1.63 | 0.73 | 1.66 | 0.73 | 1.53 | 0.72 |
| GeoBPE ($|V| = 600$) | 1.66 | 0.73 | 1.71 | 0.72 | 1.77 | 0.72 | 1.53 | 0.72 |
| GeoBPE (1%) ($|V| = 2278$) | 1.57 | 0.75 | 1.51 | 0.71 | 1.51 | 0.74 | 1.43 | 0.73 |
| GeoBPE ($|V| = 2500$) | 1.41 | 0.75 | 1.50 | 0.74 | 1.57 | 0.74 | 1.51 | 0.73 |
| GeoBPE (1%) ($|V| = 5278$) | 1.36 | 0.77 | 1.34 | 0.76 | 1.35 | 0.75 | 1.30 | 0.74 |
| GeoBPE ($|V| = 6000$) | 1.37 | 0.76 | 1.46 | 0.75 | 1.52 | 0.74 | 1.54 | 0.72 |
| GeoBPE (1%) ($|V| = 20278$) | 1.29 | 0.77 | 1.28 | 0.76 | 1.28 | 0.76 | 1.37 | 0.73 |
| GeoBPE ($|V| = 21000$) | 1.21 | 0.77 | 1.28 | 0.76 | 1.40 | 0.75 | 1.55 | 0.72 |

by constructing the vocabulary from the full pretraining dataset, it can choose more representative prototypes; hence, its vocabulary better preserves global fold. Meanwhile, a slightly lower LDDT indicates GEOBPE (1%) can capture a few local details in the 1% subset of structures better than

GEOBPE. This suggests GEOBPE (1%) is more sensitive to the individual local interactions of the small set of structures it fitted with; GEOBPE considers vastly more structures. In summary, GEOBPE is preferred for fold-preserving *compression* of whole datasets, but GEOBPE (1%) can be feasible if not superior when GEOBPE is primarily used to tokenize unseen data.

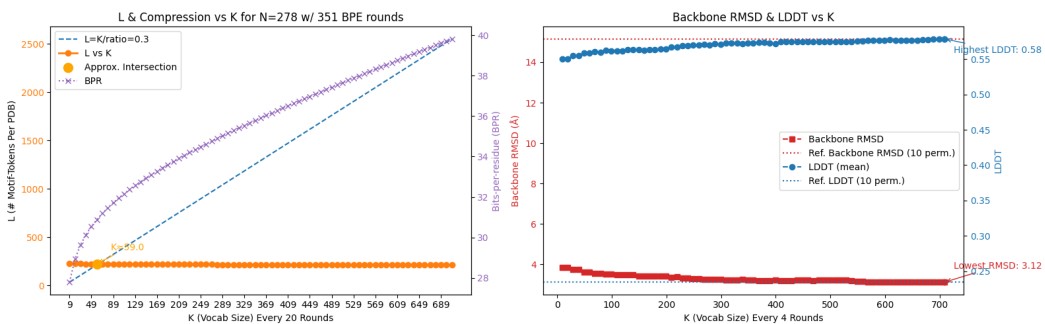

**Figure 5:** We plot the BPR (purple), length (orange), backbone distortion (RMSD, LDDT) as $|\mathcal{V}|$ across BPE steps. Ref. backbone RMSD/LDDT (dotted lines) uses random angle values for all internal angles, sampled from the empirical angle distribution.

**GEOBPE is multi-resolution, revealing finer details as more tokens are introduced.** In Fig. 5, we run a coarse-grained version of GEOBPE (small initial $|\mathcal{V}|$) to observe an interesting feature of GEOBPE's design. As newly introduced tokens *re*-quantize the occurrences from the original data (span gathering step in Alg. 9, tokenization can adaptively *increase* the resolution if the new prototypes better capture the modes of variability for those occurrences than their previous quantization. We expose this via hyperparameters bins & num_p (see App. K), which tradeoff the super-resolution effect against coarse-graining effect at different token sizes, offering fine-grained control.

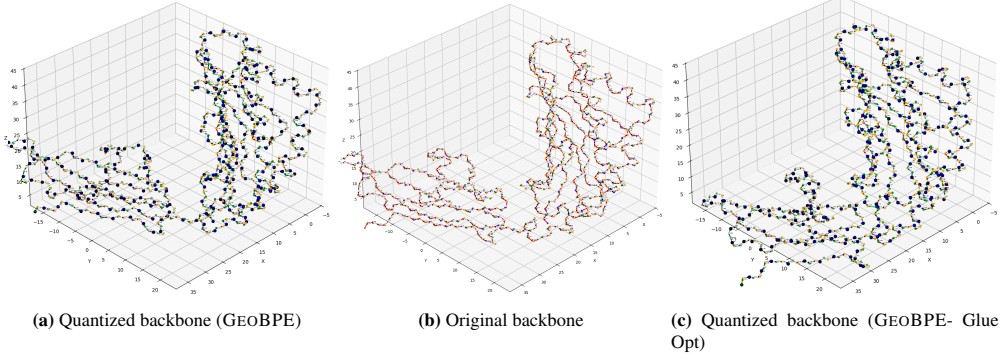

**(a)** Quantized backbone (GEOBPE)  **(b)** Original backbone  **(c)** Quantized backbone (GEOBPE- Glue Opt)

**Figure 6:** We ran an ablation for GEOBPE version with $|\mathcal{V}| = 600$, keeping all parameters the same but toggling whether glue opt is skipped in Alg. 18. We visualize the original (center), GEOBPE (left) and GEOBPE without glue opt (right) backbone states.

**Rigid body refinment as an essential step for preserving fold integrity.** If we omit the glue optimization from Algs 18 and 9 altogether, we see the effects in Fig. 6. For that experiment, we find avg. RMSD increase $1.66 \rightarrow 4.39$, and avg. LDDT drop $0.73 \rightarrow 0.69$ when glue opt is turned off. Rigid body refinement preserves the overall fold and modular architecture; turning it off causes individual domains to distort – the parallel strands drift apart – as well as the overall configuration to lose its integrity. Over the course of many time steps, global drift accumulate as local rounding occurs. Rigid body refinement is an indispensable subroutine for ensuring the overall quantization faithfully reproduces the fold integrity.

**Increasing $M_{\mathbf{max}}$ beyond a certain threshold does not yield additional distortion benefits.** We did a study comparing GEOBPE ($\mathcal{V} = 6000$, $M_{max} \leftarrow 5000$, full settings in App. K) with "higher-resolution" settings bins $\leftarrow \{1 : 5000\}$, $M_{max} \leftarrow 20000$. Interestingly, we found overall RMS-D/LDDT did *not* improve (1.40 vs 1.39, 0.76 vs 0.75, both in favor of the incumbent) despite

increased computational expenditure spent on Alg. 8. The most likely explanation is there is no marginal utility increasing $M_{\max}$ beyond 5000, and differences in distortion rates are likely due to the numerical stability of Alg. 12 more so than the hyperparameters.

## B  DATASET DETAILS

**Training.** For training GEOBPE, we started with the pretraining data splits released by Yuan et al. (2025), which follows the same criteria used to train the OpenFold2 model Ahdritz et al. (2024). For VanillaVQ and AminoASeed baselines, we use the same splits as Yuan et al. (2025) directly. For GEOBPE, we further filtered the data down to only ones with complete backbone information (e.g. backbone dihedrals are not NaN, each residue contains N, CA and C), resulting in 34818 structures. We further excluded structures shorter than 40 or longer than 512 residues, resulting in 33992 structures for training GeoBPE and 3810 for validation (only used for E).

**Held-out testing.** We use CAMEO and CASP14 test sets for evaluating the generalization of tokenizers (Robin et al., 2021; Kryshtafovych et al., 2021). For CASP14, we follow Yuan et al. (2025) and select only proteins released after the pretraining data cutoff date.

**Downstream Tasks.** Our 8 downstream tasks cover a breadth of structure and function-related predictions. They are divided into 3 categories and are assembled from 6 sources: InterPro (BindInt, Con, Rep) (Blum et al., 2025), BioLIP2 (BindBio, CatBio) (Zhang et al., 2024b), ProteinShake (BindShake) (Kucera et al., 2023), ProteinGLUE (Ept) (Capel et al., 2022), TAPE (Homo) (Rao et al., 2019) and ATLAS (FlexRMSF, FlexBFactor, FlexNEQ) (Vander Meersche et al., 2024).

1. **Functional site prediction**: Binding site prediction (BindInt), catalytic site (CatInt), conserved site prediction (Con), repeat motif prediction (Rep), epitope region prediction (Ept)

2. **Physicochemical property prediction**: Structural flexibility prediction, measured using metric RMSF (FlexRMSF), B-factor (FlexBFactor) and Neq (FlexNEQ)

3. **Structure classification** (protein-level): Remote homology detection (Homo)

**Functional site prediction** tasks predict whether each residue is in a site of functional importance (binding, catalytic activity or antibody recognition) or part of an evolutionary motif (conserved site or part of a repeated motif). PSTs which learn semantically meaningful signals like motif boundaries are expected to perform well on these tasks.

**Physicochemical property prediction** tasks predict the flexibility of each residue as a continuous value. Higher flexibility can be a clue that the residue may be more amenable to functional activity. PSTs that capture a fine-grained view of the localized protein dynamics are expected to predict residue-level flexibility well.

**Remote homology detection** is a multi-class fold classification problem. Proteins which belong to the same fold class can be distantly related or share similar functions on the whole. Therefore, PSTs that capture the overall fold-level geometry are expected to do well on this task.

For more dataset statistics and preparation details, see Yuan et al. (2025).

## C  ADDITIONAL RESULTS

**Table 6:** Additional downstream transfer performance tasks. Setup follows Table 1.

| Task | Split | Continuous PST | | Discrete PST | | | | | |
|------|-------|----------------|-----|----------|-----------|------|-----------|------------|---------------------------------|
| | | ProteinMPNN | MIF | FoldSeek | ProTokens | ESM3 | VanillaVQ | AminoAseed | GEOBPE-TRANSFER (v.s. ESM3) |
| **Functional Site Prediction (AUROC%)** | | | | | | | | | |
| Rep | Fold | 77.63 | 74.53 | 47.71 | 53.20 | 74.70 | **75.99** | 74.97 | 56.44 (-24.44%) |
| | SupFam | 80.71 | 83.11 | 52.54 | 77.25 | 82.36 | 82.09 | 84.57 | 72.98 (-11.39%) |
| Ept | Fold | 62.84 | 68.78 | 54.56 | 52.49 | 63.69 | 59.28 | 62.16 | **64.78 (+1.71)%** |
| | SupFam | 64.84 | 82.98 | 50.53 | 61.92 | 61.97 | 67.24 | 72.02 | **77.06 (+24.35)%** |
| **Physicochemical Property Prediction (Spearman's $\rho$%)** | | | | | | | | | |
| FlexBFactor | Fold | 31.88 | 34.60 | 4.17 | 6.67 | 23.60 | 22.32 | 21.30 | **37.28 (+57.97%)** |
| | SupFam | 34.56 | 35.23 | 6.99 | 5.47 | 25.80 | 23.73 | 21.76 | **35.61 (+38.02%)** |
| FlexNEQ | Fold | 69.69 | 65.32 | 5.71 | 12.98 | 45.05 | 35.95 | 49.64 | **56.65 (+25.75%)** |
| | SupFam | 68.69 | 64.82 | 2.66 | 10.51 | 35.45 | 35.61 | 50.15 | **53.98 (+52.27%)** |

**Table 7:** Additional expert agreement results. Setup follows Table 2.

|  |  | Rep | Ept | Atlas | Homo |
|---|---|---|---|---|---|
| Domain | Mean Recall | 99.93 (99.34) | 100 (100.0) | 99.93 (90.47) | 99.98 (100.0) |
|  | Mean Precision | 99.2 (43.59) | 99.75 (77.89) | 98.44 (31.29) | 99 (41.92) |
|  | Mean F1 | 99.56 (81.86) | 99.87 (82.68) | 99.17 (67.37) | 99.48 (79.07) |
|  | Mean IOU | 99.12 (82.18) | 99.75 (82.68) | 98.37 (67.07) | 98.98 (78.78) |
| Segment | Mean Recall | 100 (100.0) | 100 (100.0) | 100 (100.0) | 100 (100.0) |
|  | Mean Precision | 98.76 (61.93) | 95.52 (60.8) | 96.09 (47.29) | 95.91 (54.13) |
|  | Mean F1 | 99.38 (61.93) | 97.68 (60.8) | 98 (47.29) | 97.91 (54.13) |

**Main Text Tables.** Table 6 contains additional tasks Rep, Ept, FlexRMSF, and FlexBFactor. Table 7 contains additional task data Repeat, Ept, Atlas, Homo. Task abbrevations are defined in App. B.

**Table 8:** Secondary structure element (SSE) agreement results. Setup is the same as in Sec. G, but stratified over 8 basic secondary structure building blocks. Annotations are obtained from DSSP.

| Metric / SSE | BindInt | BindBio | BindShake | CatInt | CatBio | Con | Repeat | Ept | Atlas | Homo |
|---|---|---|---|---|---|---|---|---|---|---|
| Mean recall H | 29.11 (98.22) | 33.13 (97.42) | 30.76 (96.39) | 33.97 (98.1) | 32.93 (96.48) | 29.11 (98.22) | 33.13 (97.42) | 30.76 (96.39) | 33.97 (98.1) | 32.93 (96.48) |
| Mean recall G | 19.15 (99.4) | 31.13 (98.61) | 30.87 (98.55) | 34.66 (98.75) | 32.29 (98.71) | 19.15 (99.4) | 31.13 (98.61) | 30.87 (98.55) | 34.66 (98.75) | 32.29 (98.71) |
| Mean recall I | 10.42 (100.0) | 30.75 (100.0) | 17.59 (100.0) | 10.85 (99.26) | 10.42 (100.0) | 10.42 (100.0) | 30.75 (100.0) | 17.59 (100.0) | 10.85 (99.26) | 30.85 (99.68) |
| Mean recall E | 41.24 (99.56) | 34.75 (98.46) | 36.17 (97.49) | 30.82 (97.51) | 34.93 (97.99) | 41.24 (99.56) | 34.75 (98.46) | 36.17 (97.49) | 30.82 (97.51) | 34.93 (97.99) |
| Mean recall B | 38.13 (100.0) | 32.24 (100.0) | 38.72 (100.0) | 40.22 (100.0) | 31.67 (100.0) | 38.13 (100.0) | 32.24 (100.0) | 38.72 (100.0) | 40.22 (100.0) | 31.67 (100.0) |
| Mean recall T | 32.78 (98.74) | 33.08 (97.4) | 32.26 (96.41) | 33.75 (96.67) | 32.91 (97.08) | 32.78 (98.74) | 33.08 (97.4) | 32.26 (96.41) | 33.75 (96.67) | 32.91 (97.08) |
| Mean recall S | 33.92 (99.32) | 33.29 (98.0) | 32.91 (97.72) | 36.01 (97.4) | 33.12 (97.96) | 33.92 (99.32) | 33.29 (98.0) | 32.91 (97.72) | 36.01 (97.4) | 33.12 (97.96) |
| Mean recall - | 32.7 (97.81) | 32.76 (97.17) | 33.02 (96.16) | 34.71 (96.86) | 32.65 (96.03) | 32.7 (97.81) | 32.76 (97.17) | 33.02 (96.16) | 34.71 (96.86) | 32.65 (96.03) |
| Mean precision H | 31.34 (54.95) | 34.06 (48.88) | 31.33 (51.22) | 35.54 (48.86) | 33.32 (46.04) | 31.34 (54.95) | 34.06 (48.88) | 31.33 (51.22) | 35.54 (48.86) | 33.32 (46.04) |
| Mean precision G | 17.93 (79.42) | 29.28 (65.93) | 28.95 (62.87) | 32.3 (65.07) | 30.32 (64.77) | 17.93 (79.42) | 29.28 (65.93) | 28.95 (62.87) | 32.3 (65.07) | 30.32 (64.77) |
| Mean precision I | 10.09 (94.67) | 29.93 (87.7) | 16.55 (87.53) | 10.37 (93.27) | 30.16 (88.4) | 10.09 (94.67) | 29.93 (87.7) | 16.55 (87.53) | 10.37 (93.27) | 30.16 (88.4) |
| Mean precision E | 39.54 (47.86) | 33.59 (46.21) | 34.84 (45.6) | 33.53 (45.56) | 33.53 (45.56) | 39.54 (47.86) | 33.59 (46.21) | 34.84 (45.6) | 29.58 (53.03) | 33.53 (45.56) |
| Mean precision B | 26.8 (74.03) | 21.86 (71.01) | 26.66 (68.64) | 26.21 (61.46) | 21.2 (69.63) | 26.8 (74.03) | 21.86 (71.01) | 26.66 (68.64) | 26.21 (61.46) | 21.2 (69.63) |
| Mean precision T | 28.59 (53.66) | 28.55 (50.1) | 27.98 (50.94) | 29.18 (52.52) | 28.13 (48.36) | 28.59 (53.66) | 28.55 (50.1) | 27.98 (50.94) | 29.18 (52.52) | 28.13 (48.36) |
| Mean precision S | 26.78 (56.59) | 25.28 (50.82) | 24.66 (49.22) | 26.52 (45.95) | 25.0 (50.96) | 26.78 (56.59) | 25.28 (50.82) | 24.66 (49.22) | 26.52 (45.95) | 25.0 (50.96) |
| Mean precision - | 27.07 (54.17) | 26.5 (48.75) | 26.32 (48.44) | 28.07 (51.65) | 25.99 (48.15) | 27.07 (54.17) | 26.5 (48.75) | 26.32 (48.44) | 28.07 (51.65) | 25.99 (48.15) |
| Mean f1 H | 29.42 (61.9) | 33.12 (60.16) | 30.7 (62.58) | 34.0 (59.45) | 32.79 (58.18) | 29.42 (61.9) | 33.12 (60.16) | 30.7 (62.58) | 34.0 (59.45) | 32.79 (58.18) |
| Mean f1 G | 18.44 (83.1) | 30.03 (72.52) | 29.75 (70.88) | 33.34 (72.73) | 31.15 (72.15) | 18.44 (83.1) | 30.03 (72.52) | 29.75 (70.88) | 33.34 (72.73) | 31.15 (72.15) |
| Mean f1 I | 10.25 (95.59) | 30.24 (89.98) | 17.04 (89.31) | 10.59 (94.02) | 30.38 (90.53) | 10.25 (95.59) | 30.24 (89.98) | 17.04 (89.31) | 10.59 (94.02) | 30.38 (90.53) |
| Mean f1 E | 40.21 (59.65) | 33.88 (60.28) | 35.25 (59.17) | 29.98 (63.68) | 33.99 (60.01) | 40.21 (59.65) | 33.88 (60.28) | 35.25 (59.17) | 29.98 (63.68) | 33.99 (60.01) |
| Mean f1 B | 30.57 (75.28) | 25.3 (73.05) | 30.65 (71.04) | 30.86 (64.57) | 24.67 (71.8) | 30.57 (75.28) | 25.3 (73.05) | 30.65 (71.04) | 30.86 (64.57) | 24.67 (71.8) |
| Mean f1 T | 30.22 (62.79) | 30.27 (61.7) | 29.61 (63.41) | 30.93 (64.04) | 29.95 (61.29) | 30.22 (62.79) | 30.27 (61.7) | 29.61 (63.41) | 30.93 (64.04) | 29.95 (61.29) |
| Mean f1 S | 29.24 (61.77) | 28.04 (58.79) | 27.46 (57.52) | 29.76 (54.36) | 27.79 (59.56) | 29.24 (61.77) | 28.04 (58.79) | 27.46 (57.52) | 29.76 (54.36) | 27.79 (59.56) |
| Mean f1 - | 28.82 (61.51) | 28.59 (60.08) | 28.57 (59.89) | 30.35 (63.16) | 28.25 (60.75) | 28.82 (61.51) | 28.59 (60.08) | 28.57 (59.89) | 30.35 (63.16) | 28.25 (60.75) |
| Mean iou H | 28.4 (61.54) | 32.22 (59.93) | 29.94 (62.38) | 32.95 (59.09) | 31.96 (57.96) | 28.4 (61.54) | 32.22 (59.93) | 29.94 (62.38) | 32.95 (59.09) | 31.96 (57.96) |
| Mean iou G | 17.75 (83.07) | 28.87 (72.45) | 28.59 (70.78) | 31.99 (72.63) | 29.98 (72.11) | 17.75 (83.07) | 28.87 (72.45) | 28.59 (70.78) | 31.99 (72.63) | 29.98 (72.11) |
| Mean iou I | 10.09 (95.59) | 29.58 (89.98) | 16.52 (89.31) | 10.31 (94.02) | 29.67 (90.51) | 10.09 (95.59) | 29.58 (89.98) | 16.52 (89.31) | 10.31 (94.02) | 29.67 (90.51) |
| Mean iou E | 39.1 (59.46) | 32.76 (59.84) | 34.14 (58.81) | 29.02 (63.37) | 32.9 (59.53) | 39.1 (59.46) | 32.76 (59.84) | 34.14 (58.81) | 29.02 (63.37) | 32.9 (59.53) |
| Mean iou B | 26.79 (74.03) | 21.86 (71.14) | 26.66 (68.8) | 26.21 (61.53) | 21.2 (69.74) | 26.79 (74.03) | 21.86 (71.14) | 26.66 (68.8) | 26.21 (61.53) | 21.2 (69.74) |
| Mean iou T | 28.36 (61.64) | 28.26 (59.8) | 27.69 (61.71) | 28.9 (62.33) | 27.92 (59.16) | 28.36 (61.64) | 28.26 (59.8) | 27.69 (61.71) | 28.9 (62.33) | 27.92 (59.16) |
| Mean iou S | 26.63 (59.27) | 25.15 (53.97) | 24.47 (52.46) | 26.41 (48.85) | 24.9 (54.31) | 26.63 (59.27) | 25.15 (53.97) | 24.47 (52.46) | 26.41 (48.85) | 24.9 (54.31) |
| Mean iou - | 26.17 (57.07) | 25.93 (54.13) | 25.83 (53.73) | 27.63 (57.52) | 25.56 (53.91) | 26.17 (57.07) | 25.93 (54.13) | 25.83 (53.73) | 27.63 (57.52) | 25.56 (53.91) |
| Segment recall H | 29.35 (99.64) | 33.14 (99.88) | 30.94 (99.93) | 33.77 (99.65) | 33.07 (99.91) | 29.35 (99.64) | 33.14 (99.88) | 30.94 (99.93) | 33.77 (99.65) | 33.07 (99.91) |
| Segment recall G | 19.18 (100.0) | 31.2 (100.0) | 30.93 (100.0) | 34.89 (100.0) | 32.37 (99.85) | 19.18 (100.0) | 31.2 (100.0) | 30.93 (100.0) | 34.89 (100.0) | 32.37 (99.85) |
| Segment recall I | 10.42 (100.0) | 30.79 (100.0) | 17.63 (100.0) | 10.91 (100.0) | 30.94 (100.0) | 10.42 (100.0) | 30.79 (100.0) | 17.63 (100.0) | 10.91 (100.0) | 30.94 (100.0) |
| Segment recall E | 41.28 (100.0) | 34.8 (99.36) | 36.21 (99.8) | 35.0 (99.68) | 35.0 (99.68) | 41.28 (100.0) | 34.8 (99.36) | 36.21 (99.8) | 30.84 (100.0) | 35.0 (99.68) |
| Segment recall B | 38.01 (99.69) | 31.7 (98.92) | 38.05 (98.75) | 39.75 (99.02) | 31.2 (98.87) | 38.01 (99.69) | 31.7 (98.92) | 38.05 (98.75) | 39.75 (99.02) | 31.2 (98.87) |
| Segment recall T | 32.81 (99.4) | 33.09 (99.07) | 32.25 (98.3) | 33.74 (98.46) | 32.86 (98.61) | 32.81 (99.4) | 33.09 (99.07) | 32.25 (98.3) | 33.74 (98.46) | 32.86 (98.61) |
| Segment recall S | 33.73 (99.03) | 33.02 (98.7) | 32.5 (98.17) | 35.39 (97.75) | 32.77 (98.16) | 33.73 (99.03) | 33.02 (98.7) | 32.5 (98.17) | 35.39 (97.75) | 32.77 (98.16) |
| Segment recall - | 33.29 (98.57) | 32.94 (98.35) | 33.07 (98.3) | 34.87 (98.83) | 32.72 (98.52) | 33.29 (98.57) | 32.94 (98.35) | 33.07 (98.3) | 34.87 (98.83) | 32.72 (98.52) |
| Segment precision H | 39.24 (63.48) | 34.23 (61.03) | 34.32 (61.08) | 33.02 (59.47) | 39.24 (63.48) | 39.24 (63.48) | 34.23 (61.03) | 34.32 (61.08) | 29.43 (59.98) | 33.02 (59.47) |
| Segment precision G | 2.64 (82.35) | 4.22 (72.63) | 4.4 (71.91) | 4.56 (75.06) | 4.16 (70.14) | 2.64 (82.35) | 4.22 (72.63) | 4.4 (71.91) | 4.56 (75.06) | 4.16 (70.14) |
| Segment precision I | 1.96 (93.94) | 2.25 (83.44) | 1.2 (90.89) | 1.68 (82.77) | 1.68 (82.77) | 1.96 (93.94) | 2.25 (83.44) | 1.2 (90.89) | 0.68 (93.18) | 1.68 (82.77) |
| Segment precision E | 34.7 (62.4) | 23.29 (64.43) | 23.79 (64.45) | 17.3 (65.59) | 20.66 (63.88) | 34.7 (62.4) | 23.29 (64.43) | 23.79 (64.45) | 17.3 (65.59) | 20.66 (63.88) |
| Segment precision B | 1.28 (99.67) | 1.08 (98.39) | 1.18 (98.47) | 1.2 (98.86) | 0.96 (98.46) | 1.28 (99.67) | 1.08 (98.39) | 1.18 (98.47) | 1.2 (98.86) | 0.96 (98.46) |
| Segment precision T | 9.58 (70.95) | 9.36 (72.49) | 9.05 (71.02) | 9.59 (71.11) | 9.21 (72.8) | 9.58 (70.95) | 9.36 (72.49) | 9.05 (71.02) | 9.59 (71.11) | 9.21 (72.8) |
| Segment precision S | 8.12 (78.71) | 6.83 (83.39) | 6.37 (82.98) | 7.36 (83.94) | 6.53 (83.21) | 8.12 (78.71) | 6.83 (83.39) | 6.37 (82.98) | 7.36 (83.94) | 6.53 (83.21) |
| Segment precision - | 15.47 (81.35) | 16.31 (81.4) | 15.15 (80.22) | 16.62 (79.37) | 15.71 (83.16) | 15.47 (81.35) | 16.31 (81.4) | 15.15 (80.22) | 16.62 (79.37) | 15.71 (83.16) |
| Segment f1 H | 30.95 (63.48) | 32.18 (61.08) | 31.04 (61.09) | 30.71 (60.02) | 32.09 (59.54) | 30.95 (63.48) | 32.18 (61.08) | 31.04 (61.09) | 30.71 (60.02) | 32.09 (59.54) |
| Segment f1 G | 4.54 (82.35) | 7.27 (72.63) | 7.56 (71.91) | 7.96 (75.06) | 7.24 (70.1) | 4.54 (82.35) | 7.27 (72.63) | 7.56 (71.91) | 7.96 (75.06) | 7.24 (70.1) |
| Segment f1 I | 3.17 (93.94) | 4.06 (83.44) | 2.2 (90.89) | 1.28 (93.18) | 3.15 (82.77) | 3.17 (93.94) | 4.06 (83.44) | 2.2 (90.89) | 1.28 (93.18) | 3.15 (82.77) |
| Segment f1 E | 35.43 (62.51) | 26.32 (64.46) | 27.09 (64.47) | 21.22 (65.66) | 24.68 (63.9) | 35.43 (62.51) | 26.32 (64.46) | 27.09 (64.47) | 21.22 (65.66) | 24.68 (63.9) |
| Segment f1 B | 2.46 (99.67) | 2.06 (98.38) | 2.25 (98.44) | 2.31 (98.86) | 1.85 (98.46) | 2.46 (99.67) | 2.06 (98.38) | 2.25 (98.44) | 2.31 (98.86) | 1.85 (98.46) |
| Segment f1 T | 14.37 (71.22) | 14.33 (73.02) | 13.94 (71.35) | 14.72 (71.68) | 14.18 (73.37) | 14.37 (71.22) | 14.33 (73.02) | 13.94 (71.35) | 14.72 (71.68) | 14.18 (73.37) |
| Segment f1 S | 12.6 (79.62) | 10.96 (83.99) | 10.36 (83.6) | 11.91 (84.44) | 10.68 (83.93) | 12.6 (79.62) | 10.96 (83.99) | 10.36 (83.6) | 11.91 (84.44) | 10.68 (83.93) |
| Segment f1 - | 20.45 (84.94) | 21.18 (86.73) | 20.3 (86.08) | 22.14 (85.95) | 20.82 (88.68) | 20.45 (84.94) | 21.18 (86.73) | 20.3 (86.08) | 22.14 (85.95) | 20.82 (88.68) |

**Table 9:** Table 8 with averages over 10 datasets. We also report global averages over all 8 SSEs.

| Metric | H | G | I | E | B | T | S | - | Avg (HGIEBTS-) |
|---|---|---|---|---|---|---|---|---|---|
| Mean recall | 31.98 (97.32) | 29.62 (98.80) | 20.09 (99.79) | 35.58 (98.20) | 36.20 (100.00) | 32.96 (97.26) | 33.85 (98.08) | 33.17 (96.81) | 31.68 (98.28) |
| Mean precision | 33.12 (49.99) | 27.76 (67.61) | 19.42 (90.31) | 34.22 (47.65) | 24.55 (68.95) | 28.49 (51.12) | 25.65 (50.71) | 26.79 (50.23) | 27.50 (59.57) |
| Mean f1 | 32.01 (60.45) | 28.54 (74.28) | 19.70 (91.89) | 34.66 (60.56) | 28.41 (71.15) | 30.20 (62.65) | 28.46 (58.40) | 28.92 (61.08) | 28.86 (67.56) |
| Mean iou | 31.09 (60.18) | 27.44 (74.21) | 19.23 (91.88) | 33.58 (60.20) | 24.54 (69.05) | 28.23 (60.93) | 25.51 (53.77) | 26.22 (55.27) | 26.98 (65.69) |
| Segment recall | 32.05 (99.80) | 29.71 (99.97) | 20.14 (100.00) | 35.63 (99.77) | 35.74 (99.05) | 32.95 (98.77) | 33.48 (98.36) | 33.38 (98.51) | 31.64 (99.28) |
| Segment precision | 34.05 (61.01) | 4.00 (74.42) | 1.55 (88.84) | 23.95 (64.15) | 1.14 (98.77) | 9.36 (71.67) | 7.04 (82.45) | 15.85 (81.10) | 12.12 (77.80) |
| Segment f1 | 31.39 (61.04) | 6.91 (74.41) | 2.77 (88.84) | 26.95 (64.20) | 2.19 (98.76) | 14.31 (72.13) | 11.30 (83.12) | 20.98 (86.48) | 14.60 (78.62) |

**Secondary Structure Element (SSE) Agreement Results.** We ran a new expert agreement evaluation against the 8 basic SSEs (from DSSP). The summary is in 15. We do see above-random enrichment of SSEs in our tokens across all-metrics. The recall of existing SSEs is exceptional: $98.28\%$ block-level, $99.28\%$ segment-level, averaged over datasets and SSEs. This indicates the ability to recapitulate the 8 known elements, while the milder precision (59.57 block, 77.80 segment) hints that GeoBPE goes beyond SSEs. As prior agreement results and case studies show, GeoBPE can find biologically meaningful regions (e.g. conserved homology, functional sites) and is not constrained to SSE boundaries, with the data dictating the exact high-level clusters.

## D    TOKEN EFFICIENCY METRICS

Let $v = \{1, \ldots, K\}$ denote the codebook (size $K$). Given a corpus tokenized into a flat list of code indices, let $c_j$ be the count of code $j$ and $N = \sum_{j=1}^{K} c_j$ the total token count. We define the empirical unigram distribution

$$p(j) = \frac{c_j}{N} \quad \text{for } j \in v.$$

**Utilization rate (UR).** UR measures how many distinct codes are actually used:

$$\text{UR} \;=\; \frac{1}{K} \left| \{\, j \in v : c_j > 0 \,\} \right| \; \in [0, 1].$$

We report UR in percent. UR is important for diagnosing codebook collapse, a well-known phenomenon in VQ-VAEs where only a small number of codes are actively used Zhang et al. (2024a). This creates a quantization bottleneck, handicapping the tokenizer's performance and efficiency Yuan et al. (2025).

**Unigram entropy and perplexity.** Using the Shannon entropy (natural logarithm),

$$H \;=\; -\sum_{j \in v} p(j) \log p(j), \qquad \text{PPL} \;=\; \exp(H).$$

This *codebook perplexity* reflects how uniformly codes are used (model-free, ignores sequence context).

**Max-normalized perplexity.** Because the maximum entropy at uniform is $\log K$ (hence $\text{PPL}_{\max} = K$), we also report the scale-free ratio

$$\widetilde{\text{PPL}} \;=\; \frac{\text{PPL}}{K} \;=\; \exp\!\left(\frac{H}{\log K} \cdot \log K\right) \frac{1}{K} \;=\; \exp\!\left(H - \log K\right) \;\in (0, 1].$$

**Table 10:** We evaluate token efficiency of GEOBPE across varying $|\mathcal{V}| \in \{600, 2500, 6000\}$, as reported in Figure 4, with Hyperparameter Settings 1.

| Method | Codebook Size | UR (%) | | Perplexity | |
|---|---|---|---|---|---|
| | | CAMEO | CASP14 | CAMEO | CASP14 |
| VQ-VAE | 512 | 5.55 | 5.60 | 0.034 | 0.0337 |
| AminoASeed | 512 | 64.45 | 68.87 | 0.495 | 0.5119 |
| ESM3 | 4096 | 27.60 | 32.10 | 0.249 | 0.2841 |
| FoldSeek | 20 | 99.00 | 100.00 | 0.755 | 0.7435 |
| ProToken | 512 | 69.88 | 75.56 | 0.537 | 0.5697 |
| | 600 | 59.81 | 39.48 | 0.397 | 0.403 |
| GEOBPE | 2500 | 58.24 | 38.22 | 0.274 | 0.264 |
| | 6000 | 53.73 | 31.30 | 0.242 | 0.222 |

## E    SMALL STRUCTURE LANGUAGE MODEL EVALUATION

**Goals & Aims.** This section specifies a protocol used to compare the language modeling efficiency of GEOBPE vs VQ-VAE tokens. The goal of SSLM was to create a small, isolated environment for language model integration. Modeling is **intentionally minimalistic**–we train a small decoder-only

**Table 11:** We adopt the Small Structure Language Model evaluation protocol described in App. E. We sample 100 PDB structures. Best scTM and Designability scores are **bolded**. *Underlined* methods follow the evaluat -ion protocol in App. F.

| Small Structure Language Model Evaluation | | | | | |
|---|---|---|---|---|---|
| Method | Codebook Size | scTM | Designability (scTM > 0.5) | Diversity (1 - mean TM) | Uniqueness (TM=0.5) |
| VQ-VAE | 512 | 0.205 | 1% | 0.752 | 98% |
| AminoASeed | 512 | 0.186 | 1% | 0.476 | 16% |
| GEOBPE | 600 | 0.268 | 3% | 0.768 | 99% |
| | 2500 | 0.267 | 3% | 0.766 | 99% |
| | 6000 | 0.277 | 4% | 0.763 | 98% |
| GEOBPE (x10 data) | 600 | 0.376 | 12% | 0.743 | 83% |
| GEOBPE (x10 data, x10 params) | 600 | **0.405** | **21%** | 0.731 | 76% |

Transformer (7.3M) parameters over the same PDB splits used in Sec. 4 (48k structures). Fixing the same data splits, training, model architecture and hyperparameters (App. E.6), the aim of this experiment is to compare tokenizer options for a structure token language model. We emphasize the generation quality of the resulting models should be compared *relatively*; all models trained would be insufficient for real-world backbone design tasks. To bridge the gap with large-scale models, we provide an orthogonal study on *scalability* in App. F, where we reran SSLM with 10x more data and model parameters.

**Setup.** For GEOBPE, we use the joint geometric vocabulary learned by GEOBPE; for VQ-VAE, we use their codebook. GEOBPE +SSLM incorporates the mask constraints used at generation time (Alg. 4) during training to ensure consistency between training and sampling. The same procedure is used for evaluating VQ-VAEs. For training and sampling, the only difference is dropping the mask constraints. For inference, the samples are passed through the VQ-VAE decoder to construct back-bone coordinates instead of assembling the backbone directly via Alg. 5. This required considerably more resources, and we discuss how we implemented this in App. E.6.

---

**Algorithm 2** GEOLM-PRETRAIN — decoder-only next-token prediction on geometric tokens

---

**Require:** Corpus of proteins $\{t_\tau\}_{\tau=1}^T$ with final segmentations $\{\mathcal{P}^{(\tau)}\}$ and assigned medoids; joint vocabulary $\Sigma$ and tokenizers from Alg. 16, 17; a decoder-only Transformer $\mathsf{Tr}_\theta : \Sigma^* \to \Delta^{|\Sigma|}$ with causal mask; special BOS/EOS (optional); training steps $S$, optimizer $\mathcal{O}$.

**Ensure:** Trained parameters $\theta$.

1: **Dataset construction.** For each $\tau$, build $x^{(\tau)} = \text{BACKBONETOSEQUENCE}(t_\tau)$ (Alg. 17). Let $L_\tau = |x^{(\tau)}|$.

2: **Objective.** For any sequence $x = (x_1, \ldots, x_L)$, define

$$\mathcal{L}_{\text{NTP}}(\theta; x) = -\sum_{t=1}^{L-1} \log p_\theta(x_{t+1} \mid x_{\leq t}), \quad p_\theta(\cdot | x_{\leq t}) = \text{softmax}(\mathsf{Tr}_\theta(x_{\leq t})).$$

3: **Training loop.**
4: **for** $s = 1$ **to** $S$ **do**
5:     Sample a minibatch $\mathcal{B} \subset \{1, \ldots, T\}$.
6:     $\mathcal{L} \leftarrow \frac{1}{|\mathcal{B}|} \sum_{\tau \in \mathcal{B}} \mathcal{L}_{\text{NTP}}(\theta; x^{(\tau)})$.
7:     Update $\theta \leftarrow \mathcal{O}(\theta, \nabla_\theta \mathcal{L})$.
8: **end for**
9: **return** $\theta$.

---

### E.1 DATA PREPARATION AND SPLITS

**Tokenization.** We construct the joint vocabulary $\Sigma$ (Alg. 16) and convert each protein $t_\tau$ into a token sequence $x^{(\tau)} = (x_1^{(\tau)}, \ldots, x_{L_\tau}^{(\tau)})$ via BACKBONETOSEQUENCE (Alg. 17).

We tokenize the validation/test sets (unseen during GEOBPE training) via Algo. 10, a procedure analogous to BPE encoding. Sequences alternate strictly motif $\to \theta \to \omega \to \phi \to$ motif $\to \cdots$ and end with a *terminating* motif token (length-2 bond–residue class), hence $L_\tau \equiv 1 \pmod 4$.

---

**Algorithm 3** BUILDEMPIRICALPRIORS — length prior and first-token prior

---

**Require:** Training corpus of tokenized backbones $\{x^{(\tau)} = (x_1^{(\tau)}, \ldots, x_{L_\tau}^{(\tau)})\}_{\tau=1}^T$ constructed by Alg. 17 (motif, then $\theta, \omega, \phi$, repeating); valid sequence lengths satisfy $L_\tau \equiv 1 \pmod 4$ and end in a *terminating* motif token.
**Ensure:** Discrete priors $\Pi_L$ on lengths $K$ and $\Pi_{\text{start}}$ on the first token.
 1: **Length prior:** for every $K$ with $K \equiv 1 \pmod 4$, set

$$\Pi_L(K) \;\propto\; \left|\{\tau : \; L_\tau = K\}\right| \;\text{ and normalize }\; \sum_K \Pi_L(K) = 1.$$

 2: **First-token prior:** over motif tokens only, set

$$\Pi_{\text{start}}(i) \;\propto\; \left|\{\tau : \; x_1^{(\tau)} = i\}\right|, \quad i \in \Sigma_{\text{med}}; \quad \sum_{i \in \Sigma_{\text{med}}} \Pi_{\text{start}}(i) = 1.$$

 3: **return** $\Pi_L$, $\Pi_{\text{start}}$.

---

**Algorithm 4** UNCONDITIONALGEOLMGENERATE — motif/glue token generation

---

**Require:** Trained decoder-only Transformer $\text{Tr}_\theta$ with vocabulary $\Sigma$ from Alg. 16; id blocks

$$\Sigma_{\text{med}} = \{1, \ldots, M\} \tag{1}$$
$$\Sigma_\theta = \{M{+}1, \ldots, M{+}B_\theta\} \tag{2}$$
$$\Sigma_\omega = \{M{+}B_\theta{+}1, \ldots, M{+}B_\theta{+}B_\omega\} \tag{3}$$
$$\Sigma_\phi = \{M{+}B_\theta{+}B_\omega{+}1, \ldots, M{+}B_\theta{+}B_\omega{+}B_\phi\}; \tag{4}$$

terminating-motif set $\Sigma_{\text{term}} \subseteq \Sigma_{\text{med}}$ (motifs in the length-2 bond-residue class); priors $\Pi_L, \Pi_{\text{start}}$ (Alg. 3); temperature $\tau > 0$; maximum length $K_{\text{max}}$; number of samples $S$.
**Ensure:** $S$ unconstrained token sequences $\{x^{(s)}\}$ alternating motif and glue tokens and ending in a terminating motif.
 1: Define the **type mask by position** ($t$ starts at 1):

$$t \equiv 1 \pmod 4 \Rightarrow \text{motif } (\Sigma_{\text{med}}), \quad t \equiv 2 \Rightarrow \theta \, (\Sigma_\theta), \quad t \equiv 3 \Rightarrow \omega \, (\Sigma_\omega), \quad t \equiv 0 \Rightarrow \phi \, (\Sigma_\phi).$$

 2: **for** $s = 1$ **to** $S$ **do**
 3:     Sample a target cap $K^{\text{cap}} \sim \Pi_L$ and set $K^{\text{cap}} \leftarrow \min(K^{\text{cap}}, K_{\text{max}})$.
 4:     Sample the first token $x_1^{(s)} \sim \Pi_{\text{start}}$ (so $x_1^{(s)} \in \Sigma_{\text{med}}$).
 5:     **for** $t = 2, 3, \ldots, K^{\text{cap}}$ **do**
 6:         Compute last-position logits $z_t = \text{Tr}_\theta\big(x_{1:t-1}^{(s)}\big)$ with causal masking; let $v = |\Sigma|$.
 7:         Build a **hard mask** $m \in \mathbb{R}^v$ initialized to $-\infty$ and set:

$$\begin{cases} m_i \leftarrow 0 & \text{if } t \equiv 1 \, (4) \text{ and } i \in \Sigma_{\text{med}}, \\ m_i \leftarrow 0 & \text{if } t \equiv 2 \, (4) \text{ and } i \in \Sigma_\theta, \\ m_i \leftarrow 0 & \text{if } t \equiv 3 \, (4) \text{ and } i \in \Sigma_\omega, \\ m_i \leftarrow 0 & \text{if } t \equiv 0 \, (4) \text{ and } i \in \Sigma_\phi. \end{cases}$$

 8:         **Termination constraint at motif positions:**
        •   If $t \equiv 1 \, (4)$ and $t < K^{\text{cap}}$, then *disallow* early stop: set $m_i \leftarrow -\infty$ for all $i \in \Sigma_{\text{term}}$.
        •   If $t \equiv 1 \, (4)$ and $t = K^{\text{cap}}$, then *force* stop: set $m_i \leftarrow -\infty$ for all $i \in \Sigma_{\text{med}} \setminus \Sigma_{\text{term}}$.
 9:         Form masked logits $\tilde{z}_t = z_t + m$ and sample

$$x_t^{(s)} \sim \text{Categorical}(\text{softmax}(\tilde{z}_t/\tau)).$$

10:         **(Optional early stop)** If $t \equiv 1 \, (4)$ and $x_t^{(s)} \in \Sigma_{\text{term}}$, then **break**.
11:     **end for**
12: **end for**
13: **return** $\{x^{(s)}\}_{s=1}^S$.

---

**Splits.** We partition proteins at the *protein level* into train/validation/test (e.g., 80/10/10) to prevent leakage across chains.

---

**Algorithm 5** DEQUANTIZEANDASSEMBLE — from tokens to a full backbone

---

**Require:** One generated sequence $x = (x_1, \ldots, x_L)$ from Alg. 4; medoid dictionary $\{\mathrm{id}_{\mathrm{med}}(\kappa, j) \mapsto \Pi_j^{(\kappa)}\}$ where each prototype $\Pi_j^{(\kappa)}$ is a tuple of internal coordinates for a motif $\mathcal{M}$; glue bin edges $\{\beta_b^\theta\}_{b=0}^{B_\theta}, \{\beta_b^\omega\}_{b=0}^{B_\omega}, \{\beta_b^\phi\}_{b=0}^{B_\phi}$ (circular edges for angles, linear for lengths if used); canonical seed triad $(N_\star, \mathrm{CA}_\star, C_\star)$ and SEEDTRIAD.

**Ensure:** A complete backbone $\big\{(N_i, \mathrm{CA}_i, C_i) \in \mathbb{R}^3\big\}_{i=1}^{\widehat{N}}$ assembled from the decoded motifs and glues.

 1: **Parse tokens into motifs and glues (fixed 4-cycle).** Let the motif indices be $t \in \{1, 5, 9, \ldots\}$; write $x_t = \mathrm{id}_{\mathrm{med}}(\kappa^{(m)}, j^{(m)})$ for $m = 1, \ldots, M$ where $M = \frac{L+3}{4}$. For each boundary $m = 1, \ldots, M-1$, decode the three bins:

$$b_\theta = x_{4m-2} - M, \quad b_\omega = x_{4m-1} - (M+B_\theta), \quad b_\phi = x_{4m} - \big(M+B_\theta+B_\omega\big),$$

and **dequantize** to the bin midpoints

$$\bar\theta_m = \tfrac{1}{2}(\beta_{b_\theta-1}^\theta + \beta_{b_\theta}^\theta), \quad \bar\omega_m = \tfrac{1}{2}(\beta_{b_\omega-1}^\omega + \beta_{b_\omega}^\omega), \quad \bar\phi_m = \tfrac{1}{2}(\beta_{b_\phi-1}^\phi + \beta_{b_\phi}^\phi).$$

 2: **Recover internal coordinates.** For each motif $m$, let $\Pi_{j^{(m)}}^{(\kappa^{(m)})}$ provide the internal bond lengths $\ell$, bond angles $\theta$, and dihedrals $(\psi, \omega, \phi)$ *across its span* $\mathcal{M}^{(m)}$. Construct its internal entry→exit transform $T_{(m)}^{\mathrm{int}}$ (product of link transforms $G_i$ inside the motif; see Preliminaries).
 3: **Forward kinematics assembly.**
 4: Initialize the entry frame by seeding the very first residue: $(N_1, \mathrm{CA}_1, C_1) \leftarrow \mathrm{SEEDTRIAD}(1)$ and form $F_1 = (R_1, t_1)$ as in the Entry/Exit frame definition.
 5: **Motif 1:** Traverse the links inside $\mathcal{M}^{(1)}$ using its internal coordinates to compute frames $F_2, \ldots, F_{q_1}$ (and atom positions) by repeated $G_i$ multiplications; set the current exit frame $F_{(1)}^{\mathrm{exit}} = F_{q_1}$.
 6: **for** $m = 1$ **to** $M-1$ **do**
 7:     **Boundary glue:** form the boundary transform

$$T_{(m)}^{\mathrm{glue}} = G_{q_m}\big(\theta^{CNCA} = \bar\theta_m,\ \omega = \bar\omega_m,\ \phi = \bar\phi_m\big),$$

    i.e., the SE(3) map from the exit frame of $\mathcal{M}^{(m)}$ to the entry frame of $\mathcal{M}^{(m+1)}$ determined by the three dequantized glue angles (and adjacent bond lengths).
 8:     Set the entry frame of $\mathcal{M}^{(m+1)}$ to

$$F_{(m+1)}^{\mathrm{entry}} \leftarrow T_{(m)}^{\mathrm{glue}}\ F_{(m)}^{\mathrm{exit}}.$$

 9:     **Motif** $(m+1)$**:** traverse its internal links to produce all residue frames and atom positions; update $F_{(m+1)}^{\mathrm{exit}}$.
10: **end for**
11: **Concatenate atoms.** Collect the atoms from all traversals in order, yielding the backbone $\big\{(N_i, \mathrm{CA}_i, C_i)\big\}_{i=1}^{\widehat{N}}$, where $\widehat{N}$ is the total number of residues implied by the concatenated motif spans (the final motif is guaranteed terminating).
12: **return** the complete backbone coordinates.

---

## E.2 TRAINING OBJECTIVE WITH STRUCTURAL MASKS

We train a causal Transformer $\mathrm{Tr}_\theta$ with teacher forcing. To enforce legality at each position $t$, we apply the same *type mask by position modulo 4* used in generation (Alg. 4):

$$t \equiv 1 \,(\mathrm{mod}\ 4) \Rightarrow \Sigma_{\mathrm{med}}, \quad t \equiv 2 \Rightarrow \Sigma_\theta, \quad t \equiv 3 \Rightarrow \Sigma_\omega, \quad t \equiv 0 \Rightarrow \Sigma_\phi,$$

setting logits for all other token types to $-\infty$ before the softmax.

**Termination constraint at motif slots.** At motif positions ($t \equiv 1 \,(\mathrm{mod}\ 4)$), we impose the same termination rule as in Alg. 4: (i) if $t < L_\tau$, mask out terminating motifs $\Sigma_{\mathrm{term}}$; (ii) if $t = L_\tau$, mask out non-terminating motifs $\Sigma_{\mathrm{term}}$.

**Loss.** With masks applied, the negative log-likelihood is

$$\mathcal{L}_{\mathrm{NTP}}(\theta; x^{(\tau)}) = -\sum_{t=1}^{L_\tau - 1} \log p_\theta\big(x_{t+1}^{(\tau)} \mid x_{\leq t}^{(\tau)}\big), \qquad p_\theta(\cdot \mid x_{\leq t}) = \mathrm{softmax}(\tilde z_t),$$

where $\tilde z_t$ are masked logits. We optimize $\theta$ by minimizing the average NLL over the training set.

**Early stopping.** We select checkpoints by validation loss with a patience of 5 epochs.

### E.3 UNCONDITIONAL SAMPLING FOR QUALITATIVE EVALUATION

**Empirical priors.** We form the *length prior* $\Pi_L$ and *first-token prior* $\Pi_{\text{start}}$ from the training corpus using BUILDEMPIRICALPRIORS (Alg. 3). $\Pi_L$ is supported on legal lengths $K \equiv 1 \pmod 4$; $\Pi_{\text{start}}$ is over $\Sigma_{\text{med}}$.

**Constrained generation.** We sample with UNCONDITIONALGEOLMGENERATE (Alg. 4): draw $K^{\text{cap}} \sim \Pi_L$ (clipped by a maximum), sample the first motif $x_1 \sim \Pi_{\text{start}}$, then autoregress under the same positional type mask and termination constraint as training. Temperature and nucleus sampling are optional ablations.

**GEOBPE Dequantization and assembly.** Generated token sequences are mapped to full backbones via DEQUANTIZEANDASSEMBLE (Alg. 5): medoid tokens decode to internal coordinates over their motif spans; glue-bin tokens decode to bin-midpoint angles; forward kinematics with the seeded entry frame yields atom coordinates $\{(N_i, \text{CA}_i, C_i)\}_{i=1}^{\widehat{N}}$.

### E.4 GENERATIVE QUALITY ASSESSMENT

We evaluate unconditional samples produced by UNCONDITIONALGEOLMGENERATE (Alg. 4) and assembled by DEQUANTIZEANDASSEMBLE (Alg. 5) using four structure-centric metrics based on TM-score.[1]

**Setup.** From each model we draw a fixed number of backbones $\{\widehat{\mathcal{B}}_n\}_{n=1}^N$ (legal lengths, terminal motif constraint). Unless noted, metrics are computed on these *backbone geometries* without further post-processing.

**(1) scTM (self-consistency TM-score).** For each generated backbone $\widehat{\mathcal{B}}$, we (i) design a sequence $\widehat{s}$ with a standard inverse-folding model, (ii) predict a structure $\widetilde{\mathcal{B}}$ from $\widehat{s}$ using a single-structure predictor (e.g., ESMFold), and (iii) compute

$$\text{scTM}(\widehat{\mathcal{B}}) = \text{TM-score}(\widetilde{\mathcal{B}}, \widehat{\mathcal{B}}).$$

We report the mean scTM over the $N$ samples.

**(2) Designability (% with scTM $> 0.5$).** A backbone is deemed *designable* if its self-consistency exceeds the canonical threshold $0.5$:

$$\text{Designability} = \frac{1}{N} \sum_{n=1}^N \mathbf{1}\left\{\text{scTM}(\widehat{\mathcal{B}}_n) > 0.5\right\} \times 100\%.$$

This is the fraction of samples for which a designed sequence refolds back to the generated backbone at the fold level. We adopt the same workflow from Trippe et al. (2022); Wu et al. (2024), where ProteinMPNN Dauparas et al. (2022) proposes 8 sequences per structure and OmegaFold Wu et al. (2022) is used to compute scTM.

**(3) Diversity (mean pairwise TM).** To quantify sample-to-sample diversity, we compute the mean pairwise TM-score across the set (lower is more diverse):

$$\text{Diversity} = \frac{2}{N(N-1)} \sum_{1 \le i < j \le N} \text{TM-score}(\widehat{\mathcal{B}}_i, \widehat{\mathcal{B}}_j).$$

(When $N$ is large, we estimate this by uniform sub-sampling of pairs.)

**(4) Uniqueness (% non-duplicates at TM $< 0.5$).** We mark a sample as *unique* if its nearest neighbor among the other generated backbones has TM-score $< 0.5$:

$$\text{Uniqueness} = \frac{1}{N} \sum_{n=1}^N \mathbf{1}\left\{\max_{m \ne n} \text{TM-score}(\widehat{\mathcal{B}}_n, \widehat{\mathcal{B}}_m) < 0.5\right\} \times 100\%.$$

This measures the proportion of samples that are not near-duplicates under a fold-level threshold.

**Reporting.** For each model we report the four metrics above on the same set size $N$ (and the same sampling priors and temperature). Codebook size and token perplexity are *not* used in these downstream comparisons.

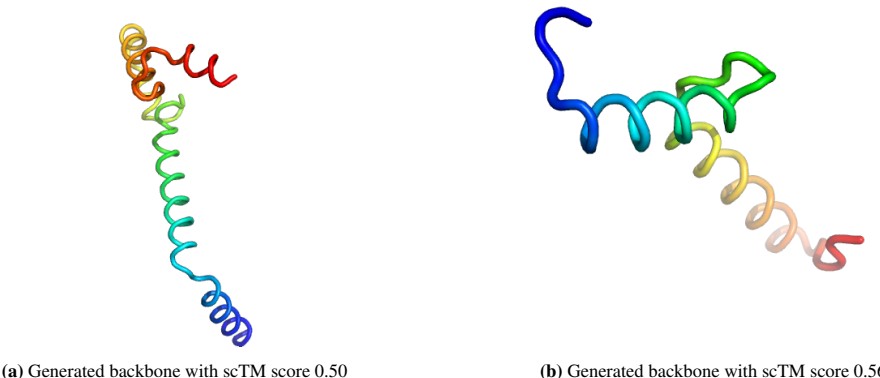

(a) Generated backbone with scTM score 0.50    (b) Generated backbone with scTM score 0.56

**Figure 7:** We visualize two backbones generated by GEOBPE SSLM-Eval (with default settings in App. K).

### E.5 GENERATED BACKBONES

In Fig. 7a, we see a long, well-structured and assembled $\alpha$-helix, which is one of the most common and stable secondary structures in proteins. The curved helical cap at the top resembles a common N-terminal capping motif, which often stabilizes helices through hydrogen bonding networks or electrostatic interactions. Such elongated $\alpha$-helices are commonly found in transmembrane helices or coiled-coil domains which are involved in dimerization and DNA-binding. The overall curvature and spatial continuity also suggest potential compatibility with membrane proteins or structural scaffolds, especially behave as substance binding receptors as well as ion channels.

In Fig. 7b, we see a structure that resembles DNA-binding motifs or cytokine folds, which are quite well-known for cellular signaling or regulation. The geometric density of this structure also suggests a pre-organized hydrophobic core, which is critical for proper folding and stability in the cytoplasmic environment. This structure exhibits a compact bundle of helices with apparent crossing angles which are similar to some small globular domains in common protein structures. The folding appears non-linear but in a quite controlled, manner which suggests potential tertiary structure forming interactions such as hydrophobic-hydrophobic interaction.

### E.6 IMPLEMENTATION DETAILS

**SSLM-Eval GEOBPE implementation and hardware details.** We train a small autoregressive Transformer on discretized geometry tokens. We use a hidden size $d_{\text{model}} = 256$, $L = 8$ Transformer layers with GELU activations, $H = 8$ attention heads, and feed-forward width $d_{\text{ff}} = 1024$. Token and positional embeddings are summed, a LayerNorm is applied before the classifier, and the output projection is weight–tied to the token embedding. A causal attention mask enforces left-to-right prediction. Sequences are padded to a dataset-dependent maximum length (the 95th percentile of training lengths by default). We optimize cross-entropy loss with Adam (learning rate $1 \times 10^{-4}$), batch size 32, for up to 100 epochs with early stopping on validation perplexity. For unconditional generation, we sample 100 sequences at temperature 1.0, drawing target lengths from the empirical length prior (restricted to valid lengths by construction) and the first token from the empirical start-token prior; decoding proceeds token-by-token under the causal mask. On a single GPU, one epoch takes just under 10 mins and converges in $\approx 60$ epochs (can vary across tokenizer settings). The data splits are the same as those for pretraining (see App. B) – 33992, 3810 training/validation structures for GEOBPE.

**SSLM-Eval VQ-VAE implementation and hardware details.** We extend the distributed Lightning setup of Yuan et al. (2025) with a self-contained evaluation step at the end of each validation epoch. Using 4 ranks, each GPU accumulates the epoch's quantized token sequences from training and validation; these are gathered and passed to a lightweight auxiliary trainer that uses the same SSLM-Eval script as GEOBPE and hyperparameters. After convergence, we sample 100 new token

---

[1]TM-score is obtained with a standard implementation (e.g., TM-align); higher is better.

sequences, decode them with the VQ-VAE decoder into backbone coordinates, and write PDBs to a directory named by the current epoch. We compute all non-SCTM metrics locally, then distribute a heavier SCTM evaluation across ranks on sharded PDB subsets. Each rank produces its shard's results, and rank-0 merges them into a single summary that is logged to the trainer.

## F SCALING GEOBPE TO LARGER DATASETS AND MODELS

**Goals & Aims.** The relatively minimalistic design of GEOBPE SSLM in App. E still begs the question whether GEOBPE reliably scales to large datasets (e.g. ESM3 scale) as both a *tokenizer* (GEOBPE only) and the foundational component of a language model (GEOBPE + SSLM). Thus, we attempt to bridge this gap by (i) tokenizing a 10x larger dataset and benchmarking wall times, and (ii) training a 10x larger SSLM model on the 10x larger tokenized corpus.

### F.1 TOKENIZING A 10X LARGER DATASET OF PREDICTED STRUCTURES

**Setup.** We downloaded the 550K Swiss-Prot structure predictions from AlphaFold DB Consortium (2024); Varadi et al. (2022), a $> 10$x increase from our PDB pretraining dataset. We adopt the pretrained $|V| = 600$ tokenizer used in the paper (Fig. 4, Tables 10 & 11) as a baseline (i.e. tokenizers with larger $|V|$ will achieve lower distortion at the tradeoff of slower throughput).

**Evaluation.** We log both wall-time and thoroughput taken to tokenize all 550k structures. We also report the distortion against AlphaFold DB predictions. We split into five 110K increments, and requested 5 jobs with 20 cores each. Each job writes each tokenized structure to a file, which allows us to log the running throughput from start to finish.

**Table 12:** We report throughput and distortion tokenizing 550k Swiss-Prot predicted structures with 5 jobs of 20 cores each.

|                            | Split (110K increments) | | | | | |
|----------------------------|-------|-------|-------|-------|-------|-------------------|
|                            | 1     | 2     | 3     | 4     | 5     | Total (5 splits). |
| Avg. Throughput (files/min)| 35.54 | 45.40 | 35.45 | 36.64 | 38.29 | 191.32            |
| RMSD                       | 1.52  | 1.54  | 1.54  | 1.53  | 1.53  | 1.53              |
| LDDT                       | 0.79  | 0.79  | 0.79  | 0.79  | 0.79  | 0.79              |

**Results.** The results are shown in Table. 12. With pooled avg. throughput of 191.32, all 550k structures were tokenized in $\sim 2$ days. Discrepancies in throughput between jobs likely explained by node traffic. We see all 5 splits achieve the same LDDT (0.79) and within 0.01 RMSD of the average RMSD over all 5 splits (1.53). These are comparable to the tokenizer's OOD test set distortions in Fig. 4 (1.53 RMSD, 0.72 on CASP).

**Conclusion.** Since GEOBPE has shown strong OOD generalization from our findings in Sec. 4, it is expected to not degrade in performance. Thus, the right approach is tokenization (Alg. 10) rather than retraining GEOBPE from scratch. In contrast with GEOBPE learning, tokenization is an *embarrassingly parallel* procedure that easily scales with the number of cores. With 100 cores, the entire process finished in $\sim 2$ days, or 200 CPU days. Further scaling by another 10x would only take $\sim 20$ days with 100 CPUs, making GEOBPE a scalable solution for tokenizing large databases of predicted structures.

### F.2 STRUCTURE LANGAUGE MODELING AT A 10X LARGER SCALE

Once we have the tokenized dataset of 550k Swiss-Prot structures, we train a larger model and probe whether generation quality of GEOBPE SSLM follows the expected improvements from scaling.

**Setup.** We increased our Transformer to $\sim 10$x parameters ($7.3M \rightarrow 65.9M$). We do so by widening the Transformer layer and deepening the model: $d_{\text{model}} \leftarrow 2d_{\text{model}} = 512$, $L \leftarrow 2.5L = 20$ layers. $H \leftarrow 2 * H = 16$ attention heads, $d_{\text{ff}} \leftarrow 2 \cdot d_{\text{ff}} = 2048$. The rest of SSLM remains the same (App. E).

**Evaluation.** We used the same evaluation protocol (50-128 AAs, 10 each) of works such as ProtDiff Trippe et al. (2022) and FoldingDiff Wu et al. (2024). Note that App. E did not follow this protocol;

we sampled from the size prior of our pretraining dataset (Alg. 3); the average generated protein was $\sim 214$ AAs.

**Results.** GeoBPE+SSLM with 10x more data and 10x more parameters **achieves an average scTM of** $0.4051$**, with 20.8% being Designable (scTM $> 0.5$)**. This is notably higher than ProtDiff ($11.8\%$) and FoldingDiff ($14.2\%$). Scaling only by 10x more data also delivers a respectable scTM of $0.376$, highlighting both scaling dimensions are throttles for generative performance. Uniqueness/diversity also remain high ($76.4\%$ and $0.73$).

**Conclusion.** This result confirms that GEOBPE behaves according to scaling law expectations of language modeling. The significant increase in designable backbones ($4\% \rightarrow 21\%$) from simply using more data and parameters justifies further scaling of data and training resources. We hope future works can adopt GEOBPE as a foundational component in future large-scale models and explore the full potential of large-scale protein structure language model development.

# G EXPERT AGREEMENT METRICS

Our method segments a protein sequence into $M$ contiguous residue spans $P_j = [p_j, q_j]$ with $q_j + 1 = p_{j+1}$ for $j = 1, \ldots, M-1$. We compare these segments against $N$ ground-truth domain annotations $D_i = [s_i, e_i]$. All sets below are sets of integer residue indices and $|\cdot|$ denotes cardinality (length in residues). We report (i) *domain-level* alignment quality for each true domain using the best consecutive block of predicted segments, and (ii) *segment-level* detection statistics at an Intersection-over-Union (IoU) threshold $\tau$. This combination captures both *how well* each domain is covered and *how economically* the predicted segments explain the annotations, while remaining robust to small boundary jitter.

**Annotation source.** Ground-truth domains come from **CATH FunFams** Das et al. (2015b). They are functional families defined by *profile HMM* hits trained on primary-sequence data Das et al. (2015a;b). Our evaluation thus measures how well the predicted segmentation aligns with functionally coherent families derived from sequence-based HMM models.

In our setting, individual predicted segments tend to be substantially shorter than the curated domain annotations. A naive one-to-one comparison would systematically penalize predictions that must be *combined* to cover a domain. To ensure a fair comparison, for each $D_i$ we first select the single best *consecutive* block of predicted segments $S_i$ that maximizes IoU with $D_i$ (below), then compute per-domain scores and *macro-average* them so that each domain contributes equally, independent of its length.

**Notation and best block per domain.** For domains $D_i = [s_i, e_i]$ ($i = 1{:}N$) and predicted segments $P_j = [p_j, q_j]$ ($j = 1{:}M$, with $q_j + 1 = p_{j+1}$), define

$$(a_i, b_i) \in \arg \max_{1 \le m \le n \le M} \frac{\left| D_i \cap \bigcup_{k=m}^{n} P_k \right|}{\left| D_i \cup \bigcup_{k=m}^{n} P_k \right|}, \qquad S_i := \bigcup_{k=a_i}^{b_i} P_k.$$

(Ties may prefer the shortest $S_i$ or fewest segments.)

**Domain-level scores (macro).** Let $\mathrm{ov}_i = |D_i \cap S_i|$, $|D_i| = e_i - s_i + 1$, $|S_i| = \sum_{k=a_i}^{b_i} (q_k - p_k + 1)$. Then

$$\mathrm{Recall}_i = \frac{\mathrm{ov}_i}{|D_i|}, \quad \mathrm{Precision}_i = \frac{\mathrm{ov}_i}{|S_i|}, \quad F_{1,i} = \frac{2 \, \mathrm{Recall}_i \, \mathrm{Precision}_i}{\mathrm{Recall}_i + \mathrm{Precision}_i}, \quad \mathrm{IoU}_i = \frac{\mathrm{ov}_i}{|D_i| + |S_i| - \mathrm{ov}_i}.$$

Macro-averages:

$$\overline{\mathrm{Recall}} = \tfrac{1}{N} \sum_i \mathrm{Recall}_i, \quad \overline{\mathrm{Precision}} = \tfrac{1}{N} \sum_i \mathrm{Precision}_i, \quad \overline{F_1} = \tfrac{1}{N} \sum_i F_{1,i}, \quad \overline{\mathrm{IoU}} = \tfrac{1}{N} \sum_i \mathrm{IoU}_i.$$

*Interpretation:* recall rewards coverage; precision rewards compactness of $S_i$; $F_1$ balances both; IoU is thresholdable and scale-invariant.

**Segment-level detection at IoU threshold $\tau$.** Let $\mathcal{U} = \bigcup_{i: \mathrm{IoU}_i \ge \tau} \{a_i, \ldots, b_i\}$. Define

$$\mathrm{SegPrec} = \frac{|\mathcal{U}|}{M}, \qquad \mathrm{SegRec} = \frac{|\{ i : \mathrm{IoU}_i \ge \tau \}|}{N}, \qquad \mathrm{SegF}_1 = \frac{2 \, \mathrm{SegPrec} \, \mathrm{SegRec}}{\mathrm{SegPrec} + \mathrm{SegRec}}.$$

*Interpretation:* SegPrec penalizes unused segments; SegRec penalizes missed/poorly aligned domains. Sweeping $\tau$ yields a PR curve.

**Randomization baseline and reporting.** Using 1000 uniform random partitions into $M$ contiguous spans of the same sequence, recompute all metrics under the same best-block protocol and average over runs. We report using the format:

$$\text{ours (random-avg)},$$

e.g., $\overline{\text{IoU}} = 0.47\,(0.18)$.

**Notes.** In degenerate cases (e.g., $|S_i| = 0$ or a zero denominator), we adopt the standard convention of returning 0 for the affected ratio or $F_1$ term.

Together, the domain-level (overlap-quality) metrics and the segment-level (parsimony and coverage) metrics directly test the two desiderata of protein-domain segmentation: (i) accurate coverage of each domain with minimal spillover, and (ii) a parsimonious set of segments that explain as many domains as possible. Macro-averaging after selecting the best block per domain ensures fairness when predicted segments are shorter than annotated domains, and the permutation baseline quantifies how far performance rises above chance given the same $M$.

## H    EXPERT CASE STUDIES

**Individual Tokens Correspond to Secondary Structures.** Figure 8 is an example of a single token GEOBPE discovers. It features an alpha helix that includes aromatic cage (formed by Tryptophan / Tyrosine) and hydrogen bonding residue. It can be a common structure in Nucleotide-recognition domains, especially the hydrogen bond donors/acceptors can serve for specific molecular recognition (e.g., methylated lysines, nucleotide bases or acetyl groups) as well as Neurotransmitter receptors. From interpretation, this motif is functionally specific.

It can serve as ligand binding pocket, which is tightly packed and evolutionarily conserved. This could behave significantly in substance recognition. The tightly packed helical scaffold in this separated motif is likely stabilizing the motif's geometry and ensuring specificity. Motif-scaffold synergy can also help to define a structure's rigidity and flexibility.

**Merge Hierarchy of GEOBPE Reflects Combination of Secondary Structures for Driving Function.**

**Figure 9.** 4xk4 is the human mitochondrial carrier protein SLC25A20 (carnitine/acylcarnitine translocase). It's a transmembrane transport protein within the mitochondrial inner membrane, responsible for shuttling carnitine and acylcarnitine molecules across the membrane. This a process critical to fatty acid oxidation and energy metabolism. The core motif that the algorithm separated out contain three similar domains, each with two transmembrane

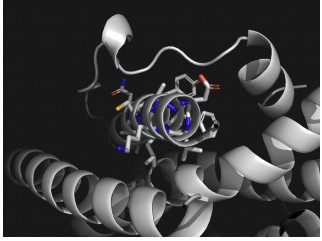

**Figure 8:** An exemplary GEOBPE token spans backbone atoms of an alpha helix (colored).

helices and a loop. It appears to lie deep within the transmembrane domain, forming part of the central binding cavity. From this know-how information, the 48-4 motif is really significant in the following three aspects:

1. It will serve for substrate recognition where the internal polar residues bind to the acylcarnitine or carnitine head group via ionic and hydrogen bonds. It will also alter the transition state of the during the transport cycles. For example, this motif can play a role in shift conformation between open-to-cytoplasm and open-to-matrix states.
2. We also observe similar motifs are found in other SLC25 family members (e.g., ADP/ATP carriers), indicating a shared mechanism of transport.
3. While the broader transmembrane region is dominated by repetitive helices, this localized motif exhibits a unique composition of diverse side chains, polar residues, and tightly packed interactions, reinforcing its functional specificity.

**Figure 10.** 6FI4 is the crystal structure of a hybrid peptide composed of a C-terminally modified Tau protein segment bound to the human 14-3-3$\sigma$ protein, solved at 2.0 Å resolution via X-ray crystallography. 14-3-3 proteins are a family of conserved regulatory molecules that bind

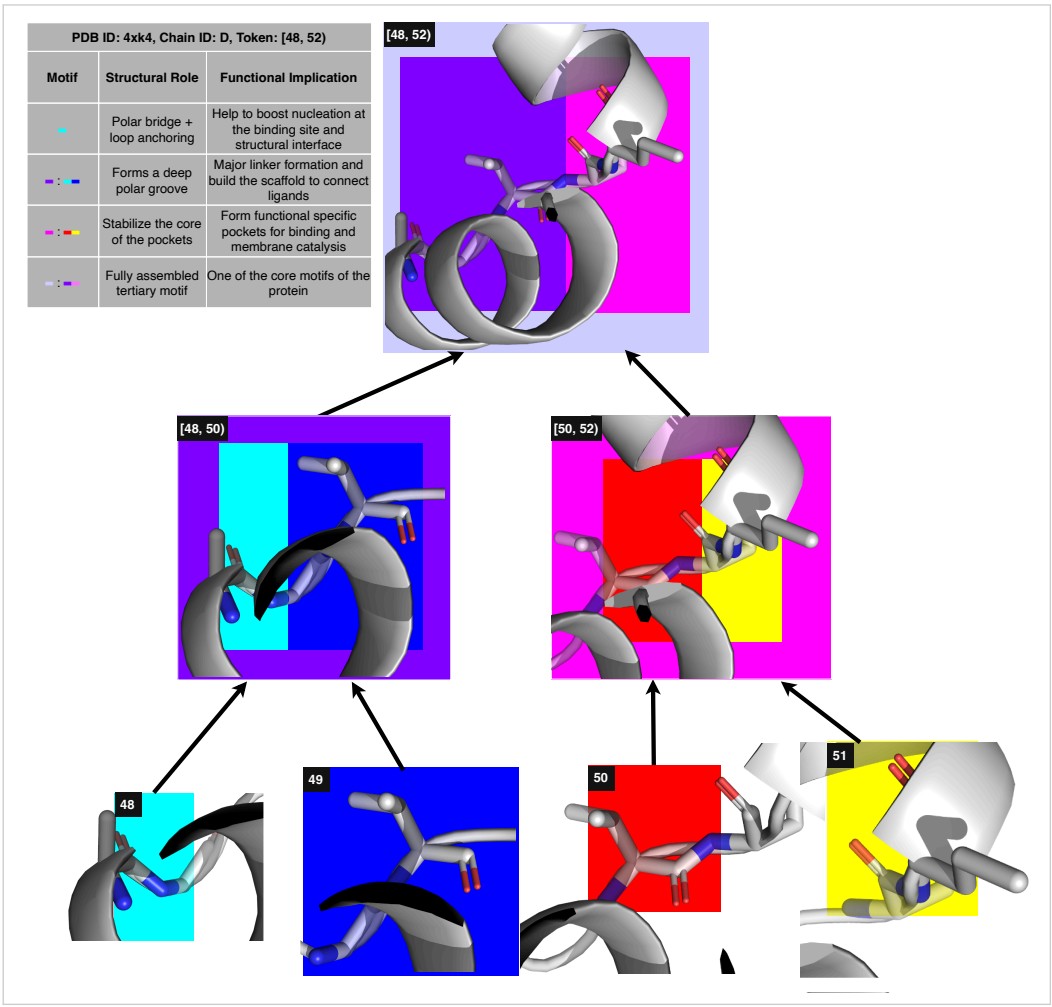

**Figure 9:** Chain D of PDB 4xk4. Hierarchical Merge Tree for Token [48, 52). GEOBPE arrived at this token by merging [48, 49) with [49, 50), [50, 51) with [51, 52), and [48, 50) with [50, 52).

phosphoserine/phosphothreonine-containing motifs on target proteins and are central to cell cycle control, apoptosis, transcriptional regulation, and signal transduction. The hybrid peptide mimics Tau phosphorylation, which is relevant to neurodegenerative disease pathology like Alzheimer's disease. From this know-how information, the 29-4 motif is significant in the following two aspects:

1. Phosphopeptide recognition and improve the binding Stability: This motif orchestrates recognition of the Tau-derived phosphoserine motif via a precise network of hydrogen bonds and electrostatic complementarity. The Lys/Arg residues (seen in blue) form salt bridges with phosphate groups, stabilizing the interaction.
2. The structure shared recognition fold across the 14-3-3 protein family: This motif, with its basic side chain tunnel and surrounding helices, represents a canonical recognition site. Similar structural motifs are observed in all 14-3-3 isozymes when binding phosphoproteins and will serve for post-translational modification signaling.

## I    PERFORMANCE ACROSS PROTEIN FOLD TYPES

**Setup.** We evaluate robustness of GEOBPE by computing distortion as defined in Sec. 3.3 across fold types, unusualness, and size with per-chain metrics. Structural "unusualness" is computed from Foldseek's TM-align mode as $100 \times (1 - \text{TM})$ using the best hit against PDB. We also attach coarse labels: *categories* by best-hit TM-score (**Near-identical** $\geq 0.90$, **Same fold** 0.50–0.89, **Distant**

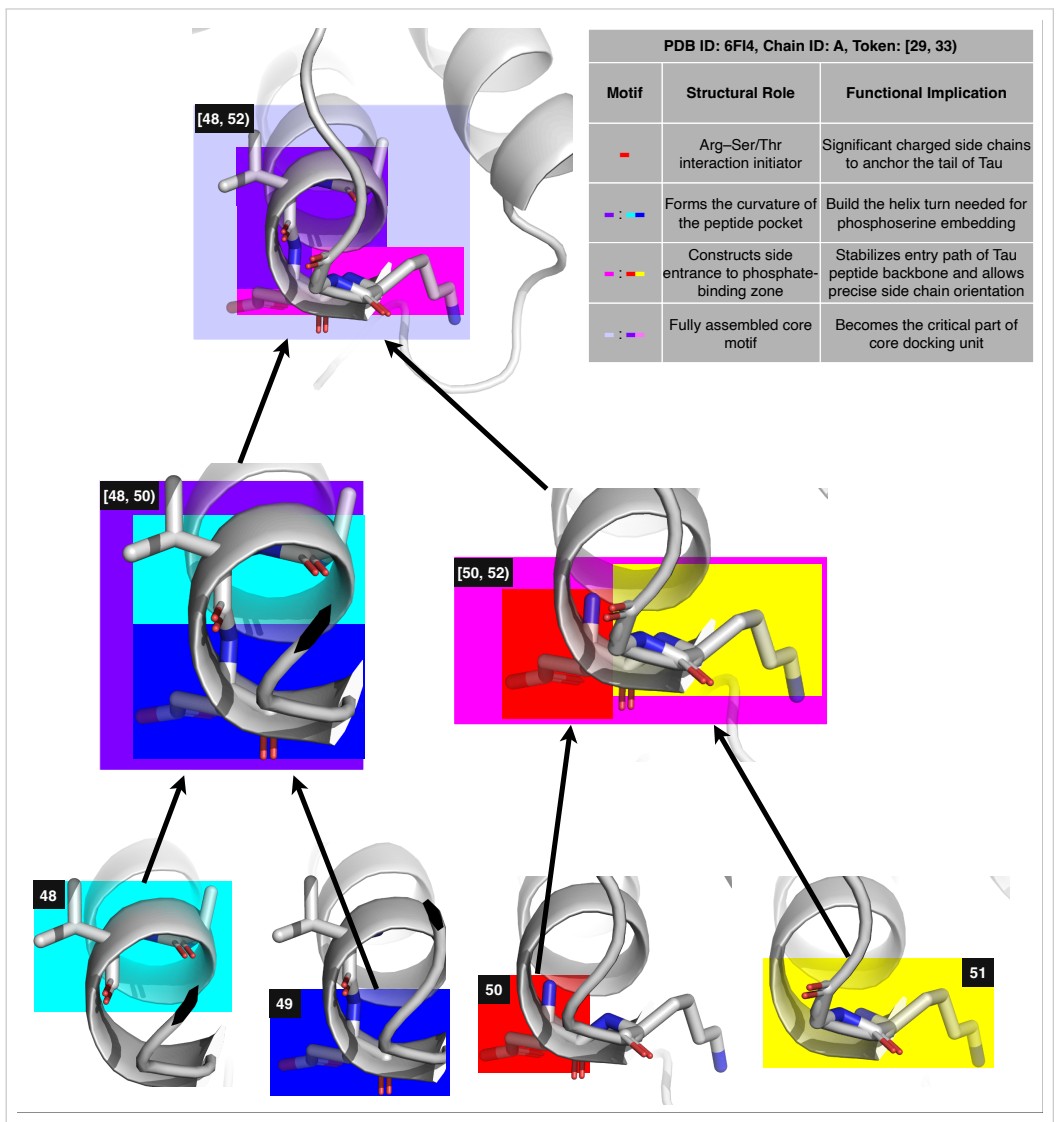

**Figure 10:** Chain A of PDB 6FI4. Hierarchical Merge Tree for Token [29, 33). GEOBPE arrived at this token by merging [29, 30) with [30, 31), [31, 32) with [32, 33), and [29, 31) with [31, 33).

0.30–0.49) and *flags* indicating **very_small** (chains with $< 70$ residues) or **weak_coverage** (low query coverage).

**Fold type labels.** For fold analyses, each chain is annotated with a CATH fold label (topology-level code) and rendered as a human-readable *Class → Architecture* title (multi-line axis label). We report per-fold distributions and means for RMSD and LDDT.

**Plots and statistics reported.** We use four concise views: (i) **Dual-axis scatter vs. unusualness**: RMSD (left, blue) and LDDT (right, orange), with per-bin mean $\pm$ std and a fitted least-squares trend line. (ii) **Group boxplots (shared $x$-axis)**: side-by-side, dual-$y$ box-and-whisker plots compare distributions across {very_small, weak_coverage, Near-identical, Same fold, Distant}; RMSD boxes map to the left axis (blue), LDDT boxes to the right axis (orange). (iii) **Dual-axis scatter vs. length**: RMSD (left) and LDDT (right) versus protein length, again with per-bin mean $\pm$ std and a fitted trend line. (iv) **Fold boxplots**: side-by-side, dual-$y$ box-and-whisker plots per frequent CATH (Class, Architecture) (top-$N$ by support). Together, these views test how specific conditions (very small chains, weak alignment coverage, and decreasing fold similarity)

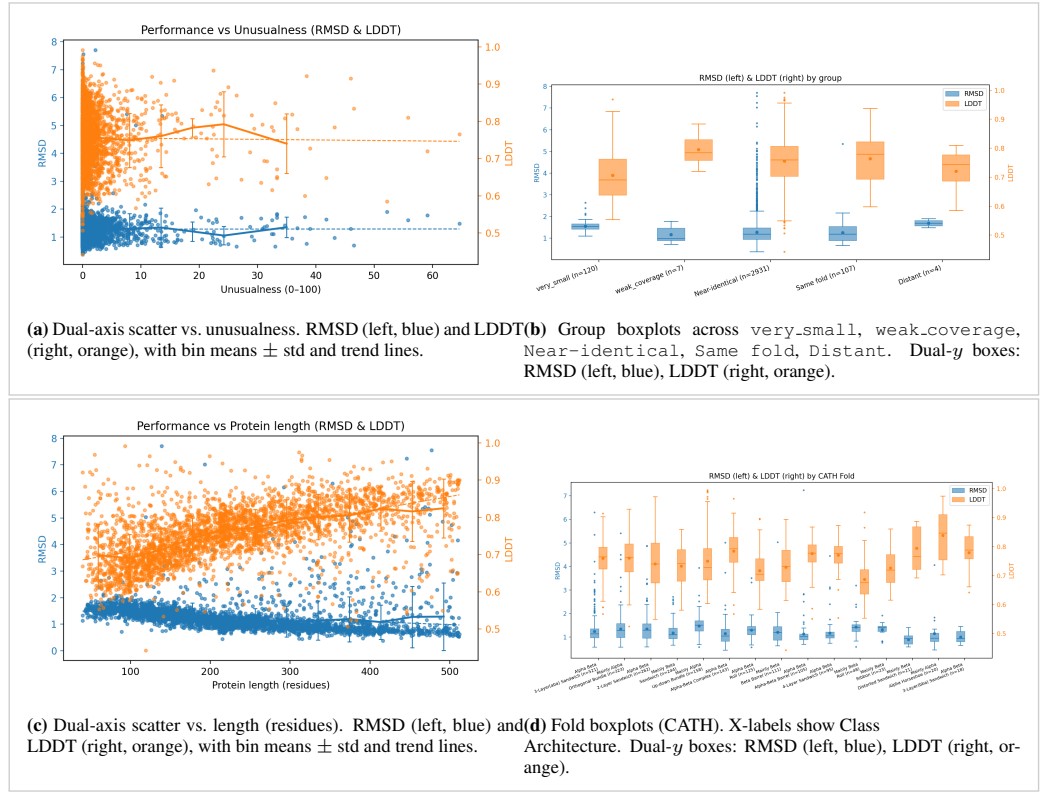

**(a)** Dual-axis scatter vs. unusualness. RMSD (left, blue) and LDDT (right, orange), with bin means ± std and trend lines.

**(b)** Group boxplots across `very_small`, `weak_coverage`, `Near-identical`, `Same fold`, `Distant`. Dual-$y$ boxes: RMSD (left, blue), LDDT (right, orange).

**(c)** Dual-axis scatter vs. length (residues). RMSD (left, blue) and LDDT (right, orange), with bin means ± std and trend lines.

**(d)** Fold boxplots (CATH). X-labels show Class Architecture. Dual-$y$ boxes: RMSD (left, blue), LDDT (right, orange).

**Figure 11:** Plots for GEOBPE robustness evaluation. (A) vs. unusualness, (B) group distributions across flags and categories, (C) vs. length, (D) per-fold distributions. Numerical summaries (Pearson's $r$ with 95% CIs; group/fold means and medians) are in the accompanying CSVs.

modulate accuracy distributions, and whether protein size systematically correlates with errors. We report the following numbers and observations.

- **No degradation on unusual structures** In Fig. 11a, we see *no* correlation (Pearson's $r$ of 0.0091 for RMSD and $-0.0096$ for LDDT, 95% intervals of $(-0.0346, 0.00365)$ and $(-0.0451, 0.0260)$) between Distortion and Unusualness. Consistent with our OOD results in Fig. 4, we see GEOBPE is robust to distributional shifts. As a geometry-grounded tokenizer, GEOBPE captures energetically favorable motifs patterns, which are universal recurrences across all protein families and fold classifications.

- **No degradation on less common folds** In Fig. 11b, we see distortion remain stable on near identical or same fold to those in FoldSeek-DB. In Fig. 11d, there is no visual trend of degradation for less common fold types (left-to-right in Fig. 11d) of the ones shown. Inter-fold discrepancy is also low: among folds with $n \geq 100$, the least faithfully preserved fold type suffers from 30.2% higher RMSD than the most faithful.

- **More faithful to larger folds than smaller folds** In Fig. 11c, we see *weak* correlation (Pearson $r$ of $-0.2141$ for RMSD and $0.5627$ for LDDT, 95% intervals of $(-0.2477, -0.1799)$ and $(0.5379, 0.5865)$) between Distortion and Length. In Fig. 11b, we see distortion slightly elevates for very small folds. Among fold types with at least $n = 100$ samples, GEOBPE achieves lowest distortion (1.130 RMSD) on Alpha-Beta Barrels (cylindrically packed, stable folds) and highest distortion (30% higher RMSD) on Mainly Alpha Up-down Bundles (smaller folds primarily of alternating alpha helices). This suggests GEOBPE has a high propensity for packed but stable folds (sandwiches, barrels).

## J    COMPUTATIONAL COMPLEXITY

**Notation.** Let $\{t^{(\tau)}\}_{\tau=1}^{T}$ be $T$ backbones with lengths $N^{(\tau)}$, and let $N := \sum_{\tau=1}^{T} N^{(\tau)}$ be the total residues. In each STEP iteration, the most frequent geo-pair key has $M_t$ occurrences. We use $K$ for the number of medoids produced when clustering a key's occurrences (a small constant in practice). For k-medoids we either: (i) cluster all $M_t$ items, or (ii) cap with $M_{\max}$ items. Let $P$ be the *period* at which GLUEOPTALL is invoked (see Alg. 9), and let $C_{\mathrm{IK}}$ denote the cost of one global IK pass (see below). The ordered map $\mathcal{D}$ stores key $\rightarrow$ occurrence-set with a priority $(\rho, -|\mathcal{O}|, \kappa)$; each insert/erase in $\mathcal{D}$ costs $O(\log |\mathcal{D}|) = O(\log N)$.

**Component building blocks.**

- **k-medoids on $m$ items:** $O(m^2)$ to build the pairwise RMSD matrix (constant fragment length), plus a small constant number of assignment/update steps
- **Priority map updates:** each merge touches $O(1)$ neighbor pairs; across the *entire* run there are $O(N)$ merges $\Rightarrow O(N \log N)$ total map operations Every merge eliminates one boundary and touches at most its two neighbors, so the total number of insert/erase operations in $\mathcal{D}$ across the full run is $O(N)$; with $O(\log N)$ per op, the total is $O(N \log N)$.
- **Global IK (GLUEOPTALL) one pass:** forward kinematics is linear in links, so one pass costs $C_{\mathrm{IK}} = O(N \cdot S_{\mathrm{FK}})$, where $S_{\mathrm{FK}}$ is the (small) number of optimizer steps $\times$ the constant forward/backward cost per link Periodic GLUEOPTALL adds $\frac{T}{P} O(N \log N)$ due to re-keying affected boundaries.

**Worst-case complexity (no subsampling cap).**

- **ResInitTokens:** $O(N^2) + O(N \log N)$.
- **Step loop over all iterations:** $O\!\left(\sum_t M_t^2\right) + O(N \log N)$.
- **Periodic global glue opt:** $\dfrac{T}{P}\left(C_{\mathrm{IK}} + O(N \log N)\right)$.
- **Total (worst case):** $O(N^2) + O\!\left(\sum_t M_t^2\right) + O(N \log N) + \dfrac{T}{P}\left(C_{\mathrm{IK}} + O(N \log N)\right)$.
- **Total (with cap):** $O(M_{\max}^2) + O(T\,M_{\max}^2) + O(N \log N) + \dfrac{T}{P}\left(C_{\mathrm{IK}} + O(N \log N)\right)$.

In the worst case $M_t = \Theta(N)$ for many steps, $\sum_t M_t^2$ can reach $\Theta(N^2)$. Here $M_{\max}$ controls runtime. Putting it together, we can make the following statements about GEOBPE's computational complexity:

- **(Alg. 1) Training (discovering the vocabulary):** dominated by k-medoids calls and periodic IK:
$$O\!\left(T\,M_{\max}^2\right) + O(N \log N) + \tfrac{T}{P}\left(C_{\mathrm{IK}} + O(N \log N)\right).$$

- **(Alg. 10) Tokenization (apply a learned vocabulary):** similar to training but without any k-medoids calls and in terms of $N^{(\tau)}$:
$$O(N^{(\tau)} \log N^{(\tau)}) + \tfrac{T}{P}\left(C_{\mathrm{IK}} + O(N^{(\tau)} \log N^{(\tau)})\right).$$

- **(Alg. 5) Detokenization (geometry reconstruction):** forward kinematics per link is $O(1)$; reconstructing all atoms is $O(N)$.

**Insights for efficient practice.** (i) Most structural variability concentrates in a small number of modes; a modest $M_{\max}$ suffices. (ii) Dictionary updates are *incremental*; our implementation uses an ordered map. (iii) In practice, we choose $P = 10$; GlueOptAll calls are infrequent enough it does not become an issue. If this becomes the practical bottleneck, we recommend GLUEOPT for local IK updates instead, which drops the $O(N \log N)$ term.

**Distortion is insensitive to $M_{\max}$.** In App. A, we observe that increasing $M_{\max}$ yields no real gains beyond 5000; any marginal gains are lost to the subsequent GlueOptAll call. This is because medoids stabilize quickly on representative modes, capping clustering with $M_{\max}$ preserves reconstruction quality while bounding the dominant $O(M_{\max}^2)$ term. This is backed by observations made by de Brevern et al. (2002); Mackenzie (2016) and others that the structural universe of possible elements are captured by a exponentially smaller number of modes.

**Increasing GLUE_OPT_EVERY does not significantly hurt performance.** In Figs. 13b & 13a, we see GEOBPE behavior remains comparable between a run where the expensive glue optimization

(all) is done every iteration, vs a run following our default recommendation of every ten iterations. The $\frac{T}{P}O(N \log N)$ term is often the key walltime bottleneck, as Table 13 shows. Therefore, increasing $P$ would help amortize the expensive rigid body refinement routine across iterations.

**Wall times from our experiments.** In Table 13, we report empirical wall times from a sample run of GEOBPE following default settings. The runtime can be accelerated with more CPUs.

**Table 13:** Using 20 CPUs, we report our job's wall-clock time. Underlined steps perform periodic glue optimization (period $P = 10$). They are followed by $P - 1$ GEOBPE steps. We report wall times for steps 0, 10, 20, 200; omitted steps interpolate predictably.

| Function | Paper Reference (Algo, Line) | Time (HH:MM:SS) |
|---|---|---|
| _init_thresholds | Algo 1 L1(Empirical Quantizer Estimation) | 00:01:33 |
| _init_res_tokens | Algo 1 L2 (Per-residue Initialization) | 02:16:02 |
| glue_opt_all | Algo 1 L3 (Global glue refinement) | 03:21:50 |
| Step 0 | Algo 1 L7 (Step) w/ Algo 9 L13 (glue opt all) | 02:36:21 |
| Steps 1-9 | Algo 1 L7 (Step) | 01:32:50 |
| Step 10 | Algo 1 L7 (Step) w/ Algo 9 L13 (glue opt all) | 01:40:58 |
| Steps 11-19 | Algo 1 L7 (Step) | 01:37:39 |
| Step 20 | Algo 1 L7 (Step) w/ Algo 9 L13 (glue opt all) | 01:27:39 |
| Steps 21-29 | Algo 1 L7 (Step) | 01:36:41 |
| | ... | |
| Step 200 | Algo 1 L7 (Step) w/ Algo 9 L13 (glue opt all) | 00:56:56 |
| Steps 201-209 | Algo 1 L7 (Step) | 00:50:06 |
| | ... | |

# K  HYPERPARAMETER DOCUMENTATION AND GUIDELINES

## K.1  MAIN HYPERPARAMETERS AND REPRODUCIBLE SETTINGS

We describe the key parameters that govern GEOBPE's behaviors in Table 14. For each, we report the default setting used by GeoBPE across most key results of the paper: Fig. 4, Tables 10 & 11 and App. A. We report any instances overriding the default settings here:

1. Token efficiency / SSLM-Eval (Tables 10, 11) set num_p ← {2:500,3:2000}, bins ← {1:1000} for codebook size $|\mathcal{V}| = 2500$ and num_p ← {2:1000,3:5000}, bins ← {1:2000} for $|\mathcal{V}| = 6000$.
2. Pareto-efficiency evaluation (Fig 4) further add the setting for $|\mathcal{V}| = 21000$ where num_p ← {2:1000,3:20000}, bins ← {1:2000}. We vary num_p elastically moves along the Pareto-efficiency plot, trading off BPR for distortion. All runs use $w_t = 1.0$, which we discover from ablation studies (see Tables 17 & 18) has better performance than $w_t = 0.1$.
3. Downstream transfer experiments (Tables 1, 7) set num_p ← {2:2,3:5,5:1,6:2,8:1}, free_bonds ← False and bins ← {1 : 50}, and bin_strategy ← histogram-cover to adaptively coarsen the resolution. GEOBPE prioritizes learning fine-to-coarse hierarchical signals over low distortion for effective transfer.

**Table 14:** We report the main hyperparameters that affect GEOBPE behavior.

| Parameter | Value | Meaning | Default Behavior |
|---|---|---|---|
| bin_strategy | histogram | Controls the strategy for empirical quantizer estimation (Alg. 1) | numpy.histogram with bins |
| bins | {1:500} | Controls the number of bins used by bin_strategy | Uses 500 quantiles |
| free_bonds | True | Whether to quantize bond lengths | Don't standardize |
| | | Setting to False standardizes all bond lengths to precomputed values | Quantize with linear histograms |
| glue_opt | True | Whether to do Glue Opt in Algs 18, 9 | Do Glue Opt |
| glue_opt_every | 10 | How often to run global glue opt (final line of Alg. 9) | Do every 10 iters |
| glue_opt_method | all | Whether to do batch glue opt (Alg. 12) or single-boundary glue opt (Alg. 19) | Do batch glue opt |
| glue_opt_prior | 1.0 | Prior weight encouraging optimized glues to match empirical distribution | 1.0 |
| w_R, w_t | 1.0, 0.1 | Rotation, translation loss term weights to IK loss (Alg. 19) | Weigh rotation error 10x translation error |
| max_num_strucs | 5000 | Max number of occurrences for clustering ($M_{\max}$ in Alg. 6) | 5000 |
| num_p | {2:100,3:500, 5:20,6:100} | $K$ determined by span length $L$ in Alg. 6, $L$ not in num_p round down to nearest key | Use $K = 100$ when $L = 2$ |
| | | | Use $K = 500$ when $L \geq 3$ |
| rmsd_super_res | True | Whether to use occurrences from *original* backbone $t_\tau$ or current backbone in Alg. 9 | Use original states |

K.2 HYPERPARAMETER SELECTION GUIDELINES

**Which ones to prioritize.** Only a few parameters in Table 14 dictate overall behavior, performance and runtime. Essential knobs are:

- **Vocabulary Growth**
    - `num_p` (number of medoids)
    - `bins` (quantizer strength)
    - `max_iter`(or # iterations to run)
- **Compression/Runtime Tradeoff**
    - `glue_opt`/`glue_opt_method`/`glue_opt_every` (glue optimization)

Aside from these, we suggest leaving the rest to default values.

**Choosing `num_p` (medoids per step) and `bins` (angle/length quantization strength).**

We define `num_p` via a step-wise schedule over motif sizes. For example,

$$\{2 : 2, \ 3 : 5, \ 5 : 10\}$$

(passed as **`--num-p 2-2:3-5:5-10`**) means:

- introduce 2 tokens for geometric keys with 2 bonds (C-terminal residue orientations),
- 5 tokens for keys with 3 bonds (all non-terminal residues),
- 10 tokens for all merged geometric keys with 5 or more bonds (every GeoBPE step after residue initialization).

The `bins` parameter uses the same syntax as `num_p`. For example,

$$\{1 : 100, \ 3 : 10\}$$

(passed as **`--bins 1-100:3-10`**) introduces 100 bins to discretize the angular histogram at initialization, with 10 bins for keys of size $\geq 3$. For brevity, BINS $= n$ is shorthand for BINS $= \{1 : n\}$.

The binning strategy is controlled by `bin-strategy`. If glue optimization produces angles outside the supported range, we snap them to the closest bin. In practice, we recommend increasing or decreasing `bins` in tandem with `num_p`.

Below we give practical recommendations by downstream use case.

**GeoBPE for compression / reconstruction.**

*Intuition.* Larger `num_p` values $\to$ more medoids per step, which improves reconstruction quality (RMSD/LDDT) at the cost of a larger vocabulary and noisier merges. Empirically, we observe diminishing returns in reconstruction beyond settings such as

$$\text{num\_p} = \{2 : 200, \ 3 : 1000\},$$

consistent with there being only a limited number of modes in the conformational variability of energetically favored backbone regions (Ramachandran landscape).

*Recommendation.* Use relatively large `num_p` to maximize reconstruction fidelity, but pair it with a high (yet not extreme) `bins[size]` to avoid a combinatorial explosion in the space of geometric keys. A good default for reconstruction-oriented use is

$$\text{bins} = \{1 : 500\},$$

combined with moderately large `num_p`.

**GeoBPE for representation learning.**

*Intuition.* GeoBPE emits both a sequence of tokens and a merge hierarchy, with the hierarchy providing the main inductive bias for downstream representation learning from residue to protein level. A useful hierarchy should:

- capture higher-level patterns (from basic secondary structure elements to functional sites),
- avoid overfitting to high-frequency local vibrations.

Here the goal shifts from pure compression to *coarsening*: we want motifs that aggregate meaningful local structure without being overly fine-grained.

*Recommendation.* Use relatively small `num_p` values and correspondingly small `bins`. For example, the configuration used in our paper for representation learning was

$$\texttt{num\_p} = \{2 : 2,\ 3 : 5,\ 5 : 1,\ 6 : 2,\ 8 : 1\},$$

paired with

$$\texttt{bins} = \{1 : 50\}$$

and the `histogram-cover` strategy. This yields coarser motifs and hierarchies that are better suited to downstream predictive tasks.

**Choosing the number of merge iterations.**

At iteration $t$, GeoBPE increases the vocabulary size by looking up `num_p`:

$$|\mathcal{V}_{\text{final}}| \approx |\mathcal{V}_{\text{init}}| + \sum_{t=1}^{T} \texttt{num\_p}[|\texttt{key}^{(t)}|],$$

where $T$ is the number of merge iterations. More iterations yield a larger and more varied vocabulary, but each "word" (motif) is then used less frequently.

The optimal stopping point depends on the downstream application.

**GeoBPE for representation learning.**

For representation learning, the merge hierarchy serves as an inductive bias: merged token pairs tend to correspond to secondary structure segments and align with domain or homology hits. Here GeoBPE should *coarsen* high-resolution details into higher-level motifs instead of growing an extremely large vocabulary.

*Recommendation.* Use a moderate number of iterations, stopping once downstream validation metrics (e.g., AUROC, Spearman $\rho$, Macro-F1) plateau. In practice, this typically occurs well before exhausting all possible merges; beyond that point, additional iterations mainly create very specific, low-usage motifs that add complexity without improving downstream performance.

**GeoBPE for compression / reconstruction.**

GeoBPE is closest to its BPE origins when used as a compression algorithm: the goal is to reduce sequence length (increase compression), preserving geometry (minimize distortion), while monitoring the amortized bits to store the growing vocabulary in the background.

*Recommendation.* Allow fewer iterations and monitor the trade-off between bit-rate (i.e., BPR) and reconstruction error (RMSD/LDDT). A general rule-of-thumb is to continue merging as long as additional iterations lowers *either* BPR or distortion. Later, one can choose the right iteration checkpoint to navigate the tradeoff. Thus, it is wise to stop immediately when both metrics begin degrading simultaneously. On very high resolutions and a moderate dataset (e.g. our pretraining dataset), this happens early on. The amortized bits used to grow the vocabulary generally outpace the bits saved from decreasing the number of tokens per structure; this relationship reverses on lower resolutions and larger datasets.

**GeoBPE for language modeling.**

When using GeoBPE as a tokenizer for protein language models, a common heuristic adapted from NLP is to set the final vocabulary size such that

$$\frac{|\mathcal{V}|}{L} \approx \frac{N}{1000},$$

where $L$ is the average number of motifs per structure and $N$ is the number of structures. Equivalently, the total number of tokens

$$T \approx L \times N$$

suggests a target vocabulary size $|\mathcal{V}| \approx T/1000$.

Table 15 shows concrete numbers for different LM scales.

**Table 15:** Heuristic vocabulary sizes $|\mathcal{V}|$ for GeoBPE when used as a tokenizer for protein language models at different data and model scales.

| LM scale | # structures $N$ | $L$ (motifs / structure) | $|\mathcal{V}| \approx (L \times N)/1000$ | Model params |
|---|---|---|---|---|
| Toy / demo | $10^3$ | 100 | $\sim 10^2$ | $\sim 10^6$ |
| Small / usable | $10^4$ | 100 | $\sim 10^3$ | 10–50M |
| Base "GPT-small" | $10^5$ | 100 | $\sim 10^4$ | $\sim 10^8$ |
| Mid-scale | $10^6$ | 100 | $\sim 10^5$ | $\gtrsim 10^9$ |

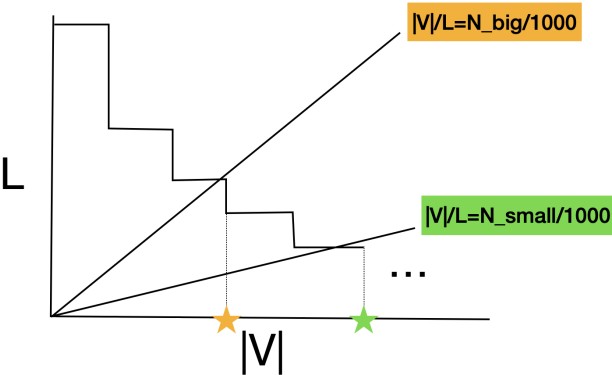

**Figure 12:** Illustration of the heuristic for choosing the number of merge iterations based on the target vocabulary size $|\mathcal{V}|$ for language modeling. The marked point indicates the recommended stopping iteration for a dataset with $N$ structures.

We implement a stopping criterion based on this heuristic: during training we track $|\mathcal{V}|$ as merges accumulate ($T$ decreases) and mark the iteration where the target $|\mathcal{V}|$ meets $T/1000$. In Fig. 12, this iteration is highlighted (e.g., with a star) and concretely in the `run_{iter}.png` plots produced in each run directory by our code.

**Practical tip.** In practice, you can set a relatively large `max_iter` and let GeoBPE proceed for many iterations while logging checkpoints. After training, select the checkpoint whose vocabulary size and downstream metrics best match your target (compression, representation quality, or LM tokenizer size), rather than trying to tune the exact stopping iteration a priori.

### K.3 SENSITIVITY STUDIES

We show how sensitive GEOBPE behavior is to key hyperparameters by running ablation experiments for selected hyperparameters, one at a time.

**$|V|$ (NUM_P, # iterations) varies.**

In Table 16 is an ablation study varying NUM_P across runs and # iterations per run; $|V|$ depends on both. We also combine both into the throttle $|V|$. We make the following observations:

1. As NUM_P values become high, the unique GeoPair keys increase exponentially. Since each iteration only looks at one key, the number of merges done falls off. Empirically, the top rows show only marginal changes to distortion as iteration increases. We omitted them for brevity.
2. As NUM_P values drop too low, GEOBPE becomes more of a coarsening algorithm (lots of merges, repetitive patterns are preserved but higher frequencies are lost). When merges happen more often, more drift is introduced, so error quickly accumulates. We can see distortion monotonically increase for the last run.
3. There exists a tension between NUM_P and merge frequency, but glue opt is still potent enough to manage drift accumulation. We see error decrease before increasing again, when eventually merges overwhelm.

**BINS** $\in \{50, 100, 300\}$**.** BINS controls the quantization of bond lengths, bond angles and torison angles connecting motifs; it trades off structural fidelity for better coarsening. Geo-Pair keys are

**Table 16:** Cluster of runs that vary NUM_P; each row is a run; lower resolution runs include periodic checkpoints to see how RMSD/LDDT/BPR changes over iterations.

| NUM_P | $|V|$ | RMSD | LDDT | BPR |
|---|---|---|---|---|
| {2: 100, 3: 500, 5: 20, 6: 100, ...} | 600 | 1.66 | 0.73 | 36.02 |
| {2: 500, 3: 2000, 5: 100, 6: 500, ...} | 2500 | 1.41 | 0.75 | 41.11 |
| {2: 1000, 3: 5000, 5: 200, 6: 1000, ...} | 6000 | 1.37 | 0.76 | 45.44 |
| {2: 1000, 3: 20000} | 21000 | 1.21 | 0.76 | 47.62 |
| {2: 50, 5: 20, 6: 100, ...} | 5200 | 2.11 | 0.68 | 37.24 |
| {2: 10, 3: 50, 5: 1, 6: 5, 8: 1} | 65 | 1.78 | 0.71 | 30.81 |
|  | 237 | 1.77 | 0.70 | 34.00 |
|  | 388 | 1.72 | 0.70 | 35.88 |
|  | 521 | 1.71 | 0.70 | 37.33 |
|  | 631 | 1.69 | 0.70 | 38.47 |
|  | 739 | 1.71 | 0.70 | 39.54 |
|  | 845 | 1.73 | 0.69 | 40.57 |
| {2: 2, 3: 5, 5: 1, 6: 2, 8: 1} | 109 | 3.96 | 0.53 | 27.26 |
|  | 309 | 4.07 | 0.53 | 30.81 |
|  | 508 | 4.23 | 0.53 | 33.46 |
|  | 707 | 4.50 | 0.53 | 35.93 |

of the form $(\mathcal{M}_{p:q}, \Gamma_q, \mathcal{M}_{q:r})$, and the space of $\gamma_q$ has size $\sim$ (BINS)$^3$ (since there are 3 glue angles). Importantly, it is orthogonal to NUM_P, which control the id's of $\mathcal{M}_{p:q}$ and $\mathcal{M}_{q:r}$, so it can be tuned independently. Fixing Hyperparameter Setting 3, we *increase* the number of bins used to discretize $\theta^{CNCA}, \omega, \phi$ angles angles by 2x, 6x. Fig. 13e uses the default value BINS = 50; Figs. 13a & 13d use BINS = 100, 300. Increasing BINS decreases frequency of merges by around the same factor, so we observe $L$ vs $K$ is flatter for higher BINS settings $L$ decreases slower. Since BINS is an important control of resolution, decreasing it increases distortion (e.g. 3.12 $\to$ 4.19 $\to$ 5.76 RMSD). Distortion is not a priority consideration for transfer experiments. Since the goal is to compress local noise into meaningful global hierarchies, introducing distortion is *necessary* to cluster common motifs. Setting BINS too low can *misrepresent* the overall structure, so we recommend BINS = 50 as a good starting value.

**GLUE_OPT_EVERY** $\in \{1, 10\}$. Fig. 13b (GLUE_OPT_EVERY = 1) only shows a 6.2% decrease in RMSD and comparable LDDT vs Fig. 13a (GLUE_OPT_EVERY = 10). As the wall times in 13 shows, decreasing the frequency of glue_opt significantly accelerates GEOBPE, regardless of how many cores are available. App. J reveals glue_opt period $P$ to directly dictate a rate-limiting term in GEOBPE's complexity. Thus, we adopt GLUE_OPT_EVERY = 10 as the default setting. We also suggest GEOBPE users to try setting GLUE_OPT_EVERY > 10 to balance the tradeoffs.

**Table 17:** We performed the following sweep over $(w_R, w_t)$ (order of magnitude changes to $w_T / w_R$); remaining settings match defaults (App. K).

| $(w_R, w_t)$ | Train | | CAMEO | | CASP14 | |
|---|---|---|---|---|---|---|
| GeoBPE (1%) | RMSD | LDDT | RMSD | LDDT | RMSD | LDDT |
| $(10, 0.1)$ | 2.846 | 0.615 | 2.767 | 0.601 | 2.608 | 0.587 |
| $(1, 0.1)$-default | 1.718 | 0.739 | 1.656 | 0.734 | 1.526 | 0.721 |
| $(1.0, 1.0)$ | 1.552 | 0.764 | 1.546 | 0.755 | 1.412 | 0.743 |
| $(0.1, 1.0)$ | 1.537 | 0.767 | 1.532 | 0.758 | 1.396 | 0.745 |
| $(0.1, 10)$ | 1.533 | 0.768 | 1.533 | 0.758 | 1.407 | 0.745 |

**W_R/W_T** $\in \{10^{-2}, \ldots, 10^2\}$. Table 17 shows $w_t$ is relatively more important than $w_R$ for reconstruction. The interpretation is correct positions are more critical than correct orientations. We observe diminishing returns once $w_t \geq w_R$ ($|\Delta_{LDDT}| \approx 10^{-3}, |\Delta_{RMSD}| \approx 10^{-2}$).

**Table 18:** We compare the default $(w_R, w_t)$ setting with $(1.0, 1.0)$, which in Table 17 resulted in lower distortion for GEOBPE (1%).

| GeoBPE | | Train | | Valid | | CAMEO | | CASP14 | |
|---|---|---|---|---|---|---|---|---|---|
| $|V|$ | $(w_R, w_t)$ | RMSD | LDDT | RMSD | LDDT | RMSD | LDDT | RMSD | LDDT |
| 600 | $(1, 0.1)$-default | 1.66 | 0.73 | 1.71 | 0.72 | 1.77 | 0.72 | 1.53 | 0.72 |
| | $(1.0, 1.0)$ | 1.55 | 0.75 | 1.58 | 0.74 | 1.65 | 0.74 | 1.39 | 0.74 |
| 2500 | $(1, 0.1)$-default | 1.41 | 0.75 | 1.50 | 0.74 | 1.57 | 0.74 | 1.51 | 0.73 |
| | $(1.0, 1.0)$ | 1.29 | 0.78 | 1.36 | 0.77 | 1.41 | 0.77 | 1.33 | 0.76 |
| 6000 | $(1, 0.1)$-default | 1.37 | 0.76 | 1.46 | 0.75 | 1.52 | 0.74 | 1.54 | 0.72 |
| | $(1.0, 1.0)$ | 1.23 | 0.78 | 1.30 | 0.78 | 1.37 | 0.77 | 1.35 | 0.75 |
| 21000 | $(1, 0.1)$-default | 1.21 | 0.77 | 1.28 | 0.76 | 1.40 | 0.75 | 1.55 | 0.72 |
| | $(1.0, 1.0)$ | 1.05 | 0.80 | 1.12 | 0.79 | 1.25 | 0.78 | 1.35 | 0.76 |

**FREE_BONDS** $\in$ {**False**, **True**}. FREE_BONDS decides whether bond lengths are free variables, or standardized to fixed values. Generally, the backbone bond lengths are very close to fixed and most workflows (e.g. X-ray diffraction, NMR, Cryo-EM) that solve structures make such assumptions. GEOBPE is designed to be fully general, allowing variable bond lengths. In lieu of the known fact that they have relatively narrow ranges, we ran a sanity check to see if GEOBPE is sensitive to FREE_BONDS. Comparing Figs. 13c & 13e, we see the run with free bonds achieves only $1.69\%$ lower RMSD, which is negligible.

## L  LARGE LANGUAGE MODEL USAGE

We used LLMs mainly for polishing the writing, including prompts to check for grammar mistakes, improving clarity of mathematical notation, and formatting the text to save space.

## M  ALGORITHMIC DETAILS

**Additional notation for algorithms.**

We reuse all geometric and GeoBPE notation from Secs. 3.1–3.2. For convenience we collect the additional symbols that appear only inside the algorithmic pseudocode.

$\mathcal{S}, \mathcal{A}$  Set of motif (or motif–pair) occurrences and its sampled subset used by RMSD_PARTITION (Alg. 6); each $u \in \mathcal{S}$ indexes a motif $\mathcal{M}_{i_u:k_u}^{(t_u)}$.

$\mathcal{S}_3, \mathcal{S}_2$  Collections of interior and terminal bond–residue occurrences used to build residue-level codebooks $\mathcal{A}_3, \mathcal{A}_2$ (Algo. 18).

$\widehat{\mathcal{M}}, c(\cdot)$  Medoid set and assignment map returned by RMSD_PARTITION, used as inputs to glue-optimization routines (Algos. 19,12).

$\mathcal{D}^{(\star)}, \mathcal{O}^{(\star)}(\kappa)$  Single-backbone geo-pair map and occurrence sets for a new backbone $t^\star$ during tokenization (Algo. 10).

$\Sigma, \Sigma_{\mathrm{med}}$  Token dictionary used for geometric language modeling (Algos. 16, 17); $\Sigma_{\mathrm{med}}$ contains only motif tokens; $\Sigma$ also includes glue angle tokens.

$\mathrm{id}_{\mathrm{med}}, \mathrm{id}_{\mathrm{bin}}$  Integer maps assigning token IDs to motif medoids $(\kappa, j)$ and to glue-angle bins $(\mathrm{type}, b)$, respectively.

$x^{(\tau)}$  Token sequence encoding backbone $t_\tau$ obtained by alternating motif and glue-bin tokens (Algo. 17).

$h_i^\uparrow, c_i^\uparrow, \bar{h}_i, \bar{c}_i$  Upward and downward TreeLSTM states at node $i$ in the up–down encoder (Algos. 13–15).

$z_{\tau,i}^{\mathrm{res}}, z_\tau^{\mathrm{prot}}$  Final residue-level and protein-level embeddings produced by the up–down encoder on the merge hierarchy $\mathcal{F}^{(\tau)}$ (Algo. 15).

---

**Algorithm 6** RMSD_PARTITION on motif–pair occurrences

---

**Require:** Motif–pair occurrences $\mathcal{S} = \{u = 1, \ldots, M\}$ with $\mathcal{M}_{i_u:k_u}^{(t_u)}$, common span length $L = k_u - i_u + 1$, and either $\forall u, k_u = N^{(t_u)}$ or $\forall u, k_u < N^{(t_u)}$; target $K \geq 1$; optional $M_{\max}, T, \varepsilon$.

**Ensure:** Medoids $\widehat{\mathcal{M}} = \{\widehat{m}_1, \ldots, \widehat{m}_K\} \subseteq \mathcal{S}$ and assignments $c : \{1, \ldots, M\} \to \{1, \ldots, K\}$.

1: For each $u \in \mathcal{S}$, compute $\mathbf{X}_u \in \mathbb{R}^{3L \times 3}$ via COMPUTE_COORDS$(i_u:k_u)$.
2: Let $\mathcal{A} \subseteq \mathcal{S}$ be a uniform sample without replacement of size $\min(M, M_{\max})$ (or $\mathcal{A} = \mathcal{S}$).
3: Build $D \in \mathbb{R}^{|\mathcal{A}| \times |\mathcal{A}|}$ with $D_{uv} = $ KABSCH_RMSD$(\mathbf{X}_u, \mathbf{X}_v)$ for $u, v \in \mathcal{A}$.
4: Initialize $\mathcal{M} \leftarrow \{m_1, \ldots, m_K\}$ as $K$ distinct uniform indices from $\{1, \ldots, |\mathcal{A}|\}$.
5: **for** $t = 1$ **to** $T$ **do**
6:   Assign: $c(u) \leftarrow \arg\min_{j \in \{1, \ldots, K\}} D_{u, m_j}$ for all $u \in \mathcal{A}$.
7:   Update each $j$: $\mathcal{C}_j = \{u \in \mathcal{A} : c(u) = j\}$. If $\mathcal{C}_j = \emptyset$, reseed $m_j$ uniformly from $\mathcal{A}$; else

$$m'_j \leftarrow \arg\min_{u \in \mathcal{C}_j} \sum_{v \in \mathcal{C}_j} D_{uv}.$$

8:   If $\sum_{j=1}^{K} D_{m_j, m'_j} < \varepsilon$ **break**; else set $m_j \leftarrow m'_j$ for all $j$.
9: **end for**
10: Map $\mathcal{M} = \{m_1, \ldots, m_K\}$ (indices in $\mathcal{A}$) to $\widehat{\mathcal{M}} = \{\widehat{m}_1, \ldots, \widehat{m}_K\}$ (indices in $\mathcal{S}$).
11: For each $u \in \mathcal{S}$, set $c(u) \leftarrow \arg\min_{j \in \{1, \ldots, K\}}$ KABSCH_RMSD$(\mathbf{X}_u, \mathbf{X}_{\widehat{m}_j})$.
12: **return** $\widehat{\mathcal{M}}$ and $c(\cdot)$.

---

**Algorithm 7** KABSCH_RMSD$(\mathbf{P}, \mathbf{Q})$

---

**Require:** $\mathbf{P}, \mathbf{Q} \in \mathbb{R}^{n \times 3}$ with $n = 3L$.

1: $\bar{\mathbf{p}} = \frac{1}{n} \sum_i \mathbf{P}_i, \bar{\mathbf{q}} = \frac{1}{n} \sum_i \mathbf{Q}_i$
2: $\tilde{\mathbf{P}} = \mathbf{P} - \bar{\mathbf{p}}, \quad \tilde{\mathbf{Q}} = \mathbf{Q} - \bar{\mathbf{q}}$
3: $\mathbf{H} = \tilde{\mathbf{P}}^\top \tilde{\mathbf{Q}}, \quad \mathbf{U}\Sigma\mathbf{V}^\top = $ SVD$(\mathbf{H})$
4: $\mathbf{R} = \mathbf{U}\mathbf{V}^\top$; if $\det(\mathbf{R}) < 0$, set $\mathbf{V}_{:,3} \leftarrow -\mathbf{V}_{:,3}$ and recompute $\mathbf{R} = \mathbf{U}\mathbf{V}^\top$
5: $\mathbf{Q}_{\text{aligned}} = (\mathbf{Q} - \bar{\mathbf{q}})\mathbf{R}^\top + \bar{\mathbf{p}}$
6: **return** $\sqrt{\frac{1}{n} \sum_{i=1}^{n} \|\mathbf{P}_i - \mathbf{Q}_{\text{aligned},i}\|^2}$

---

**Algorithm 8** K_MEDOIDS on a precomputed distance matrix

---

**Require:** Symmetric $D \in \mathbb{R}^{N \times N}$, number of clusters $K$, iterations $T$, tolerance $\varepsilon$.

**Ensure:** Medoid set $\{m_1, \ldots, m_K\}$ and assignments $c(\cdot)$ on $\{1, \ldots, N\}$.

1: Initialize medoids $\{m_j\}$ as $K$ distinct random indices.
2: **for** $t = 1$ **to** $T$ **do**
3:   $c(u) \leftarrow \arg\min_j D_{u, m_j}$ for all $u$
4:   **for** $j = 1$ **to** $K$ **do**
5:     $\mathcal{C}_j = \{u : c(u) = j\}$; if $\mathcal{C}_j = \emptyset$, re-seed $m_j$ at random
6:     $m'_j \leftarrow \arg\min_{u \in \mathcal{C}_j} \sum_{v \in \mathcal{C}_j} D_{uv}$
7:   **end for**
8:   If $\sum_{j=1}^{K} D_{m_j, m'_j} < \varepsilon$, **break**; else $m_j \leftarrow m'_j$ for all $j$
9: **end for**
10: **return** $\{m_j\}$ and $c(\cdot)$

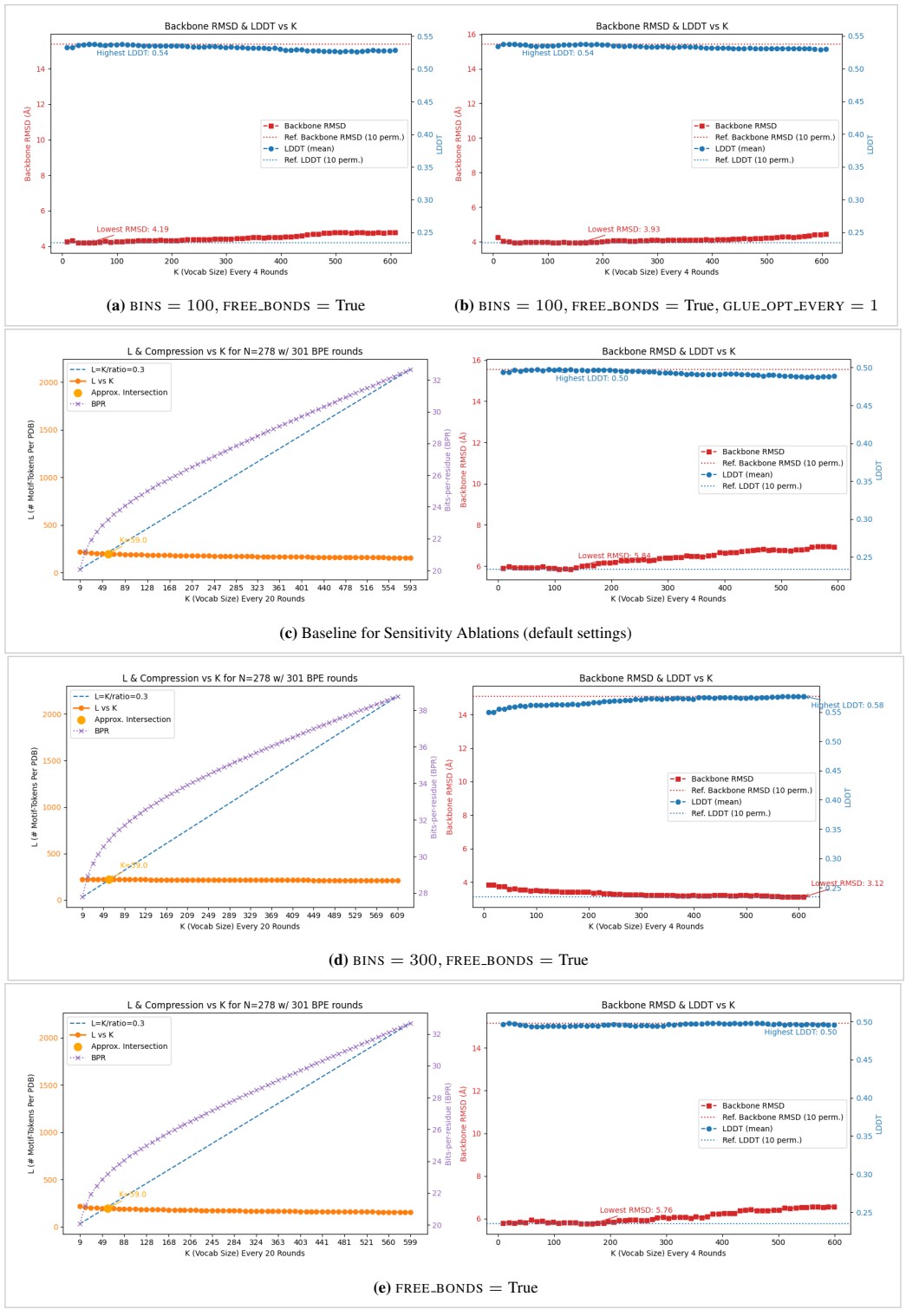

**Figure 13:** Sensitivity analysis on BINS ← {50 (default), 100, 300}), GLUE_OPT_EVERY ← {1, 10 (default)} and FREE_BONDS ← {True, False (default)}. Rest of hyperparameters match defaults (Settings 3). We show GEOBPE (1%) progress plots at ITER = 300 for all ablation settings. $L$ is # avg. tokens per structure; $K$ is $|V|$; see K.2 for details.

---

**Algorithm 9** STEP — one GEOBPE merge iteration

---

**Require:** Current segmentations $\{\mathcal{P}^{(\tau)}\}_{\tau=1}^T$ and merge hierarchies $\{\mathcal{F}^{(\tau)}\}_{\tau=1}^T$ (frontier leaves of $\mathcal{F}^{(\tau)}$ equal $\mathcal{P}^{(\tau)}$); priority-ordered map $\mathcal{D}$ with keys $\pi(\kappa) = (\rho(\kappa), -|\mathcal{O}(\kappa)|, \kappa)$ and values $\mathcal{O}(\kappa)$; current vocabulary $\mathcal{V}$ (map: key $\rightarrow$ prototype set); boundary-glue quantizers $Q_{\theta^{CNCA}}, Q_\omega, Q_\phi$; optional glue mode $\in \{none, each, all\}$ and, if $all$, a period.

**Ensure:** Updated $(\{\mathcal{P}^{(\tau)}\}, \{\mathcal{F}^{(\tau)}\}, \mathcal{D}, \mathcal{V})$.

1: **Select the merge key.**
$$\big((\rho^\star, -c^\star, \kappa^\star),\ \mathcal{O}(\kappa^\star)\big) \leftarrow \text{FRONT}(\mathcal{D}).$$

   Write each occurrence as $(\mathcal{L}, \mathcal{R}) \in \mathcal{O}(\kappa^\star)$ with $\mathcal{L} = \mathcal{M}_{p:q}^{(t_\tau)}$ and $\mathcal{R} = \mathcal{M}_{q+1:r}^{(t_\tau)}$.

2: **Prototype assignment (create-or-assign).**

3: **if** $\rho^\star = 1$ (*no prototypes yet*) **then**

4:    Gather concatenated spans $\{\mathcal{M}_{p:r}^{(t_\tau)}\}$ from *original* $t_\tau$ for all $(\mathcal{L}, \mathcal{R}) \in \mathcal{O}(\kappa^\star)$ (identical length).

5:    Run RMSD_PARTITION (Alg. 6) to obtain medoids and $c : \mathcal{O}(\kappa^\star) \rightarrow \{1, \ldots, K_{|\kappa^\star|}\}$.

6:    Define $\mathcal{A}_{\kappa^\star} = \{\Pi_j^{(\kappa^\star)}\}_{j=1}^{K_{|\kappa^\star|}}$ (medoid spans' internal-parameter tuples).

7:    Update vocabulary: $\mathcal{V}[\kappa^\star] \leftarrow \mathcal{A}_{\kappa^\star}$ and set $\rho^\star \leftarrow 0$.

8: **else**

9:    For each occurrence, set $c(\mathcal{L}, \mathcal{R}) = \arg\min_j \text{RMSD}\big(\mathcal{M}_{p:r}^{(t_\tau)}, \Pi_j^{(\kappa^\star)}\big)$ using $\mathcal{V}[\kappa^\star]$.

10: **end if**

11: **Greedy, non-overlapping merges (and hierarchy updates).** For each backbone $t_\tau$, sort occurrences by $p$ and choose a maximal disjoint subset $S^{(\tau)}$ left-to-right. For every $(\mathcal{L}, \mathcal{R}) \in S^{(\tau)}$ with label $j = c(\mathcal{L}, \mathcal{R})$:

   1. **Form merged motif** $\widetilde{\mathcal{M}} = \mathcal{M}_{p:r}^{(t_\tau)}$ and *overwrite* its internals by the prototype:
   $$\big(\ell, \theta, \psi, \omega, \phi, \{\Gamma_i\}\big)\big|_{\widetilde{\mathcal{M}}} \leftarrow \Pi_j^{(\kappa^\star)}.$$

   2. **Update segmentation** $\mathcal{P}^{(\tau)}$: replace $(\mathcal{L}, \mathcal{R})$ by $\widetilde{\mathcal{M}}$.

   3. **Update hierarchy** $\mathcal{F}^{(\tau)}$: add a *parent* node for span $[p{:}r]$ with left child the node of $\mathcal{L}$ and right child the node of $\mathcal{R}$; update the *frontier* (replace the two leaves by their parent so the frontier again equals $\mathcal{P}^{(\tau)}$).

   4. **(Optional) single-boundary glue opt** at link $p{-}1 \rightarrow p$ if mode=*each*; re-snap the three boundary angles.

12: **Update counts and priorities in $\mathcal{D}$.** For each merged $(\mathcal{L}, \mathcal{R})$:

   1. **Merged pair decrement:** remove this occurrence from $\mathcal{O}(\kappa^\star)$; let the new count be $c_{\text{new}}$. Erase $\pi_{\text{old}} = (0, -c^\star, \kappa^\star)$ and, if $c_{\text{new}} > 0$, insert $(0, -c_{\text{new}}, \kappa^\star) \mapsto \mathcal{O}(\kappa^\star)$.

   2. **Neighbor decrements:** with neighbors $\mathcal{L}^-$ and $\mathcal{R}^+$ (when defined), compute $k_L = \text{COMPUTEGEOKEY}(\mathcal{L}^-, \mathcal{L})$ and $k_R = \text{COMPUTEGEOKEY}(\mathcal{R}, \mathcal{R}^+)$. For each $k \in \{k_L, k_R\}$ whose count decreases to $c_{\text{new}}$, erase $(\rho(k), -c_{\text{old}}, k)$ and, if $c_{\text{new}} > 0$, insert $(\rho(k), -c_{\text{new}}, k)$.

   3. **Neighbor increments:** compute $k'_L = \text{COMPUTEGEOKEY}(\mathcal{L}^-, \widetilde{\mathcal{M}})$ and $k'_R = \text{COMPUTEGEOKEY}(\widetilde{\mathcal{M}}, \mathcal{R}^+)$ (when defined); increment their counts and (re)insert with priorities $(\rho(k), -c_{\text{new}}, k)$, where $\rho(k) = \mathbf{1}[k \notin \text{dom}(\mathcal{V})]$.

13: **(Optional periodic global glue opt).** If mode=*all* and the schedule triggers, apply GLUEOPTALL (Alg. 12) to all modified backbones; recompute keys for their adjacent pairs, and for every affected key $k$, perform the same erase/insert priority update with $\rho(k) = \mathbf{1}[k \notin \text{dom}(\mathcal{V})]$. If FRONT($\mathcal{D}$) then exposes a recurring key ($\rho = 0$) promoted by glue refinement, immediately re-invoke STEP (no new clustering).

14: **return** $\{\mathcal{P}^{(\tau)}\}, \{\mathcal{F}^{(\tau)}\}, \mathcal{D}$, and $\mathcal{V}$.

---

---

**Algorithm 10** TOKENIZE — use learned GEOBPE vocabulary to tokenize new backbone

---

**Require:** New backbone $t^\star$ (length $N$); learned residue codebooks $\mathcal{A}_3, \mathcal{A}_2$; learned vocabulary $\mathcal{V}$ whose *pair-keys* are ordered by training insertion, written

$$\mathrm{Order}(\mathcal{V}) = \langle \kappa_1, \kappa_2, \ldots, \kappa_{|\mathcal{V}|} \rangle,$$

with each $\kappa_\ell$ mapped to its fixed prototype set $\mathcal{A}_{\kappa_\ell} = \{\Pi_j^{(\kappa_\ell)}\}_{j=1}^{K_{|\kappa_\ell|}}$; quantizers $Q_{\theta^{CNCA}}, Q_\omega, Q_\phi$; optional glue mode $\in \{none, each, all\}$ and (if *all*) a period $P$.

**Ensure:** Tokenized segmentation $\mathcal{P}^{(\star)}$ and merge hierarchy $\mathcal{F}^{(\star)}$ for $t^\star$.

1: **Per-residue init (no new clustering).** Set $\mathcal{P}^{(\star)} \leftarrow (\mathcal{M}_{1:1}^{(t^\star)}, \ldots, \mathcal{M}_{N:N}^{(t^\star)})$. Assign each residue motif to the nearest element of $\mathcal{A}_3$ (interior) or $\mathcal{A}_2$ (terminal) by Kabsch-aligned RMSD, and overwrite its internal parameters accordingly.

2: **Initialize hierarchy.** Let $\mathcal{F}^{(\star)}$ be a binary forest whose leaves (in order) are $\{\mathcal{M}_{i:i}^{(t^\star)}\}_{i=1}^N$; its frontier equals $\mathcal{P}^{(\star)}$.

3: **Optional one-time global glue.** If mode=*all*, run GLUEOPTALL (Alg. 12) on $t^\star$ once; snap $(\theta^{CNCA}, \omega, \phi)$ to $Q$.

4: **Build single-structure geo-pair map.** Compute $\mathcal{D}^{(\star)} \leftarrow \mathrm{BINHELPER}(t^\star, \mathcal{P}^{(\star)}, Q)$ (Alg. 11), which maps any geo-pair key $\kappa$ to its occurrence set $\mathcal{O}^{(\star)}(\kappa)$ on $t^\star$.  (Uses COMPUTEGEOKEY with raw medoid internals and quantized boundary glue.)

5: **Apply learned merges in training order (no new keys).**

6: **for** $s = 1$ **to** $|\mathcal{V}|$ **do**

7:    $\kappa \leftarrow \kappa_s$   (the $s$-th key in $\mathrm{Order}(\mathcal{V})$).

8:    **if** $\kappa \notin \mathcal{D}^{(\star)}$ **then**

9:       **continue**

10:   **end if**

11:   **Assign prototypes (no clustering).** For each $(\mathcal{L}, \mathcal{R}) \in \mathcal{O}^{(\star)}(\kappa)$ with $\mathcal{L} = \mathcal{M}_{p:q}^{(t^\star)}$, $\mathcal{R} = \mathcal{M}_{q+1:r}^{(t^\star)}$, set

$$c(\mathcal{L}, \mathcal{R}) = \arg \min_{j \in \{1, \ldots, K_{|\kappa|}\}} \mathrm{RMSD}(\mathcal{M}_{p:r}^{(t^\star)}, \Pi_j^{(\kappa)}).$$

12:   **Greedy disjoint merges & hierarchy updates (left→right).** Order $\mathcal{O}^{(\star)}(\kappa)$ by increasing $p$; select the maximal disjoint subset $S^{(\star)}$. For each $(\mathcal{L}, \mathcal{R}) \in S^{(\star)}$ with label $j = c(\mathcal{L}, \mathcal{R})$:

    1.   Form $\widetilde{\mathcal{M}} = \mathcal{M}_{p:r}^{(t^\star)}$ and overwrite its internals by $\Pi_j^{(\kappa)}$.

    2.   Update segmentation $\mathcal{P}^{(\star)}$: replace $(\mathcal{L}, \mathcal{R})$ by $\widetilde{\mathcal{M}}$.

    3.   *Update hierarchy* $\mathcal{F}^{(\star)}$: add a parent for span $[p{:}r]$ with left child the node of $\mathcal{L}$ and right child the node of $\mathcal{R}$; update the frontier so it equals the new $\mathcal{P}^{(\star)}$.

    4.   If mode=*each* and the boundary $p{-}1 \to p$ exists, apply GLUEOPT (Alg. 19); snap its three angles to $Q$.

13:   **Maintain the single-structure map.** For each merged $(\mathcal{L}, \mathcal{R}) \in S^{(\star)}$, update $\mathcal{D}^{(\star)}$ locally: remove the occurrence of $\kappa$; decrement keys of neighbors $(\mathcal{L}^-, \mathcal{L})$ and $(\mathcal{R}, \mathcal{R}^+)$ (when defined); insert the new neighbor keys $(\mathcal{L}^-, \widetilde{\mathcal{M}})$ and $(\widetilde{\mathcal{M}}, \mathcal{R}^+)$ using COMPUTEGEOKEY.

14:   **Optional periodic global glue.** If mode=*all* and $s \bmod P = 0$, run GLUEOPTALL on $t^\star$; then recompute keys adjacent to changed boundaries via COMPUTEGEOKEY and refresh their occurrences in $\mathcal{D}^{(\star)}$.

15: **end for**

16: **return** $\mathcal{P}^{(\star)}$ and $\mathcal{F}^{(\star)}$.

---

---

**Algorithm 11** BINHELPER — build geo-pair map for one backbone

---

**Require:** Backbone $t^\star$ with current segmentation $\mathcal{P}^{(\star)}$; quantizers $Q_{\theta^{CNCA}}, Q_\omega, Q_\phi$.
**Ensure:** $\mathcal{D}^{(\star)} : \kappa \mapsto \mathcal{O}^{(\star)}(\kappa)$.
  1: Initialize $\mathcal{D}^{(\star)} \leftarrow \emptyset$.
  2: **for** each adjacent pair $(\mathcal{L}, \mathcal{R})$ in $\mathcal{P}^{(\star)}$ **do**
  3:     $\kappa \leftarrow$ COMPUTEGEOKEY$(\mathcal{L}, \mathcal{R})$ using:
          • *Raw* medoid internals for $\mathcal{L}$ and $\mathcal{R}$ (as assigned in initialization or prior merges);
          • boundary glue $(\theta^{CNCA}, \omega, \phi)$ snapped by $Q$.
  4:     Insert the occurrence $(\mathcal{L}, \mathcal{R})$ into $\mathcal{O}^{(\star)}(\kappa)$.
  5: **end for**
  6: **return** $\mathcal{D}^{(\star)}$.

---

---

**Algorithm 12** GLUEOPTALL — global differentiable inverse kinematics over glue angles

---

**Require:** Medoids $\widehat{\mathcal{M}}$ and assignments $c(\cdot)$ from RMSD_PARTITION; occurrences $\mathcal{S} = \{u\}$ with spans $\mathcal{M}_{i_u:k_u}^{(t_u)}$; target frames $F_i^{\star,(t)} = (R_i^{\star,(t)}, t_i^{\star,(t)})$ (with $F_1^{\star,(t)}$ from SEEDTRIAD); weights $(w_R, w_t)$; optimizer steps $T$ and step size $\eta$
**Ensure:** Updated glues $\{\Gamma_i^{(t)}\}$ and frames $\{\widehat{F}_i^{(t)}\}$
  1: **Snap internals:** for $u \in \mathcal{S}$, set internals of $\mathcal{M}_{i_u:k_u}^{(t_u)} \leftarrow$ those of its medoid $m(u) = \widehat{m}_{c(u)}$
  2: **Init glues:** copy original $\Gamma_i^{(t)}$ for all backbones $t$ and links $i = 1{:}N^{(t)}-1$ (these are the optimization variables)
  3: **Loss:**
$$\mathcal{L}(\Gamma) = \sum_t \sum_{i=2}^{N^{(t)}} \Big( w_R \big\| \log((\widehat{R}_i^{(t)})^\top R_i^{\star,(t)}) \big\|_2^2 + w_t \big\| \widehat{t}_i^{(t)} - t_i^{\star,(t)} \big\|_2^2 \Big)$$
  4: **Forward kinematics (FK):** with $\widehat{F}_1^{(t)} = F_1^{\star,(t)}$,
$$\widehat{F}_{i+1}^{(t)} = \widehat{F}_i^{(t)} \, \widehat{G}_i^{(t)}\big(\Gamma_i^{(t)}; \text{ current internals}\big), \quad \widehat{G}_i^{(t)} \text{ from internals and } \Gamma_i^{(t)} = \{\theta_i^{CNCA}, \psi_i, \phi_{i+1}\}^2$$
  5: **Optimize glues (autodiff):** for $s = 1{:}T$:
  6:     run FK, evaluate $\mathcal{L}$; backprop $\nabla_\Gamma \mathcal{L}$; update all $\Gamma_i^{(t)}$
  7:     wrap $\psi, \phi \in (-\pi, \pi]$; project $\theta^{CNCA} \in (0, \pi)$
  8: **return** $\{\Gamma_i^{(t)}\}, \{\widehat{F}_i^{(t)}\}$

---

---

**Algorithm 13** BINARY TREE–LSTM CELL (Tai et al., 2015)

---

**Require:** Left $(h_\ell, c_\ell) \in \mathbb{R}^d \times \mathbb{R}^d$, right $(h_r, c_r)$; $W \in \mathbb{R}^{5d \times 2d}, b \in \mathbb{R}^{5d}$
**Ensure:** $(h_p, c_p) \in \mathbb{R}^d \times \mathbb{R}^d$
  1: $u \leftarrow \begin{bmatrix} h_\ell \\ h_r \end{bmatrix}$; $\begin{bmatrix} i \\ f_\ell \\ f_r \\ o \\ g \end{bmatrix} \leftarrow W u + b$
  2: $i, f_\ell, f_r, o \leftarrow \sigma(\cdot)$; $g \leftarrow \tanh(g)$
  3: $c_p \leftarrow f_\ell \odot c_\ell + f_r \odot c_r + i \odot g$
  4: $h_p \leftarrow o \odot \tanh(c_p)$
  5: **return** $(h_p, c_p)$

---

---

**Algorithm 14** DOWNWARD BINARY TREE–LSTM CELL

---

**Require:** Parent downward $(\bar{h}_p, \bar{c}_p) \in \mathbb{R}^d \times \mathbb{R}^d$, sibling upward $(h_s, c_s)$; $\widetilde{W} \in \mathbb{R}^{5d \times 2d}, \widetilde{b} \in \mathbb{R}^{5d}$
**Ensure:** $(\bar{h}_c, \bar{c}_c) \in \mathbb{R}^d \times \mathbb{R}^d$

1: $u \leftarrow \begin{bmatrix} \bar{h}_p \\ h_s \end{bmatrix}$; $\begin{bmatrix} \bar{i} \\ \bar{f}_p \\ \bar{f}_s \\ \bar{o} \\ \bar{g} \end{bmatrix} \leftarrow \widetilde{W} u + \widetilde{b}$

2: $\bar{i}, \bar{f}_p, \bar{f}_s, \bar{o} \leftarrow \sigma(\cdot)$;  $\bar{g} \leftarrow \tanh(\bar{g})$
3: $\bar{c}_c \leftarrow \bar{f}_p \odot \bar{c}_p + \bar{f}_s \odot c_s + \bar{i} \odot \bar{g}$
4: $\bar{h}_c \leftarrow \bar{o} \odot \tanh(\bar{c}_c)$
5: **return** $(\bar{h}_c, \bar{c}_c)$

---

**Algorithm 15** UP–DOWN TREE ENCODER ON A FOREST (one protein)

---

**Require:** Protein $t_\tau$ with $N^{(\tau)}$ residues; binary forest $\mathcal{F}^{(\tau)} = (V^{(\tau)}, E^{(\tau)})$ whose frontier (in order) is $\mathcal{P}^{(\tau)}$; leaf embeddings $\{e_i^{(\tau)} \in \mathbb{R}^d\}_{i=1}^{N^{(\tau)}}$ (e.g., ESM3[3]; internal-edge topological order $E^{(\tau)} = \{(p, \ell, r)\}$; roots $R^{(\tau)} \subset V^{(\tau)}$; parameters $\Theta = \{W, b, \widetilde{W}, \widetilde{b}\}$; combiner $\oplus \in \{\text{concat}, \text{sum}\}$.
**Ensure:** $z_\tau^{\text{prot}} \in \mathbb{R}^{d_z}$; $\{z_{\tau,i}^{\text{res}}\}_{i=1}^{N^{(\tau)}}$.

1: **Upward.** For leaves $i \leq N^{(\tau)}$: $h_i^\uparrow \leftarrow e_i^{(\tau)}$, $c_i^\uparrow \leftarrow 0$. For $(p, \ell, r) \in E^{(\tau)}$ in order:

$$(h_p^\uparrow, c_p^\uparrow) \leftarrow \text{TREELSTMCELL}(h_\ell^\uparrow, c_\ell^\uparrow, h_r^\uparrow, c_r^\uparrow; W, b) \text{ (Alg. 13)}.$$

2: **Super-root.** $h_{\text{SR}}^\uparrow \leftarrow |R^{(\tau)}|^{-1} \sum_{r \in R^{(\tau)}} h_r^\uparrow$; set node SR with $(h_{\text{SR}}^\uparrow, c_{\text{SR}}^\uparrow = 0)$.
3: **Downward.** $(\bar{h}_{\text{SR}}, \bar{c}_{\text{SR}}) \leftarrow (0, 0)$. For each tree rooted at $r \in R^{(\tau)}$, recurse: for internal $p$ with children $(\ell, r)$ and given $(\bar{h}_p, \bar{c}_p)$,

$$(\bar{h}_\ell, \bar{c}_\ell) \leftarrow \text{DOWNTREELSTM}((\bar{h}_p, \bar{c}_p), (h_r^\uparrow, c_r^\uparrow); \widetilde{W}, \widetilde{b}),$$

$$(\bar{h}_r, \bar{c}_r) \leftarrow \text{DOWNTREELSTM}((\bar{h}_p, \bar{c}_p), (h_\ell^\uparrow, c_\ell^\uparrow); \widetilde{W}, \widetilde{b}) \text{ (Alg. 14)}.$$

4: **Representations.** For any node $v$: $u_v^\downarrow \leftarrow \bar{h}_v$.
5: **if** $\oplus = \text{concat}$ **then**
6:   $z_{\tau,i}^{\text{res}} \leftarrow [h_i^\uparrow; u_i^\downarrow] \in \mathbb{R}^{2d}$ $(i=1{:}N^{(\tau)})$;  $z_\tau^{\text{prot}} \leftarrow [h_{\text{SR}}^\uparrow; u_{\text{SR}}^\downarrow] \in \mathbb{R}^{2d}$
7: **else**
8:   $z_{\tau,i}^{\text{res}} \leftarrow h_i^\uparrow + u_i^\downarrow \in \mathbb{R}^d$ $(i=1{:}N^{(\tau)})$;  $z_\tau^{\text{prot}} \leftarrow h_{\text{SR}}^\uparrow + u_{\text{SR}}^\downarrow \in \mathbb{R}^d$
9: **end if**
10: **return** $z_\tau^{\text{prot}}, \{z_{\tau,i}^{\text{res}}\}_{i=1}^{N^{(\tau)}}$.

---

---

**Algorithm 16** BUILDJOINTVOCAB — medoids then glue–angle bins

---

**Require:** GEOBPE vocab $\mathcal{V} = \{\kappa \mapsto \mathcal{A}_\kappa\}$ with key introduction order $(\kappa^{(1)}, \ldots, \kappa^{(S)})$; medoids $\mathcal{A}_\kappa = \{\Pi_j^{(\kappa)}\}_{j=1}^{K_{|\kappa|}}$; glue quantizers $Q_{\theta^{CNCA}}, Q_\omega, Q_\phi$ with bin centers $\{\mu_b^\theta\}_{b=1}^{B_\theta}$, $\{\mu_b^\omega\}_{b=1}^{B_\omega}$, $\{\mu_b^\phi\}_{b=1}^{B_\phi}$

**Ensure:** Dictionary $\Sigma$; maps $\mathrm{id}_{\mathrm{med}} : (\kappa, j) \mapsto \{1, \ldots, |\Sigma_{\mathrm{med}}|\}$ and $\mathrm{id}_{\mathrm{bin}} : (\mathrm{type} \in \{\theta, \omega, \phi\}, b) \mapsto \{|\Sigma_{\mathrm{med}}|+1, \ldots, |\Sigma|\}$

1: **Medoids (in introduction order):** $\Sigma_{\mathrm{med}} \leftarrow [\,]$.
2: **for** $s = 1$ **to** $S$ **do**
3:    **for** $j = 1$ **to** $K_{|\kappa^{(s)}|}$ **do**
4:       Append $\langle \kappa^{(s)}, j \rangle$ to $\Sigma_{\mathrm{med}}$; set $\mathrm{id}_{\mathrm{med}}(\kappa^{(s)}, j)$ to its index.
5:    **end for**
6: **end for**
7: **Glue bins (appended after medoids):** let $M = |\Sigma_{\mathrm{med}}|$.
8: $\mathrm{id}_{\mathrm{bin}}(\theta, b) = M + b$;   $\mathrm{id}_{\mathrm{bin}}(\omega, b) = M + B_\theta + b$;   $\mathrm{id}_{\mathrm{bin}}(\phi, b) = M + B_\theta + B_\omega + b$.
9: $\Sigma \leftarrow \Sigma_{\mathrm{med}} \cup \{\text{all glue-bin tokens}\}$ (*optional*: add BOS/EOS)
10: **return** $\Sigma, \mathrm{id}_{\mathrm{med}}, \mathrm{id}_{\mathrm{bin}}$

---

**Algorithm 17** BACKBONETOSEQUENCE — tokenize a segmented backbone

---

**Require:** Protein $t_\tau$ with segmentation $\mathcal{P}^{(\tau)} = (\mathcal{M}_{p_1:q_1}^{(t_\tau)}, \ldots, \mathcal{M}_{p_M:q_M}^{(t_\tau)})$; for each $\mathcal{M}_{p_m:q_m}^{(t_\tau)}$ its key $\kappa_m$ and medoid $j_m$ (prototype $\Pi_{j_m}^{(\kappa_m)}$); boundary glue $\Gamma_{q_m} = \{\theta_{q_m}^{CNCA}, \omega_{q_m}, \phi_{q_m+1}\}$ for $m = 1{:}M{-}1$; quantizers $Q_\theta, Q_\omega, Q_\phi$; token id maps from Alg. 16.

**Ensure:** Token sequence $x^{(\tau)} = (x_1, \ldots, x_L) \in \Sigma^L$.

1: $x^{(\tau)} \leftarrow [\,]$ (optionally prepend BOS/append EOS)
2: **for** $m = 1$ **to** $M$ **do**
3:    **Motif:** $x^{(\tau)}.\mathrm{append}\big(\mathrm{id}_{\mathrm{med}}(\kappa_m, j_m)\big)$
4:    **if** $m < M$ **then**
5:       **Glue quantize:** $b_\theta \leftarrow Q_\theta(\theta_{q_m}^{CNCA})$,  $b_\omega \leftarrow Q_\omega(\omega_{q_m})$,  $b_\phi \leftarrow Q_\phi(\phi_{q_m+1})$
6:       **Emit (fixed order):**   $x^{(\tau)}.\mathrm{append}\big(\mathrm{id}_{\mathrm{bin}}(\theta, b_\theta)\big)$;   $x^{(\tau)}.\mathrm{append}\big(\mathrm{id}_{\mathrm{bin}}(\omega, b_\omega)\big)$; $x^{(\tau)}.\mathrm{append}\big(\mathrm{id}_{\mathrm{bin}}(\phi, b_\phi)\big)$
7:    **end if**
8: **end for**
9: **return** $x^{(\tau)}$

---

---

**Algorithm 18** RESINITTOKENS — initialize bond–residue codebook and quantize all residues

---

**Require:** Backbones $\{t^{(1)}, \ldots, t^{(T)}\}$ with lengths $N^{(\tau)}$; targets $K_3$ (interior bond–residues), $K_2$ (terminal bond–residues)

**Ensure:** Codebooks $\mathcal{A}_3 = \{\Pi_j^{(3)}\}_{j=1}^{K_3}$, $\mathcal{A}_2 = \{\Pi_j^{(2)}\}_{j=1}^{K_2}$; labels $c^{(3)}, c^{(2)}$; backbones with per-residue internals set to their prototypes

1: **Collect occurrences:**

$$\mathcal{S}_3 = \{u \equiv (\tau, i) : 1 \le i < N^{(\tau)}, \mathcal{M}_{i:i}^{(t_\tau)} \text{ interior}\}, \quad \mathcal{S}_2 = \{u \equiv (\tau, i) : i = N^{(\tau)}, \mathcal{M}_{i:i}^{(t_\tau)} \text{ terminal}\}.$$

2: **Cluster interiors:** RMSD_PARTITION$(\mathcal{S}_3, K_3) \to$ medoids $\widehat{\mathcal{M}}_3 = \{\widehat{m}_j^{(3)}\}_{j=1}^{K_3}$ and labels $c^{(3)} : \mathcal{S}_3 \to \{1, \ldots, K_3\}$

3: **Define interior prototypes:** for $j = 1{:}K_3$, let $u^\star = \widehat{m}_j^{(3)}$ and

$$\Pi_j^{(3)} = \left(\ell_{i_{u^\star}}^{N-CA}, \ell_{i_{u^\star}}^{CA-C}, \ell_{i_{u^\star}}^{C-N}, \theta_{i_{u^\star}}^{NCAC}, \theta_{i_{u^\star}}^{CACN}, \psi_{i_{u^\star}}, \omega_{i_{u^\star}}, \phi_{i_{u^\star}}\right)$$

(*omit undefined terms for a single residue if using a minimal parameterization*).

4: **Quantize interiors:** for $u = (\tau, i) \in \mathcal{S}_3$ with $j = c^{(3)}(u)$,

$$\left.(\ell, \theta, \psi, \omega, \phi)\right|_{\mathcal{M}_{i:i}^{(t_\tau)}} \leftarrow \Pi_j^{(3)}.$$

5: **Cluster terminals:** RMSD_PARTITION$(\mathcal{S}_2, K_2) \to$ medoids $\widehat{\mathcal{M}}_2 = \{\widehat{m}_j^{(2)}\}_{j=1}^{K_2}$ and labels $c^{(2)} : \mathcal{S}_2 \to \{1, \ldots, K_2\}$

6: **Define terminal prototypes & quantize:** for $j = 1{:}K_2$, let $u^\star = \widehat{m}_j^{(2)}$ and set the appropriate terminal tuple (e.g., $\ell_i^{N-CA}, \ell_i^{CA-C}, \theta_i^{NCAC}$) as $\Pi_j^{(2)}$; for $u = (\tau, N^{(\tau)})$ with $j = c^{(2)}(u)$,

$$\left.(\ell, \theta)\right|_{\mathcal{M}_{N^{(\tau)}:N^{(\tau)}}^{(t_\tau)}} \leftarrow \Pi_j^{(2)}.$$

7: **return** $\mathcal{A}_3, \mathcal{A}_2, c^{(3)}, c^{(2)}$, and updated backbones

---

---

**Algorithm 19** GLUEOPT — single-boundary IK to absorb one rounding drift

---

**Require:** Occurrence $u$ with motif $\mathcal{M}_{i_u:k_u}^{(t_u)}$ and medoid $\widehat{m}_{c(u)}$ from RMSD_PARTITION; frames $\{F_i^{\star,(t_u)}\}$ with $F_1^{\star,(t_u)}$ from SEEDTRIAD; weights $(w_R, w_t)$; steps $T$, step size $\eta$

**Ensure:** $\Gamma_{i_u-1}^{(t_u)}, \widehat{F}_{k_u}^{(t_u)}$

1: **Snap internals:** replace $\mathcal{M}_{i_u:k_u}^{(t_u)}$ by its medoid $\mathcal{M}_{i_{\widehat{m}_{c(u)}}:k_{\widehat{m}_{c(u)}}}^{(t_{\widehat{m}_{c(u)}})}$; set $T_u^{\text{med}} \leftarrow T_{i_{\widehat{m}_{c(u)}}:k_{\widehat{m}_{c(u)}}}^{\text{int}}$

2: **Drift:** $T_u^{\text{occ}} \leftarrow T_{i_u:k_u}^{\text{int}}$; $\Delta T_u \leftarrow T_u^{\text{occ}}(T_u^{\text{med}})^{-1}$

3: **Vars:** $\Gamma_{i_u-1}^{(t_u)} = \{\theta_{i_u-1}^{CNCA}, \omega_{i_u-1}, \varphi_{i_u}\}$ are the *only* optimization variables[4]; init to originals

4: **FK:** keep $F_{i_u-1}^{\star,(t_u)}$ fixed; for any $\Gamma_{i_u-1}^{(t_u)}$,

$$\widehat{F}_{k_u}^{(t_u)} = F_{i_u-1}^{\star,(t_u)} \widehat{G}_{i_u-1}^{(t_u)}(\Gamma_{i_u-1}^{(t_u)}) T_u^{\text{med}}$$

5: **Loss:**

$$\mathcal{L}_u(\Gamma_{i_u-1}^{(t_u)}) = w_R \left\| \log((\widehat{R}_{k_u}^{(t_u)})^\top R_{k_u}^{\star,(t_u)}) \right\|_2^2 + w_t \left\| \widehat{t}_{k_u}^{(t_u)} - t_{k_u}^{\star,(t_u)} \right\|_2^2$$

6: **Optimize (autodiff):** for $s = 1{:}T$: run FK & evaluate $\mathcal{L}_u$; compute $\nabla_{\Gamma_{i_u-1}^{(t_u)}} \mathcal{L}_u$; update $\Gamma_{i_u-1}^{(t_u)}$ (e.g., Adam, lr $\eta$); wrap $\psi, \varphi \in (-\pi, \pi]$, project $\theta^{CNCA} \in (0, \pi)$

7: **return** $\Gamma_{i_u-1}^{(t_u)}, \widehat{F}_{k_u}^{(t_u)}$

---

---

**Algorithm 20** GLUEOPTALL (WRAPPER) — apply rounding, then call the core global IK over glues

---

**Require:** Medoids $\widehat{\mathcal{M}}$ and assignments $c(\cdot)$ from RMSD_PARTITION; a set of occurrences $\mathcal{S} = \{u\}$ to round, each $\mathcal{M}_{i_u:k_u}^{(t_u)}$; cached original exit frames for each backbone $t$ (seeded by SEED-TRIAD); histogram bin centers & thresholds for $(\omega, \theta^{CNCA}, \varphi)$; prior weights; loss weights $(w_R, w_t)$.

**Ensure:** Updated glue angles (snapped to bins) for all boundaries in all affected backbones; recomputed frames $\{\widehat{F}_i^{(t)}\}$.

1: **(Quantize internals)** For each $u \in \mathcal{S}$, replace

$$\mathcal{M}_{i_u:k_u}^{(t_u)} \longleftarrow \mathcal{M}_{i_{\widehat{m}_{c(u)}}:k_{\widehat{m}_{c(u)}}}^{(t_{\widehat{m}_{c(u)}})}$$

by copying the medoid's internal coordinates (hard assignment).

2: **(Ensure targets are cached)** For each backbone $t$, ensure original exit frames $\{(R_{\text{occ}}^{\star,(t)}[j], t_{\text{occ}}^{\star,(t)}[j])\}_{j=1}^{N^{(t)}-1}$ are available (compute once if missing).

3: **(Global glue optimization)** *Call the core routine* GLUEOPTALL *on all backbones:*

$$\text{GLUEOPTALL}\Big(\{t\}, \text{BinCenters}, \text{Thresholds}, \text{GluePrior}, w_R, w_t\Big),$$

which jointly optimizes every boundary's glue triplet $\Gamma_i = \{\theta_i^{CNCA}, \omega_i, \varphi_{i+1}\}$ via differentiable FK and snaps each angle to the nearest histogram bin.

4: **return** updated glues and frames $\{\widehat{F}_i^{(t)}\}$.

---

---

**Algorithm 21** BINGEOPAIRS — build the dictionary of geo-pair occurrences (with hierarchy)

---

**Require:** For each backbone $t^{(\tau)}$: its current segmentation $\mathcal{P}^{(\tau)} = (\mathcal{M}_{p_1:q_1}^{(t_\tau)}, \ldots, \mathcal{M}_{p_{M_\tau}:q_{M_\tau}}^{(t_\tau)})$ *and* its merge hierarchy $\mathcal{F}^{(\tau)}$ whose frontier leaves, in order, equal $\mathcal{P}^{(\tau)}$; precomputed boundary-glue quantizers $Q_{\theta^{CNCA}}, Q_\omega, Q_\phi$ (used only at pair boundaries).

**Ensure:** A *priority-ordered* map $\mathcal{D}$ from keys to occurrence sets, with ordered keys $\pi(\kappa) = (\rho(\kappa), -|\mathcal{O}(\kappa)|, \kappa)$ where $\rho(\kappa) = \mathbf{1}[\kappa \notin \text{dom}(\mathcal{V})]$ indicates if the key already has prototypes.

1: Initialize an empty ordered map $\mathcal{D}$.

2: **for** $\tau = 1$ **to** $T$ **do**

3:     Let $(\mathcal{M}_{p_1:q_1}^{(t_\tau)}, \ldots, \mathcal{M}_{p_{M_\tau}:q_{M_\tau}}^{(t_\tau)})$ be the frontier leaves of $\mathcal{F}^{(\tau)}$ (these equal $\mathcal{P}^{(\tau)}$).

4:     **for** $j = 1$ **to** $M_\tau - 1$ **do**

5:         $(\mathcal{L}, \mathcal{R}) \leftarrow \big(\mathcal{M}_{p_j:q_j}^{(t_\tau)}, \mathcal{M}_{p_{j+1}:q_{j+1}}^{(t_\tau)}\big)$

6:         $\kappa \leftarrow$ COMPUTEGEOKEY$(\mathcal{L}, \mathcal{R})$       (raw medoid internals inside $\mathcal{L}, \mathcal{R}$; boundary glue quantized by $Q$)

7:         Insert the occurrence $(\mathcal{L}, \mathcal{R})$ into $\mathcal{O}(\kappa)$.

8:     **end for**

9: **end for**

10: **for** each key $\kappa$ with nonempty $\mathcal{O}(\kappa)$ **do**

11:     Set $\rho(\kappa) \leftarrow \mathbf{1}[\kappa \notin \text{dom}(\mathcal{V})]$.

12:     Insert $\big((\rho(\kappa), -|\mathcal{O}(\kappa)|, \kappa) \mapsto \mathcal{O}(\kappa)\big)$ into $\mathcal{D}$.

13: **end for**

14: **return** $\mathcal{D}$.

---

---

**Algorithm 22** COMPUTEGEOKEY — discrete key for an adjacent motif pair

---

**Require:** Adjacent motifs $\mathcal{L} = \mathcal{M}_{p:q}^{(t)}$, $\mathcal{R} = \mathcal{M}_{q+1:r}^{(t)}$ on backbone $t$
**Ensure:** Canonical, hashable key $\kappa$ for the geo-pair
 1: **Interiors (as stored post-quantization):**

$$\text{Int}(\mathcal{L}) = \left(\{\ell, \theta, \psi, \omega, \phi\}\big|_{i=p}^{q}, \{\Gamma_i\}_{i=p}^{q-1}\right), \quad \text{Int}(\mathcal{R}) = \left(\{\ell, \theta, \psi, \omega, \phi\}\big|_{i=q+1}^{r}, \{\Gamma_i\}_{i=q+1}^{r-1}\right)$$

(kept unchanged in the key).
 2: **Boundary glue (quantized):**

$$\Gamma_q = (\theta_q^{CNCA}, \omega_q, \phi_{q+1}), \qquad \widetilde{\Gamma}_q = \left(Q_\theta(\theta_q^{CNCA}), Q_\omega(\omega_q), Q_\phi(\phi_{q+1})\right)$$

($Q_\bullet$ wrap to a fixed $2\pi$ interval before snapping).
 3: **Canonical record & hash:**

$$\text{rec} = \left(\text{Int}(\mathcal{L}), \widetilde{\Gamma}_q, \text{Int}(\mathcal{R})\right), \qquad \kappa \leftarrow \text{HASH(rec)}$$

 4: **return** $\kappa$

---

**Algorithm 23** COMPUTE_COORDS — Internal $\rightarrow$ Cartesian for a bond–residue motif $i{:}j$

---

**Require:** Internal geometry for $r = i, \ldots, j$ (bond lengths/angles/dihedrals).
**Ensure:** $\mathbf{X} \in \mathbb{R}^{3(j-i+1)\times 3}$ for $(N_i, \text{CA}_i, C_i, \ldots, N_j, \text{CA}_j, C_j)$
 1: **Seed residue** $i$: $(N_i, \text{CA}_i, C_i) \leftarrow \text{SEEDTRIAD}(i)$
 2: $\mathcal{C} \leftarrow [N_i, \text{CA}_i, C_i]$
 3: **for** $r = i+1$ **to** $j$ **do**
 4: $\quad N_r \leftarrow \text{PLACEDIHEDRAL}\left(\mathcal{C}[-3], \mathcal{C}[-2], \mathcal{C}[-1]; \ell_{r-1}^{C-N}, \theta_{r-1}^{CACN}, \psi_{r-1}\right); \mathcal{C}.\text{append}(N_r);$
 5: $\quad \text{CA}_r \leftarrow \text{PLACEDIHEDRAL}\left(\mathcal{C}[-3], \mathcal{C}[-2], \mathcal{C}[-1]; \ell_r^{N-CA}, \theta_{r-1}^{CNCA}, \omega_{r-1}\right); \mathcal{C}.\text{append}(\text{CA}_r)$
 6: $\quad C_r \leftarrow \text{PLACEDIHEDRAL}\left(\mathcal{C}[-3], \mathcal{C}[-2], \mathcal{C}[-1]; \ell_r^{CA-C}, \theta_r^{NCAC}, \varphi_r\right); \mathcal{C}.\text{append}(C_r)$
 7: **end for**
 8: $\mathbf{X} \leftarrow [\, N_i, \text{CA}_i, C_i, \ldots, N_j, \text{CA}_j, C_j \,]$
 9: **return** $\mathbf{X}$

---

**Algorithm 24** PLACEDIHEDRAL$(a, b, c; L, \beta, \tau)$

---

 1: Right-handed local frame at $c$:

$$\hat{\mathbf{b}} = \frac{c - b}{\|c - b\|}, \quad \mathbf{n} = \frac{(b - a) \times \hat{\mathbf{b}}}{\|(b - a) \times \hat{\mathbf{b}}\|}, \quad \tilde{\mathbf{n}} = \mathbf{n} \times \hat{\mathbf{b}}.$$

 2: Local offset:

$$\mathbf{d} = \left[-L\cos\beta\right]\hat{\mathbf{b}} + \left[L\cos\tau\sin\beta\right]\tilde{\mathbf{n}} + \left[L\sin\tau\sin\beta\right]\mathbf{n}.$$

 3: Return $d = c + \mathbf{d}$.

---

**Algorithm 25** SEEDTRIAD$(r)$ — seed triad for residue $r$

---

**Require:** Canonical seed $(N_\star, \text{CA}_\star, C_\star)$; target $L_{CA-C} = \ell_r^{CA-C}$, $L_{N-CA} = \ell_r^{N-CA}$, and $\theta_{NCAC} = \theta_r^{NCAC}$.
 1: Place $\widetilde{\text{CA}}_r$ on the ray from $C_\star$ toward $\text{CA}_\star$ at distance $L_{CA-C}$.
 2: Let $\mathbf{u} = N_\star - \widetilde{\text{CA}}_r$ and $\mathbf{v} = C_\star - \widetilde{\text{CA}}_r$. Rotate $\mathbf{u}$ about axis $\mathbf{u} \times \mathbf{v}$ to achieve angle $\theta_{NCAC}$, then rescale to length $L_{N-CA}$; translate by $\widetilde{\text{CA}}_r$ to get $N_r$.
 3: Set $C_r \leftarrow C_\star$ and return $(N_r, \widetilde{\text{CA}}_r, C_r)$.

---

