# OpenReview forum: "Protein Structure Tokenization via Geometric Byte Pair Encoding"
_ICLR.cc/2026/Conference — ICLR 2026 Poster_

### Official Review · Reviewer_4qDK · 2025-10-31

**Soundness:** 4
**Presentation:** 2
**Contribution:** 4
**Rating:** 10
**Confidence:** 5

**Summary:**

The authors present GeoBPE, a protein structure tokenizer which produces hierarchical tokenizations of proteins using a byte-pair encoding-like technique. In brief, residue-level backbone geometric features are clustered into a vocabulary, and then iteratively merged to reduce the size of the vocabulary and make it less sensitive to noise. This induces distortions in the resulting geometry of a protein, which they resolve by making corrections at each stage of the tokenization. The approach is remarkably effective, resulting in increased compression and reconstruction accuracy over strong baselines. ProToken and GeoBPE form a Pareto front, trading off reconstruction accuracy for compression, with GeoBPE performing roughly 10x better on BPR (compression) than ProToken.

**Strengths:**

Originality: I think the authors have come up with a creative and quite distinct solution to the problem of structure tokenization--while many others have tried to improve on the VQ-VAE idea, they have not broken out of that fundamental paradigm. GeoBPE is entirely different and at the same time simpler and more intuitive.

Quality: The analysis is thorough, covering structure reconstruction, codebook usage, and several downstream tasks of high biological relevance.

Significance: GeoBPE substantially outperforms most other structure tokenizers in nearly all evaluations, and provides clear trade-offs between compression and accuracy. I imagine that it will be of great utility to researchers modeling proteins, and hopefully will find use in a protein LM soon.

**Weaknesses:**

Clarity: The writing in section 3 could use a lot of improvement. I found it difficult to follow, and I'm still not entirely sure I know what all symbols mean. Many terms are only properly defined _after_ they've been used multiple times. The notation section is clear, but takes a lot of space that might be better given to (clearer) descriptions of the method itself.

It would be very helpful to visualize the hierarchy of structure tokens/motifs. I imagine, for instance, that there should be a large group of loop-like structures, with a smaller number of tokens devoted to common structures like alpha-helices and different kinds of beta sheets. Given the local nature of the representation, I would expect high level clusters corresponding to the 8 basic secondary structure elements.

**Questions:**

In this emerging field, there is some debate over what level and type of structural information each token should have. That is, each token can represent not just information internal to the residues represented, but more global information about how the residue is positioned in the protein. GeoBPE is essentially backbone-local--it does not use information about nearest spatial neighbors at all. By contrast ESM3's tokenizer does contain substantial information about residues which can be very distant in the linear sequence of the protein, yet GeoBPE does better on global reconstruction tasks. This is perhaps surprising and I wonder if the authors could comment on it.

Generally the peptide backbone bond lengths are very close to fixed (in fact there are usually restraints on these when solving structures by X-ray diffraction, NMR, or Cryo-EM). I wonder whether these are actually needed in GeoBPE; do the authors find clusters of motifs where these are very different, or are the bond lengths similar between all motifs? My guess is that the clusters differ primarily in their backbone dihedral angles

---

> ### Author Response · Authors · 2025-11-21
> **Response by GeoBPE Authors**
>
> **Thank you for your exceptionally thoughtful and generous review. We deeply appreciate your recognition of the novelty of GeoBPE’s approach. Your comments on the paradigm shift beyond VQ-VAE–style tokenizers are especially meaningful to us and encourage us to continue exploring new ideas around AI in structural biology!**
>
> *Clarity: The writing in section 3 could use a lot of improvement. I found it difficult to follow, and I'm still not entirely sure I know what all symbols mean. Many terms are only properly defined after they've been used multiple times. The notation section is clear, but takes a lot of space that might be better given to (clearer) descriptions of the method itself.*
>
> We took the following steps to improve accessibility of section 3:
>
> - Expanded method description:
> - Notations: Added Fig. 1 to visually communicate the notation; tightened the notations and corrected for typos.
> - Captions: Annotated Fig. 2’s caption with hand-styled icons to align with the figure.
>
> *It would be very helpful to visualize the hierarchy of structure tokens/motifs. I imagine, for instance, that there should be a large group of loop-like structures, with a smaller number of tokens devoted to common structures like alpha-helices and different kinds of beta sheets. Given the local nature of the representation, I would expect high level clusters corresponding to the 8 basic secondary structure elements.*
>
> Thank you for the suggestion! We’ve drawn concrete hierarchies of structure tokens for 2 biologically meaningful cases in App. H (was there before, we’ve redrawn to align with the style of the hierarchies in Fig 1).
>
> **Re the alignment with secondary structures:** your intuition is right. We ran a new expert agreement evaluation against the 8 basic SSEs (from DSSP) and added the results to App. C. The summary is follows:
>
> |Metric|H|G|I|E|B|T|S|-|Avg (HGIEBTS-)|
> |-|:-:|:-:|:-:|:-:|:-:|:-:|:-:|:-:|:-:|
> |Mean recall|31.98 (97.32)|29.62 (98.80)|20.09 (99.79)|35.58 (98.20)|36.20 (100.00)|32.96 (97.26)|33.85 (98.08)|33.17 (96.81)|31.68 (98.28)|
> |Mean precision|33.12 (49.99)|27.76 (67.61)|19.42 (90.31)|34.22 (47.65)|24.55 (68.95)|28.49 (51.12)|25.65 (50.71)|26.79 (50.23)|27.50 (59.57)|
> |Mean f1|32.01 (60.45)|28.54 (74.28)|19.70 (91.89)|34.66 (60.56)|28.41 (71.15)|30.20 (62.65)|28.46 (58.40)|28.92 (61.08)|28.86 (67.56)|
> |Mean iou|31.09 (60.18)|27.44 (74.21)|19.23 (91.88)|33.58 (60.20)|24.54 (69.05)|28.23 (60.93)|25.51 (53.77)|26.22 (55.27)|26.98 (65.69)|
> |Segment recall|32.05 (99.80)|29.71 (99.97)|20.14 (100.00)|35.63 (99.77)|35.74 (99.05)|32.95 (98.77)|33.48 (98.36)|33.38 (98.51)|31.64 (99.28)|
> |Segment precision|34.05 (61.01)|4.00 (74.42)|1.55 (88.84)|23.95 (64.15)|1.14 (98.77)|9.36 (71.67)|7.04 (82.45)|15.85 (81.10)|12.12 (77.80)|
> |Segment f1|31.39 (61.04)|6.91 (74.41)|2.77 (88.84)|26.95 (64.20)|2.19 (98.76)|14.31 (72.13)|11.30 (83.12)|20.98 (86.48)|14.60 (78.62)|
>
> We do see above-random enrichment of SSEs in our tokens across all-metrics. The recall of existing SSEs is exceptional: 98.28% block-level 99.28% segment-level, averaged over datasets and SSEs. This indicates the ability to recapitulate the 8 known elements, while the milder precision (59.57 block, 77.80 segment) hints that GeoBPE goes beyond SSEs. As prior agreement results and case studies show, GeoBPE can find biologically meaningful regions (e.g. conserved homology, functional sites) and is not constrained to SSE boundaries, with the data dictating the exact high-level clusters.
>
> (continues in next reply)

---

> ### Author Response · Authors · 2025-11-21
> **Response by GeoBPE Authors**
>
> *In this emerging field, there is some debate over what level and type of structural information each token should have… global information about how the residue is positioned in the protein…. By contrast ESM3's tokenizer does contain substantial information about residues which can be very distant… yet GeoBPE does better on global reconstruction tasks. This is perhaps surprising and I wonder if the authors could comment on it.*
>
> Thank you for this insightful question. As you wonderfully put it, GeoBPE offers a simpler and more intuitive break from the ESM3 led VQ-VAE paradigm. Our understanding on why our "simpler" method achieves better reconstruction is that they are inspired by different sources and fundamentally differ in objectives. VQ-VAE tokenizers descend from autoencoders: their aim is to learn a compact latent space of high-dimensional data and approximate the intractable posterior. This allows them to encode complex information like distant residues and global context into an informative space. VQ-VAE ``tokens” are discrete posterior codes in this latent space, rather than compressed representations of the original data; reconstruction becomes contingent on an artificial latent space. GeoBPE, meanwhile, directly manipulates a canonical representation. It is a continuous analog of BPE, which was originally a compression algorithm. Thus, GeoBPE can be understood as bit-rate reduction of geometric data while preserving the original content faithfully.
>
> *Generally the peptide backbone bond lengths are very close to fixed (in fact there are usually restraints on these when solving structures by X-ray diffraction, NMR, or Cryo-EM). I wonder whether these are actually needed in GeoBPE; do the authors find clusters of motifs where these are very different, or are the bond lengths similar between all motifs? My guess is that the clusters differ primarily in their backbone dihedral angles*
>
> Your guess is correct. We actually have a hyperparameter (App. K) `free_bonds` which decides whether bond lengths are free variables, or standardized to fixed values. Our experience toggling this parameter does show comparable distortion.
> Our results in Table 1 were actually obtained from a GeoBPE run with the latter setting (we omitted this detail, now added).
> In our experiment logs, we found two runs where only `free_bonds` differs. The one with free bonds achieves only *1.69\%* lower RMSD, which is negligible. We’ve commented on this in our documentation. Regarding bond angles, we indeed see very narrow histograms of their values. This is consistent with steric/chemical constraints that nearly determine the bond angles. Though fixing them is justified in principle, we chose to make them variable after we introduced learnable corrections. Our insight was they can help "absorb" accumulating error caused by dihedrals. Making the glue bond angles learnable can promote global reconstruction. To ensure they don't optimize to unphysical values, our quantizer $Q_{\theta}$ still enforces they stay between the two ends of the histogram. In Fig. 3, we show the bond angles --red arcs -- adjust slightly to correct for quantization error.
>
> Do let us know if you have other questions or thoughts! We greatly appreciate your feedback.
>
> GeoBPE authors

---

### Official Review · Reviewer_F5Hs · 2025-11-01

**Soundness:** 3
**Presentation:** 3
**Contribution:** 3
**Rating:** 8
**Confidence:** 3

**Summary:**

The paper “Protein Structure Tokenization via Geometric Byte Pair Encoding (GEOBPE)” proposes a hierarchical, geometry-aware tokenizer that discretizes protein backbones (N, Cα, C) into interpretable structural “words.” Inspired by text BPE, it merges frequent geometric motif pairs using clustering and inverse-kinematics “glue” corrections to preserve structural consistency. GEOBPE achieves superior compression–distortion tradeoffs, strong out-of-distribution generalization, and improved downstream task performance over VQ-VAE-based baselines. Its tokens align with functional protein motifs (e.g., CATH domains) and enable structure language modeling and backbone generation, offering a scalable and interpretable discrete representation of 3D protein geometry.

**Strengths:**

- The GEOBPE idea is elegant: motif clustering + quantization + glue angle correction giving both local motif reuse and global structural consistency, and iteratively building a vocabulary. That hierarchical tree/merge structure is meaningfully analogous to BPE and advantageous in representing protein structure at multiple resolutions.
- On compression vs distortion, GEOBPE traces a favorable Pareto front.
- Out-of‐distribution generalization appears very good compared to VQ-VAE
- Downstream tasks benefit from the tokenization
- The tokens learned are not just “latent vectors” but actual observed motifs (medoids) with real structure, and they show alignment with functional families (CATH) and interpretability in case studies.
- The authors highlight that they trained on much less data (~7% or even ~1% of typical size) and yet got strong results (Sec 5).

**Weaknesses:**

- The method involves many steps (clustering motifs, quantizing, global inverse kinematics to correct glue angles, iterative merging). It may be computationally heavy (they discuss complexity in Appendix H). The practicality at very large scale (hundreds of millions of structures) may still be challenging, and training time/compute cost details may be less emphasized.
- The method depends on choices: how to cluster, how many medoids (K), when to merge, how to set glue‐IK weights, how to quantize, etc. These may introduce sensitivity and hyperparameter tuning cost. The robustness across many protein fold types or extremely large or unusual proteins may vary.
- The method tokenizes only backbone geometry (ignores side-chains, ligands, post-translational modifications). For many functional tasks side-chains or ligand contacts matter a lot.

**Questions:**

- What is the wall-clock/time/compute cost to build the vocabulary (i.e., cluster motifs, compute glue corrections) for e.g., the full PDB (hundreds of thousands of chains)?
- How sensitive is the performance (compression vs distortion) to choices of K (number of medoids), merge iterations, and glue‐IK weights? Is there a guideline for selecting these hyperparameters for a new domain (e.g., membrane proteins or antibody loops)?
- The work focuses on backbone only. Do the authors envision an extension to include side-chain geometry (Cβ, other atoms) or ligand binding sites?

---

> ### Author Response · Authors · 2025-11-21
> **Response by GeoBPE Authors**
>
> **Thank you for your incredibly positive review and recognizing the comprehensive merits of GeoBPE!**
>
> *The method involves many steps (clustering motifs, quantizing, global inverse kinematics to correct glue angles, iterative merging). It may be computationally heavy (they discuss complexity in Appendix H)... What is the wall-clock/time/compute cost to build the vocabulary (i.e., cluster motifs, compute glue corrections) for e.g., the full PDB (hundreds of thousands of chains)?*
>
> **Re. complexity.** In our attached supplementary, you can find nearly all GeoBPE steps are parallelized (foldingdiff/bpe.py, glue_opt_all, _init_res_tokens, _cache_exit_frames, k_medoids, _assignment_worker, _quantize_worker, etc.) and optimized for performance. The few stateful routines (e.g., GeoPair-map updates) are not on the critical path and incur basically negligible cost.
>
> **Re. Walltimes.** Here is a summary from the logs of our paper’s run:
> Empirical walltime (20 cores) of GeoBPE steps from our original training run (on the 48k PDB data splits we adopt from [1]):
>
> |Function|Paper Reference (Algo, Line)|Time (HH:MM:SS)|(NCPU)|
> |-|-|-|-|
> |_init_thresholds|Algo 1 L1(Empirical Quantizer Estimation)|00:01:33|20|
> |_init_res_tokens|Algo 1 L2 (Per-residue Initialization)|02:16:02|20|
> |glue_opt_all|Algo 1 L3 (Global glue refinement)|03:21:50|20|
> |Step 1|Algo 1 L7 (Step)|00:11:20|20|
>
> We will add a more detailed breakdown into the paper.
>
> *The practicality at very large scale (hundreds of millions of structures) may still be challenging, and training time/compute cost details may be less emphasized…*
>
> **Re. Large-scale Practicality.** GeoBPE interleaves multi-threading on I/O-bound subtasks with standard multi-processing patterns on compute-bound tasks. Since we work with GeoBPE on a busy HPC cluster, we've optimized each step to be robust to interruptions and exploit caching, checkpoint frequently, and minimize overhead.
>
> **A New Large-Scale Tokenization Study** To stress test GeoBPE and address the practicality at large-scale, we downloaded the 550k Swiss-Prot set from AlphaFold DB [2] (>10x PDB files than our previous pretraining dataset) and benchmarked walltime to tokenize it all with 100 cores.
>
> **Results.** The average throughput was 191.32 files/min, so it took just around 2 days to finish. We observe runtime scales linearly to #cores (expected of parallelization).
>
> **Downstream Application.** With the Swiss-Prot dataset tokenized, we were able to rerun our SSLM protocol and train a $9$x larger model, which achieved a **4\% => 20.8\%** improvement in Designability. This shows GeoBPE not only supports, but can scale to large-scale modeling. We hope we can take our results further, provided the means.
>
> *The method depends on choices: how to cluster, how many medoids (K), when to merge, how to set glue‐IK weights, how to quantize, etc. These may introduce sensitivity and hyperparameter tuning cost…How sensitive is the performance (compression vs distortion) to choices of K (number of medoids), merge iterations, and glue‐IK weights? Is there a guideline for selecting these hyperparameters for a new domain (e.g., membrane proteins or antibody loops)?*
>
> **Re. Ablation study on performance sensitivity to K, #iterations, glue-IK.**
>
> *Glue-IK weights.* We wrote glue-IK weights (wR, wt) in Algo. 1 to generalize the IK loss, though we previously hard-coded them (1, 0.1) to obtain the results in the paper.
>
> We performed the following sweep over (wR, wt) (order of magnitude changes to wT/wR), with the remaining settings matching our paper’s $|V|=600$ tokenizer:
>
> |(wR, wt)|Train||CAMEO||CASP14||
> |-|-|-|-|-|-|-|
> |N=300|RMSD|LDDT|RMSD|LDDT|RMSD|LDDT|
> |(10, 0.1)|2.846|0.615|2.767|0.601|2.608|0.587|
> |(1, 0.1)|1.718|0.739|1.656|0.734|1.526|0.721|
> |(1.0, 1.0)|1.552|0.764|1.546|0.755|1.412|9.743|
> |(0.1, 1.0)|1.537|0.767|1.532|0.758|1.396|0.745|
> |(0.1, 10)|1.533|0.768|1.533|0.758|1.407|0.745|
>
> This shows wt controls performance (correct positions is more critical than correct orientations). There’s diminishing returns once $wt/wR\ge 1$ ($|\Delta_{LDDT}| \approx  1e-3, |\Delta_{RMSD}| \approx 1e-2)$),  (Incidentally, this means our paper’s Fig. 1 results were sub-optimal).
>
> (response continues in next reply)

---

> ### Author Response · Authors · 2025-11-21
> **Response by GeoBPE Authors**
>
> **K (num_p) and #iterations.** You’ve correctly identified the two most important hyperparameters of GeoBPE, since they determine control vocabulary growth (per-step and #steps).
>
> We can find experiments differing only by these two hyperparameters in our Experiment Logs (see below).
>
> Note 1: The control is usually $|V|$, which accounts for both `num_p` and #iterations. It allows us to more easily compare different runs with differing `num_p`.
>
> Note 2: We define `num_p` to follow a step schedule. For example, `{2:2, 3:5, 5: 10}` (passed as `--num-p 2-2:3-5:5-10`) means to introduce $2$ tokens for geo keys with $2$ bonds (C-terminus residue orientations), $5$ for geo keys with $3$ bonds (all non-terminus residues), $10$ for all merged geo keys with $5$ or more bonds (every GeoBPE step after residue initialization).
>
> Briefly, we share a relevant cluster of runs with varying settings to `num_p`. I added incremental statistics to see how RMSD/LDDT/BPR changes with the iteration.
>
> |num_p|\|V\||rmsd|lddt|bpr|
> |------------------------------------------|-------|-------------|--------------|-------------|
> |{2: 100, 3: 500, 5: 20, 6: 100, ...}|600|1.656926385|0.729722218|36.02222013|
> |{2: 500, 3: 2000, 5: 100, 6: 500, ...}|2500|1.414444334|0.7508450139|41.10996177|
> |{2: 1000, 3: 5000, 5: 200, 6: 1000, ...}|6000|1.37273265|0.7560054117|45.4400983|
> |{2: 1000, 3: 20000}|21000|1.212136861|0.7648766102|47.61765477|
> |{2: 50, 5: 20, 6: 100, ...}|5200|2.114771369|0.6750034882|37.23970995|
> |{2: 10, 3: 50, 5: 1, 6: 5, 8: 1}|65|1.77818163|0.71001117|30.80628808|
> ||237|1.77352635|0.70339111|33.99630606|
> ||388|1.72190544|0.69976154|35.87554165|
> ||521|1.7080282|0.7001499|37.33429966|
> ||631|1.69103167|0.69967895|38.4688073|
> ||739|1.70724237|0.69611838|39.54225799|
> ||845|1.725974273|0.69296629|40.56836822|
> |{2: 2, 3: 5, 5: 1, 6: 2, 8: 1}|109|3.96104103|0.53499012|27.26012218|
> ||309|4.07446509|0.53422586|30.81214168|
> ||508|4.23385662|0.53086872|33.45501564|
> ||707|4.504176401|0.5299156411|35.93496264|
>
>
>
> 1. As `num_p` values become high, the unique GeoPair keys increase exponentially. Since each iteration only looks at one key, the #merges done falls off. Empirically, the top rows show only marginal changes to distortion as iteration # increases. We omitted them for brevity.
> 2. As `num_p` values drop too low, GeoBPE becomes more of a coarsening algorithm (lots of merges, repetitive patterns are preserved but higher frequencies are lost). When merges happen more often, more drift is introduced, so error quickly accumulates. We can see distortion monotonically increase for the last run.
> 3. In the middle is where the relationship becomes *really interesting*. There exists a tension between `num_p` and merge frequency, but glue opt is still potent enough to manage drift accumulation. We see error decrease before increasing again, when eventually merges overwhelm.
>
> We have more insights and recommendations on the intricate interplay between these parameters in the new Guidelines doc (below).
>
> **Re. choices.** While GeoBPE offers flexibility, we understand guidelines, documentation & sensitivity ablations are essential to streamline adoption.
> Thus, we’ve added the following in the revision:
> - **Significantly expanded App. K** on which hyperparameters matter most, their defaults, intuition behind their control of GeoBPE behavior, and recommendations & guidelines for each use case.
> - **Sensitivity Ablations** on the key hyperparameters, supporting our sensitivity estimates (below).
>
> We also added the following in the supplementary materials:
> - **GeoBPE API and Usage Guidelines Doc** with names, descriptions, default values and sensitivity estimates for each noteworthy hyperparameter (inspired by scikit-learn documentation). We also include custom visualizations/tables and examples from real experiments we ran to share our intuition.
> - **Experiments logs** (anonymized, of course) cataloging our runs, varied hyperparameter settings, and performance. We hope this can aid users of GeoBPE to save time and learn from our experience.
>
> (response continues in next reply)

---

> ### Author Response · Authors · 2025-11-21
> **Response by GeoBPE Authors**
>
> *The robustness across many protein fold types or extremely large or unusual proteins may vary.*
>
> **Re. robustness across protein types.** Thank you for this comment, which prompted us to conduct an interesting analysis. We used FoldSeek to TM-align our PDB test set, and use 1-max(TM score) as a proxy for unusualness. In App. I, we’ve added extensive figures and analysis on tokenizer performance against unusualness, fold types (CATH class/arch), and size. We observe, surprisingly, GeoBPE suffers no degradation on unusual (low TM score against FoldSeek scan of PDB) chains
>
> *The method tokenizes only backbone geometry (ignores side-chains, ligands, post-translational modifications). For many functional tasks side-chains or ligand contacts matter a lot…. The work focuses on backbone only. Do the authors envision an extension to include side-chain geometry (Cβ, other atoms) or ligand binding sites?*
>
> Absolutely! It is well-known that side-chain geometry can be coded as lists of named chi angles (well-known lookup from AA type). These can be interleaved into the backbone angles (N, Ca, chi1, chi2, …, C). The representation remains linear fundamentally, but becomes sensitive to context-the length of this intermission is set by the AA type. This introduces semantic constraints to GeoBPE-bond-residue motifs can only be compiled for certain (i, j) pairs-which requires more engineering choices (e.g. do we initialize modes at rotamer-level, do we allow merges that spill over from backbone to side chains, are side-chain motifs counted the same as backbone motifs). To maintain generality and allow users to make such choices, we present GeoBPE as backbone-only, but these are both next steps for us and future users of GeoBPE!
>
> [1] Protein Structure Tokenization: Benchmarking and New Recipe. https://arxiv.org/abs/2503.00089
>
> [2] AlphaFold Protein Structure Database. https://alphafold.ebi.ac.uk/download

---

### Official Review · Reviewer_3EbP · 2025-11-05

**Soundness:** 3
**Presentation:** 1
**Contribution:** 3
**Rating:** 6
**Confidence:** 3

**Summary:**

This paper proposes GEOBPE, a geometric analog of byte-pair encoding (BPE) for protein backbones. Instead of tokenizing amino acid sequences, GEOBPE tokenizes 3D structures into discrete motifs, building a hierarchical vocabulary through iterative clustering and merging of “Geo-Pairs.” And it enforces geometric consistency using rigid-body optimization during BPE process. This method does not reply on neural networks to "automatically" learn the discretized tokens like VQ-VAE-like protein structure tokenizers.

GEOBPE supports downstream applications like function prediction and shows strong interpretability by aligning the tokens with CATH functional families. The tokenizer is also used to train generation tasks, though it's not the focus on this paper's experiment section.

**Strengths:**

- **Novel conceptual framing**: this paper continues an "old fashion" way to heuristically tokenize structures. GEOBPE extends BPE from linguistic to protein structure data, introducing a hierarchical, interpretable tokenizer for protein structures. This is process is far complicated than applying BPE on protein sequence data. And the authors seem to design a way to solve it.

- **Interpretability**: A highlight difference for GEOBPE with popular VQ-VAE-like structure tokenizers is the interpretability: (1) the vocabulary construction process is interpretable, unlike VQ-VAE relies on automatically learning from data; (2) the hierarchical vocabulary offers intuitive geometric primitives that align with known structural motifs in CATH.

- **Empirical strength**: As shown in Table 1, tokens from GEOBPE contain rich functional information, and performs the best on related benchmarking tasks.

**Weaknesses:**

- **Per-residue task ambiguity**: Since tokenization is no longer residue-level, how can we make GEOBPE's tokens aligned for per-residue benchmarks? This is a crucial design and evaluation gap.

- **Severe Presentation issues**: Excessive information density makes the paper difficult to parse; algorithmic pseudocode dominates over conceptual clarity. And referring readers to appendix for very detailed algorithms in appendix and replying on these algorithms to inform what is happening inside the method is a very bad way to deliver ideas. The paper would benefit from improved figure explanations and appendix referencing.

**Questions:**

1. Will all the code be released?

2. Line 150 has notation conflicts with Line 161, \phi_{i+1} should be \phi_{i}?

3. Even though GEOBPE can trained with only 1% PDBs to get good performance. The algorithm complexity seems to be high and prevents GEOBPE to scale further to large scale datasets (current training data is around 48k proteins). Could you please analyze the complexity of your algorithm and the empirical time used for GEOBPE?

4. Suggestion to modify Figure 2: based on my understanding, the top-right two motifs in “step 1” subfigure stand for “prototypes” and the arrows mean that each motif is mapped to those prototypes. And then this pair of prototype is selected to be merged in BPE process.

5. How does GEOBPE tokenize a new unseen structure once the vocabulary is learned? Is there a decoding or segmentation step described explicitly? Not sure if I missed anything.

6. Concerns for GEOBPE to generate good structures when coupled with LMs. For example, in ProteinAE [1] Table 2, they can perform well on "unconditional protein backbone generation" when compared with competitive.

---

> ### Author Response · Authors · 2025-11-21
> **Response by GeoBPE Authors**
>
> **Thank you for recognizing the novelty, interpretability and empirical strength of our work!**
>
> *Per-residue task ambiguity: Since tokenization is no longer residue-level, how can we make GEOBPE's tokens aligned for per-residue benchmarks? This is a crucial design and evaluation gap.*
>
> Thanks for your question. GeoBPE emits not just final tokens (which are no longer residue-level), but also a hierarchical merge tree ($\mathcal{F}$) connecting individual residues to higher-level substructures learned by GeoBPE. We leverage the rich, hierarchical inductive bias $\mathcal{F}$ provides to selectively broaden the receptive field of residues. This does not affect applicability on residue-level tasks or their evaluation (11/12 of our tasks are residue-level). It is a GeoBPE-guided enhancement of residue-level embeddings.
>
> More elaboration on how GeoBPE benefits residue-level tasks:
>
> - GeoBPE-Transfer is a Up-Down Tree LSTM that propagates residue-level base features (from ESM3) up merge nodes, aggregates them into virtual root, then back down to the residue (leaf) level.
> - Our Table 1 results show a significant boost on residue-level tasks than from base residue-level features alone (see vs ESM3 numbers in Table 1).
> - We revised the final paragraph of 3.2 to make it clear the final embeddings are still residue-level for per-residue benchmarks.
>
> *Severe Presentation issues: Excessive information density makes the paper difficult to parse; algorithmic pseudocode dominates over conceptual clarity. And referring readers to appendix for very detailed algorithms in appendix and replying on these algorithms to inform what is happening inside the method is a very bad way to deliver ideas. The paper would benefit from improved figure explanations and appendix referencing.*
>
> Thank you for the detailed suggestions—this feedback was extremely helpful.
>
> We have expanded Method sections 3.1 and 3.2 with *more intuitive* explanations by motivating GeoBPE from BPE, discussing the key challenges our components are designed to overcome.
>
> We also made several targeted revisions to improve clarity through figure referencing:
> 1. **New Fig. 2** illustrating notation and internal-angle parameterization.
> 2. **Expanded algorithm figure caption (now Fig. 3)** with synchronized styling between figure and caption, using bordered/filled rectangles to guide visual parsing.
> 3. **Enhanced hierarchy visualizations (in Appendix G)** to match the diagram in Fig. 1.
>
> We believe these changes substantially improve the readability and conceptual accessibility of the method.
>
> *Will all the code be released?*
>
> Yes! We've uploaded GeoBPE (fully anonymized, self-contained) in the supplementary material, including installation instructions and example workflows. Please see our general comment.
>
> *Line 150 has notation conflicts with Line 161, \phi_{i+1} should be \phi_{i}?*
>
> Good catch! We fixed all occurrences of this inconsistency.
>
> *Even though GEOBPE can trained with only 1% PDBs to get good performance. The algorithm complexity seems to be high and prevents GEOBPE to scale further to large scale datasets (current training data is around 48k proteins). Could you please analyze the complexity of your algorithm and the empirical time used for GEOBPE?*
>
> **GeoBPE learning complexity.** We analyzed GeoBPE’s complexity in App. J, with insights on how to make it efficient in practice. The big-O formulas reveal two bottlenecks-$M_{\max}$ (#medoids) and $P$ (periodic IK)-due to $M_{\max}^2$ and $\frac{T}{P}N\log N$ terms. The final ablation in App. A supports a modest $M_{\max}$ is sufficient; our results did not sacrifice performance by setting P=10. The 1% ablation you mentioned suggests a modest $N$ is sufficient to train a good GeoBPE tokenizer.
>
> **Scalability to large datasets**. To directly address scalability at inference time, we conducted a new large-scale experiment: using 100 CPU cores, we tokenized the entire 550k-structure SwissProt subset of AF-DB (>10x the dataset used in our main experiments).
> Throughput averaged 191.32 structures/min, finishing in just over 2 days, with wall-time scaling linearly with number of cores (embarrassingly parallel).
>
> **Parallel implementation notes.** In our attached supplementary, you can find nearly all GeoBPE steps are parallelized (foldingdiff/bpe.py, glue_opt_all, _init_res_tokens, _cache_exit_frames, k_medoids, _assignment_worker, _quantize_worker, etc.) and optimized for performance. Non-parallelized (stateful) routines like updating the GeoPair map are not bottlenecks in walltime.
>
> Empirical walltime (20 cores) of GeoBPE steps from our original training run:
>
> |Function|Paper Reference (Algo, Line)|Time (HH:MM:SS)|(NCPU)|
> |-|-|-|-|
> |_init_thresholds|Algo 1 L1(Empirical Quantizer Estimation)|00:01:33|20|
> |_init_res_tokens|Algo 1 L2 (Per-residue Initialization)|02:16:02|20|
> |glue_opt_all|Algo 1 L3 (Global glue refinement)|03:21:50|20|
> |Step 1|Algo 1 L7 (Step)|00:11:20|20|
>
> (response continues in next reply)

---

> > ### Author Response · Authors · 2025-11-21
> > **Response by GeoBPE Authors**
> >
> > *Suggestion to modify Figure 2: based on my understanding, the top-right two motifs in “step 1” subfigure stand for “prototypes” and the arrows mean that each motif is mapped to those prototypes. And then this pair of prototype is selected to be merged in BPE process.*
> >
> > Thank you—your interpretation was exactly right. We’ve incorporated your suggestion with hand-drawn icons and modified the figure (now Fig. 3) to clarify the difference between motif occurrences and prototypes. We hope our changes visually communicate these concepts better.
> >
> > *How does GEOBPE tokenize a new unseen structure once the vocabulary is learned? Is there a decoding or segmentation step described explicitly? Not sure if I missed anything.*
> >
> > Thanks for your attention to detail and noticing this omission. We added **Algo. 10** to explicitly describe how GeoBPE tokenizes a new structure (a similar procedure as training, but skipping the one-time initializations and applying the learned vocabulary instead of popping frequent GeoPairs; this is analogous to how BPE would traverse the byte lookup table). We also edited the main text to reference it appropriately.
> >
> > *Concerns for GEOBPE to generate good structures when coupled with LMs. For example, in ProteinAE [1] Table 2, they can perform well on "unconditional protein backbone generation" when compared with competitive.*
> >
> > Thank you for pointing us to ProteinAE [1]; it is a strong and relevant model. However, a direct comparison with our GeoBPE+SSLM results **would not be meaningful** due to differences in goals, scale, and modeling assumptions:
> >
> >
> > 1. **Different objectives.** ProteinAE is a continuous latent-space autoencoder designed to support high-capacity generative modeling (e.g., PLDM). GeoBPE, in contrast, is a tokenizer—its purpose is to learn discrete, compositional vocabularies for downstream tasks. We compare with VQ-VAEs (like ESM3) as tokenizers because they quantize the latent space into a codebook, producing discrete sentence-like codes of structure. ProteinAE removes this quantization bottleneck and so is no longer a tokenizer but a general autoencoder guided by bottleneck principles.
> >
> > 2. **Role of PLDM.** Backbone design in [1] is done by PLDM, a large, dedicated generative model. ProteinAE supports PLDM by learning a latent space, but PLDM is a large-scale, standalone 200M generative model trained on 500K AF structures, and one of their central contributions.
> >
> > 3. **Downstream Task Performance.** GeoBPE supports SOTA residue & protein-level supervised tasks via transferring a rich, hierarchical vocabulary. In ProteinAE [1] Table 3, they report numbers for only 2 of 12 benchmarking tasks- GeoBPE matches performance on one and achieves 27% higher $\rho$ on the other.
> >
> > 4. **Purpose of SSLM.** Our **Small** SLM protocol is minimalistic by design (7.3M vanilla Transformer trained with 48k experimental structures). The experiment was designed to compare GeoBPE indirectly to VQ-VAEs. With the same compute/architecture, a small Transformer trained on GeoBPE tokens leads to significantly better results than VQ-VAE tokens (the current de-facto large-scale tokenizer). We believe our preliminary SSLM results show GeoBPE has *potential* to replace VQ-VAE as the tokenizer in ESM3-scale models, not a mature generative model solution. Thus, we’ve revised Sections 4, 5 and App. E to state the aims of SSLM eval more clearly. Additionally, we have **stronger Designability** results now. Please see our response to MhKe for more details.
> >
> > We hope this makes the relationship between our work and ProteinAE clearer.
> >
> > Should there be further questions or suggestions, please let us know!

---

### Official Review · Reviewer_MhKe · 2025-11-05

**Soundness:** 3
**Presentation:** 3
**Contribution:** 3
**Rating:** 6
**Confidence:** 3

**Summary:**

This paper introduces Geometric Byte-Pair Encoding (GeoBPE), a novel algorithm for protein structure tokenization. The authors identify key limitations in existing protein structure tokenizers, particularly the popular VQ-VAE variant. They argue these methods suffer from fixed token sizes that prevents multi-scale resolution, and produce vector-based tokens that lack the interpretability and hierarchical structure of methods like Byte-Pair Encoding (BPE) in natural language processing. GeoBPE is designed as a geometric analog to BPE. It works by iteratively building a hierarchical vocabulary of structural motifs, which involves: iteratively identifying the most frequent pairs of structural motifs and clustering their occurrences using k-medoids, then replacing these occurrences with a representative prototype from the new vocabulary. This local quantization inevitably introduces geometric errors, and the authors correct this by optimizing the boundary glue angles through differentiable inverse kinematics under SE(3) loss. Experimental results demonstrate that GeoBPE achieves high compression, highly data-efficient, and generalizes well to unseen structures. Furthermore, its tokens are shown to be interpretable, aligning with CATH functional families.

**Strengths:**

1. GeoBPE produces a hierarchical vocabulary of discrete structural motifs, which is a significant advantage as the resulting tokens have functional meaning. Experimental results show that the tokens align with CATH functional domain annotations and are validated by expert case studies.
2. GeoBPE addresses the fixed token size limitation of VQ-VAE. Its hierarchical BPE-like approach allows it to capture multi-scale features, and its performance can be tuned to trade compression (BPR) for distortion (LDDT/RMSD), which other tokenizers with fixed dimensions cannot do.
3. The paper's explicit handles the geometric drift. The use of differentiable inverse kinematics to optimize glue angles is a principled solution to the problem of maintaining global structural integrity while performing local quantization. Ablation study shows this step is essential, as omitting it causes reconstruction RMSD to jump from 1.66 to 4.39.
4. GeoBPE shows excellent out-of-distribution generalization. It maintains a test/train distortion ratio of 1.16-1.28, whereas the VQ-VAE baseline reportedly degrades by as much as 6.4x on the same test sets.

**Weaknesses:**

I am not very familiar with structural biology and thus may not be able to provide deep insights into the algorithmic details of GeoBPE. My primary concerns are focused on the experimental validation and its implications:
1. The SSLM experiments (Table 7) indicate that the language model trained on the GeoBPE vocabulary struggles significantly with unconditional generation. This is evidenced by the low scTM (less than 0.3) and designability (less than 5%). Performance at this level is insufficient for most practical applications. This finding leads to serious concerns about using GeoBPE as a foundational component for practical development (e.g., for building multimodal structure and sequence language models, like ESM3).
2. The paper demonstrates GeoBPE's high data efficiency, with appendix results showing that using only 1% of the training data will not introduce performance loss. While this is an advantage, it could also be interpreted as a potential risk. This finding might suggest that the model has a low capacity and hits a performance ceiling almost immediately, making it incapable of capturing the complex, subtle, and diverse structural patterns that can only be learned from large-scale datasets.

Given these points, I hope the authors will provide a detailed discussion about these concerns.

**Questions:**

See above weaknesses.

---

> ### Author Response · Authors · 2025-11-21
> **Response by GeoBPE Authors**
>
> **Thank you for recognizing the technical innovations and distinctive features of GeoBPE!**
>
> *The SSLM experiments (Table 7) indicate that the language model trained on the GeoBPE vocabulary struggles significantly with unconditional generation... low scTM (less than 0.3) and designability (less than 5%) insufficient for most practical applications… leads to concerns about using GeoBPE as a foundational component for practical development.*
>
> **Clarifications on purpose of SSLM eval (revised in App. E).**
> The goal of SSLM was to create a small, isolated protocol for gauging the *potential* of language model integration, not to ship a competitive generative model. Its design is intentionally minimalistic--we fixed a small 7.3M params architecture and trained on 40k PDB structures with no parameter tuning. Our goal was only to obtain a relative ranking among PSTs, not to report absolute performance. What we showed is: under the same resources, setup & conditions (data, compute, model architecture), training with GeoBPE tokens leads to **significantly better** results than **VQ-VAE tokens** (the current de-facto large-scale tokenizer).
>
> **GeoBPE is orthogonal to LM and data scaling.**
> GeoBPE should be evaluated as a tokenizer, not a generative model. If using large-scale datasets & model architectures, we are confident GeoBPE+LM will be a competitive model. However, you have every right to be skeptical, given the toy nature of SSLM. To close the gap without overstepping the scope of the paper, we spent the last 2 weeks **scaling up SSLM eval** by 10x along both the data and model dimension.
>
> **This led to a significant improvement in designability after 10x data & model scale up: 4% $\Rightarrow$ 20.8%.**
>
> *Setup.* We downloaded the 550K Swiss-Prot dataset from AlphaFold DB [1] and increased our Transformer to ~10x parameters (7.3M $\Rightarrow$ 65.9M). The rest of SSLM remains the same (App. E).
>
> *Evaluation.* We used the **same evaluation protocol (50-128 AAs, 10 each, OmegaFold for scTM)** of ProtDiff & FoldingDiff [3, 4].
> Note: Our previously reported 4% designability did not follow this protocol; we sampled from the size prior; the average generated protein was ~214 AAs.
>
> *Results*. GeoBPE+SSLM achieves an average scTM of 0.4051, with 20.8% being scTM > 0.5. This is notably **higher than ProtDiff (11.8%) and FoldingDiff (14.2%)**. Uniqueness/diversity also remain high (76.4% and 0.73). We have explained the protocol and written about these new results in the new Appendix section F. We hope this addresses your concern!
>
> *The paper demonstrates GeoBPE's high data efficiency, with appendix results showing that using only 1% of the training data will not introduce performance loss. While this is an advantage, it could also be interpreted as a potential risk. This finding might suggest that the model has a low capacity and hits a performance ceiling almost immediately, making it incapable of capturing the complex, subtle, and diverse structural patterns that can only be learned from large-scale datasets.*
>
> **Clarification on the 1\% result.** Thank you for probing this study further. We want to clarify that the setup for the ablation was: use 1% of pretraining data to learn the tokenizer (Algo. 1), then evaluate the *downstream transfer* task suite from Table 1. The data-efficiency pertains to downstream transfer ("how useful the vocabulary is"), not the *performance* (compression/distortion) metrics. However, the way we stated the conclusion can be ambiguously read as a general claim on GeoBPE, and we have revised that section to make it clear what the ablation was.
>
> **Connection between GeoBPE and downstream tasks.**
>
> Even if using more data to learn a tokenizer does not lead to better downstream transfer, the downstream task model (leveraging GeoBPE's vocabulary) **should not be limited in capacity**. In addition to the LM scaling study above, we added another ablation study to App. A on the data-efficiency of the *downstream transfer* model. GeoBPE hierarchies are used to learn residue and protein-level representations (see revised Sec. 3.2), a setting where we expect complex, subtle and diverse patterns to emerge from training on more data.
>
> *Setup*. Fixing the same val/test splits as in Table 1, we vary % of the task training set used from 20-40-60-80. We fix the same $|V|=600$ GeoBPE tokenizer checkpoint used throughout the paper.
>
> **Ablation Results**.
>
> (response continues in next reply)

---

> ### Author Response · Authors · 2025-11-21
> **Response by GeoBPE Authors**
>
> |Task|Split|GeoBPE-Transfer|||||
> |-|-|-|-|-|-|:-:|
> |||20\%|40\%|60\%|80\%|100\%|
> |**Functional Site Prediction (AUROC\%)**|||||||
> |BindInt|Fold|58.78|61.18|59.79|59.45|59.19|
> ||SupFam|89.55|89.88|90.42|90.71|91.31|
> |...middle 6 tasks omitted for brevity...|||||||
> |Ept|Fold|63.63|53.28|61.72|61.66|64.78|
> ||SupFam|71.21|49.39|73.59|76.01|77.06|
> |Average AUROC\%||79.19|77.42|79.93|79.79|**80.20**|
> |**Physicochemical Property Prediction (Spearman’s $\rho$\%)**|||||||
> |FlexRMSF|Fold|41.49|43.33|38.48|39.02|40.89|
> ||SupFam|34.74|45.41|44.61|47.06|47.17|
> |FlexBFactor|Fold|23.90|27.86|33.83|34.82|37.28|
> ||SupFam|23.80|25.46|37.70|36.49|35.61|
> |FlexNEQ|Fold|54.40|56.07|55.98|57.53|56.65|
> ||SupFam|51.12|53.77|52.52|54.96|53.98|
> |Average $\rho$\%||38.24|41.98|43.85|44.98|**45.26**|
> |**Structure Property Prediction (Macro F1\%)**|||||||
> |Homo|Fold|14.35|20.55|23.74|24.25|23.60|
> ||SupFam|27.63|35.11|43.09|43.98|47.28|
> ||Fam|65.79|73.66|80.77|82.04|85.75|
> |Average Macro F1\%||35.92|43.11|49.20|50.09|**52.21**|
>
> - For physicochemical (residue-level regression) and fold-level tasks, the model indicates a clear propensity for more training data, with step-wise gains for every 20% of training data. 20% $\Rightarrow$ 100\% training data sees performance lift significantly: $38.24\rightarrow 45.46$, $35.92\rightarrow 52.21$.
> - For residue-level classification tasks, the lift is only marginal. We hypothesize the cause is not limited capacity, but rather that the tasks are localized label predictions; a residue-level receptive field is sufficient when features are informative.
> - This shows GeoBPE’s data-efficiency at learning a vocabulary does not limit its capacity on downstream tasks that require multi-scale resolution (e.g. structural flexibility or fold-level classification), which existing fixed-size tokenizers cannot
>  at both residue and structure-level.
>
> We have included this study (with full results) right after the 1% tokenizer training one in App. A, clarifying the data-efficiency strengths (or limitations) of GeoBPE vocabulary learning is orthogonal to data-scaling the downstream ``model”. Many tasks seeing large gains in performance with more training data, learning complex, hierarchical structural patterns that can only be learned from large-scale datasets.
>
> We renamed GeoBPE in downstream transfer experiments to GeoBPE-Transfer. We apologize for any confusion caused regarding what the “model” is. Also, the GeoBPE average computes out to be 80.20 (not 79.93), which is now fixed. We apologize for the slip.
>
> Should there be further questions or suggestions, please let us know!
>
> [1] AlphaFold Protein Structure Database. https://alphafold.ebi.ac.uk/download
>
> [2] Diffusion probabilistic modeling of protein backbones in 3D for the motif-scaffolding problem. https://arxiv.org/abs/2206.04119
>
> [3] Protein structure generation via folding diffusion. Nature Communications.

---

### Author Response · Authors · 2025-11-21
**GeoBPE code package is ready to use!**

Dear reviewers,

The GeoBPE code package has been attached in the supplementary zip (`iclr-anon-pkg`). It contains a self-contained artifact which is ready to use. Full instructions to reproduce our results are in the README.md. Feel free to run a quick smoke test or dive into the details of implementation. We also include the following additions:

- **GeoBPE API and Usage Guidelines Doc** (`docs/hparams_guide.md`) with descriptions, intuitions, and guidelines on how to effectively and efficiently use GeoBPE.
- **Experiment Logs** (`GeoBPE-logged-runs.pdf`) cataloging our past experiments varying hyperparameters settings. Quickly lookup settings & performance to save future iteration time.

Let us know if you run into any issues or questions.

Best,

GeoBPE authors

---

### Author Response · Authors · 2025-11-21
**Responses posted and new result**

Dear reviewers (MhKe, 3EbP, F5Hs, 4qDK),

We are incredibly happy to see many merits of GeoBPE be acknowledged, including (1) Originality & Novelty (4qDK, MhKe, 3EbP, F5Hs) (2) Interpretability & Functional Significance (MhKe, 3EbP, F5Hs), (3) Compression & Performance (4qDK, F5Hs, MhKe), (4) Downstream Transfer (3EbP, F5Hs, 4qDK), (5) Structural Hierarchy (MhKe, 3EbP, F5Hs, 4qDK), (6) OOD Generalization (MhKE, F5Hs), (7) Adaptive/Multi-Scale Resolution (MhKe, F5Hs) and (8) Data-Efficiency (F5Hs).

We have also posted point-by-point responses to your reviews, with additional ablations on hyperparameters, empirical results, analyses, and clarifications.

We’d like to share a latest result of ours: we find a **10x increase in data & model parameter count** of our Small Structure Language Model (SSLM) led to a **major improvement in backbone design metrics (<5% => 21% designable)**. This addresses concerns around the practicality of integrating GeoBPE with large models. We discuss this result more carefully in our individual responses, clarifying the purpose and scope of SSLM and of GeoBPE.

Finally, we hope each reviewer can read the rebuttals given to other reviewers, as some points may have been addressed elsewhere.

We look forward to hearing back from you. Let us know if there are any more questions/concerns.

Best,

GeoBPE authors

---

### Author Response · Authors · 2025-12-03
**Message to Area Chair**

Dear Area Chair,

We understand your time is limited, so we wanted to summarize the proceedings on this forum for you.

## Initial Reviews

Reviewers acknowledged the many contributions of GeoBPE (**Average Score of 7.5**), including:

*Originality & Novelty* (4qDK, MhKe, 3EbP, F5Hs), *Interpretability & Functional Significance* (MhKe, 3EbP, F5Hs), *Compression & Performance* (4qDK, F5Hs, MhKe), *Downstream Transfer* (3EbP, F5Hs, 4qDK), *Structural Hierarchy* (MhKe, 3EbP, F5Hs, 4qDK), *OOD Generalization* (MhKE, F5Hs), *Adaptive/Multi-Scale Resolution* (MhKe, F5Hs) and *Data-Efficiency* (F5Hs).

## Our Responses and Revisions

Reviewers also raised some weaknesses/questions, which we have addressed point-by-point in our responses to them and revised into the manuscript.

The current revision includes **15 additional pages of new figures, tables, analyses, and text** since initial submission.

We compiled **every concern raised in their initial reviews**, and documented the **specific changes** we made to the manuscript to resolve each of those concerns. Below we list the specific changes, tagged by reviewer ID.

## A. Main Paper – Changes to Core Text and Sections

### A1. Section 3: Method & Notation

- [3EbP, 4qDK] Expanded the description in Section 3.2 for the conceptual story, especially to motivate GeoBPE's core components from its BPE roots.
- [3EbP, 4qDK] Added **notation figure** (new Fig. 2) which includes all major symbols used in Section 3, and that the text references this figure when introducing notation.
- [3EbP, 4qDK] Defined core GeoBPE notations properly in new notation paragraph in 3.2
- [3EbP, 4qDK] Enumerated key notations at the start of Algorithmic Details (Appendix M) needed to understand mathematical algorithmic descriptions, ensuring key symbols are defined before use
- [3EbP, 4qDK] Corrected any lingering typos including the specific conflict where $\phi_{i+1}$ should be $\phi_i$.
- [3EbP] Revised the **final paragraph of Section 3.2** to explicitly state that the final representations GeoBPE outputs are **residue-level embeddings**, clarifying a key misconception
- [3EbP] Added a higher-level summary of the hierarchical computation of final representations
  - GeoBPE emits a **hierarchical merge tree $\mathcal{F}$**
  - GeoBPE-Transfer aggregates residues to higher-level intermediate motifs, then back down to residue-level leaf nodes
  - This procedure broadens receptive fields while final outputs remain residue-level

### A2. Downstream Transfer Model Naming & Interpretation

- [3EbP] Consistently rename the downstream transfer model to **“GeoBPE-Transfer”** wherever it appears (figures, tables, captions, and text) to preempt future misconceptions.

### A3. Sections 4 & 5: Experiments and SSLM Evaluation

- [MhKe, 3EbP] Revised Sections 4 and 5 to clearly state the **purpose of the SSLM evaluation**: a small, controlled protocol for *relative* comparison between tokenizers, not a full-scale generative model.

### A4. Limitations

- [MhKe, 3EbP] Explicitly describe GeoBPE as a **tokenizer**, orthogonal to language model architecture and data scale; clarify that generative performance depends on both the tokenizer and a sufficiently large LM/data.
- [MhKe, 3EbP] “Walk back” or soften any language implying that the current SSLM setup is itself a state-of-the-art generative model; instead emphasize that results are **preliminary evidence** of GeoBPE’s promise as a tokenizer for larger PLMs.
- [F5Hs] Describe a straightforward extension of GeoBPE to incorporate side-chains and amino acid identities to the internal angle formulation and vocabulary domain

---

## B. Main Paper – New and Revised Figures and Tables

### B1. Method Figures

- [3EbP, 4qDK] Added a figure illustrating **notation and internal angle parameterization** (referred to as new Fig. 2); ensure all core symbols appear here and are referenced in the text.
- [3EbP] Updated the **algorithm figure** caption (now Fig. 3) to:
  - Use bordered/filled rectangles and consistent styling to make the flow easier to parse.
  - Clearly differentiate **motif occurrences** vs **prototypes** (e.g., with icons or distinct shapes).
  - Synchronize the caption with the visual elements (explicitly referencing the shapes/icons the reviewer asked about).

### B2. Reconstruction / Compression Figure (Fig. 1)

- [F5Hs] Updated **Fig. 1** to reflect improved glue-IK weight settings $(w_R, w_t)$, since results from the ablation F5Hs suggested shows that the previous $(1, 0.1)$ setting is suboptimal.
- [F5Hs] Made sure the caption points to App. K, which notes that the revised figure uses the improved weight choice discovered during the ablations.

---

> ### Author Response · Authors · 2025-12-03
> **Message to Area Chair (continued)**
>
> ### B3. Table 1 – Reconstruction Metrics & Bond-Length Detail
>
> - [4qDK] In the caption of Table 1, link GeoBPE results to correct set of hyperparameters, with **standardized bond lengths** (`free_bonds = False`).
> - [4qDK] Mentioned in the new section on Sensitivity Ablations that toggling `free_bonds` to allow free bond lengths only yields a small RMSD change (~1.69%), as detailed in the appendices/experiment logs.
>
> ### B4. Downstream Transfer Table(s)
>
> - [MhKe] Corrected the **GeoBPE average AUROC** in the downstream table from 79.93 to **80.20** (task numbers correct; we only slipped on the averaging).
>
> ### B5. SSLM Evaluation Table (Table 9)
>
> - [MhKe, 3EbP] Added the **scaled-up SSLM results** to Table 11, using the ProtDiff/FoldingDiff-style protocol:
>   - Model: Transformer scaled from 7.3M → **65.9M** parameters.
>   - Data: **550k Swiss-Prot** structures from AlphaFold DB (10× original SSLM data).
>   - Protocol: sequence lengths 50–128 AAs, OmegaFold for scTM, 10 samples each.
>   - Results: average scTM ≈ **0.405**; designability **21%** (scTM > 0.5)
>   - Comparison: designability higher than ProtDiff (11.8%) and FoldingDiff (14.2%).
> - [MhKe, 3EbP] Added a note explaining that the earlier ~4–5% designability figure used a **different length prior** (size prior with mean ~214 AAs) and therefore is not directly comparable to above baselines.
>
> ---
>
> ## C. Appendices – Content Changes and Additions
>
> ### C1. Appendix A - 2 New Data Efficiency Ablations
>
> - [MhKe] In the accompanying text and new caption of Table 3, we clarified the misattribution of GeoBPE-Transfer results to GeoBPE (the tokenizer):
>   - The **downstream transfer model benefits significantly** (GeoBPE-Transfer) from more labeled data (e.g., AUROC and F1 improving from 20% → 100% training data).
>   - There is no evidence that **GeoBPE-Transfer** is capacity-limited by its 1% tokenizer training. But what about the **tokenizer performance** (the subject of MhKe's concern)?
> - [MhKe] We added a **new ablation on GeoBPE-Transfer data efficiency** (Table 4), varying the fraction of supervised training data (20%, 40%, 60%, 80%, 100%) with fixed val/test splits, including full tables for:
>   - Functional site prediction (AUROC) across tasks and folds.
>   - Physicochemical property prediction (Spearman’s ρ).
>   - Structure property prediction (macro F1).
> - [MhKe] We added a new **1% tokenizer-training ablation** showing that GeoBPE’s vocabulary training *is data-efficient* (Table 5).
>   - 1% data is sufficient for the vocabulary to achieve good performance *when used OOD* but dataset compression benefits from more training data. GeoBPE maintains **noticeably higher global fidelity** (RMSD) than GeoBPE (1%) but marginally worse LDDT, indicating less sensitivity to local interactions.
>   - These results and analyses provide necessary nuance to the limited capacity comment by MhKe (presuming their initial interpretation).
>
> ### C2. Appendix C – New Secondary Structure Elements (DSSP) Analysis
>
> - [4qDK] We added a new evaluation relating GeoBPE tokens to the **8 DSSP secondary structure elements** (HGIEBTS-), reporting metrics such as mean recall, precision, F1, IoU, and segment-level scores.
> - [4qDK] Summarized that:
>   - Recall of SSEs is **very high** (≈98% block-level, ≈99% segment-level).
>   - Precision is more moderate, confirming 4qDK's intuition that GeoBPE captures SSEs but also that GeoBPE discovers additional structural motifs that *go beyond* strict SSE boundaries.
>
> ### C3. Appendix G – Hierarchy Visualizations
>
> - [3EbP, 4qDK] Redrew **concrete hierarchy visualizations** of structure tokens for the two biologically meaningful case studies, aligned stylistically with the main-text hierarchy Fig. 1. They now clearly show hierarchical grouping (e.g., loops, helices, sheets, more complex motifs) to address 3EbP's suggestion for **improved figure connection** to the technical writing and 4qDK's desire to see concrete examples of GeoBPE hierarchies.
>
> ### C4. Appendix H – Complexity of GeoBPE
>
> - [3EbP, F5Hs] Added new Table 13 of **walltimes** for the original 20-core GeoBPE training run, including walltimes per component:
>   - `_init_thresholds` (Algo 1 L1).
>   - `_init_res_tokens` (Algo 1 L2).
>   - `glue_opt_all` (Algo 1 L3).
>   - Steps throughout the run (Algo 1 L7 with periodic calls to Algo 9 L13).
>   - Each with HH:MM:SS and NCPU columns.
> - [3EbP, F5Hs] Added a new **large-scale tokenization study** in Table 12:
>   - Dataset: **550k SwissProt** structures from AF-DB (>10× pretraining structures).
>   - Hardware: **100 CPU cores**.
>   - Throughput: ≈185 structures/min.
>   - Total time: ≈2 days.
>   - Emphasize that tokenization is **embarrassingly parallel** and scales almost linearly with #cores.
> - These additions address both 3EbP's concern of scaling GeoBPE to larger datasets and 5FHs's request for walltimes.

---

> ### Author Response · Authors · 2025-12-03
> **Message to Area Chair (continued)**
>
> ### C5. New Appendix I – Robustness of GeoBPE
>
> - [F5Hs] We added Figure 11 (4 subfigures), with new plots/analysis of robustness vs:
>   - “Unusualness” (1 − TM-score from FoldSeek against PDB)
>   - CATH class/architecture (fold types).
>   - Protein size (including extremely large proteins).
> - [5FHs] We show that GeoBPE shows **no degradation** on unusual or large structures in these analyses, directly addressing 5FHs's comment about GeoBPE's "robustness to extremely large or unusual proteins".
>
> ### C6. Appendix K – Hyperparameters & New Sensitivity Ablations
>
> - [F5Hs] Following F5Hs's suggestion, we added details, extended guidelines, and accessible intuition around **hyperparameters** in the new Section K.2. We include:
>   - Hyperparameters that matter most (e.g., K/`num_p`, #iterations, glue-IK weights).
>   - Default values and a short intuition for how each affects GeoBPE’s behavior.
> - [F5Hs] 5 new **sensitivity ablations** in the new Section K.3 for:
>   - Glue-IK weights $(w_R, w_t)$ (Tables 17 & 18)
>   - K and #iterations (`num_p` and merge frequency) (Table 16)
>   - `bins` (Figure 13): controls resolution of quantization,
>   - In our experiments varying `bins` together with `num_p`, we identified three regimes of GeoBPE behavior:
>     - Too high `num_p` & `bins`: too many keys, few merges per iteration → marginal improvement.
>     - Too low `num_p` & `bins`: coarsening with excessive merges → error accumulates.
>     - Middle regime: glue optimization effectively controls drift, improving distortion before later degradation.
>   - glue_opt_every (Figure 13b): trading off accuracy and runtime complexity, tied to our complexity analysis and "Insights for efficient practice" in App. J
>   - [4qDK] free_bonds (Figure 13e): see below
> - [5FHs] Provided practical **guidelines** for choosing hyperparameters, based on the ablations (Table 15, Figure 12)
> - [4qDK] Documented the `free_bonds` hyperparameter, explaining:
>   - `free_bonds = False`: bond lengths standardized/fixed.
>   - `free_bonds = True`: bond lengths treated as free variables.
> - [4qDK] Explicitly stated that main Table 1 results use `free_bonds = False`.
> - [4qDK] Noted that the run with `free_bonds = True` yields only a small RMSD change (~1.69%), supporting 4qDK's observations that backbone bond lengths are close to fixed.
>
> ### C7. Appendix E – SSLM Evaluation
>
> - [MhKe, 3EbP] Rewrote Goals & Aims of Appendix E to frame SSLM as a **small, controlled experiment** for comparing tokenizers rather than a full-scale generative model.
>   - Revised Performance Metrics paragraph in Section 4
> - [MhKe, 3EbP] Emphasized that GeoBPE is a promising tokenizer for future large-scale PLMs in Limitations while avoiding overclaiming current SSLM as SOTA
> - [MhKe, 3EbP] Following up our previous comment where we shared the new results, we documented the **scaled-up SSLM experiment** in detail:
>   - Architecture scaling (7.3M → 65.9M parameters).
>   - Dataset scaling (48k experimental structures → 550k AF-DB SwissProt).
>   - Evaluation protocol (ProtDiff/FoldingDiff style).
>   - Final metrics and comparison to prior work.
> - [MhKe, 3EbP] Clarified the difference between the older 4–5% designability number (length prior ~214 AAs) and the new standardized protocol (length 50–128).
>
> ---
>
> ## D. Algorithms & Notation (Main Text + Appendices)
>
> - [3EbP] We added **Algorithm 10** that explicitly describes how a trained GeoBPE tokenizer tokenizes a new unseen structure (we thank 3EbP for pointing out the omission). It has the same general procedure as training (Algo 1), but without one-time initializations.
>   - It applies the **learned vocabulary and GeoPairs** rather than popping frequent pairs, consistent with standard BPE decoding by applying the lookup table.
> - [3EbP] We inserted explicit references to Algorithm 10 in the main text (importantly, in Algorithm 1 to support both learning the tokenizer and applying it on new structures); we also reference Algo 10 where needed in the Appendix (E.1, F, M), e.g. during data preprocessing and the new large-scale Tokenization Study.
>
> ---
>
> All the additions we made to the paper are highlighted in a new color so you can easily spot them. We hope you can see the extensive efforts we put in to improve the paper and leave no concern unaddressed.
>
> ## Concluding Thoughts
>
> As 4qDK wrote, GeoBPE is an original, creative, and distinct solution to the current paradigm for structure tokenization (adopted by e.g. ESM3) which is both “entirely different and at the same time simpler and more intuitive”.
>
> We believe GeoBPE is a significant advance to connecting LLMs to structural biology and should be highlighted (as an Oral) at ICLR.
>
> Thank you for your time.
>
> Sincerely,
>
> GeoBPE authors

---

### Meta-Review · Area_Chair_Vnov · 2026-01-06

**Summary:**

The reviewers were concerned about clarity, the performance of the generative model trained using the proposed tokenization, a lack of residue-wise embeddings, and whether data efficiency is the result of a lack of model capacity.

**Reviewer Concerns:**

**1. Performance of the downstream generative model.**

The authors clarified that the tokenization, not the downstream generative model, is the primary contribution. In addition, they scale up the generative model and show that it outperforms some older protein diffusion methods. It would have been better to compare to a more modern model, such as RFdiffusion or Proteus.

**2. Residue-wise embeddings*.*

The authors clarified that there are intermediate residue-wise embeddings that can be used for downstream tasks.

**3. Data efficiency or lack of model capacity?**

The reconstruction performance is sufficient either way, so I'm not convinced that this matters.

**4. Clarity.**

The authors expanded the methods to be more descriptive of the actual method. By my reading, it is still very dense, but I'm not sure there's a great way around that.

**Reviewer Scores:**

4qDK: 10 -> 10

F5Hs: 8 -> 8

MhKe: 6 -> 8

3EbP: 6 -> 8

---

### Decision · Program_Chairs · 2026-01-26

Accept (Poster)